**Subject Category:**
Biology (whole organism)

ecology

climate change, marine heatwave, marine mammals, reproduction, humpback whales

**Author for correspondence:**
R. Cartwright
e-mail: rachel.cartwright@csuci.edu

†Cesere Brothers Fine Art Photography, Paia, HI 96779, USA.

# Fluctuating reproductive rates in Hawaii's humpback whales, *Megaptera novaeangliae*, reflect recent climate anomalies in the North Pacific

R. Cartwright[1,2], A. Venema[1], V. Hernandez[1], C. Wyels[1,3], J. Cesere[1,4] and D. Cesere[1,4]

[1]The Keiki Kohola Project, Kihei, HI 96753, USA
[2]Department of Environmental Science and Resource Management, and
[3]Department of Mathematics, California State University Channel Islands, One University Drive, Camarillo, CA 93012, USA
[4]Fine Art Photography, Paia, HI 96779, USA

RC, 0000-0002-2175-574X

Alongside changing ocean temperatures and ocean chemistry, anthropogenic climate change is now impacting the fundamental processes that support marine systems. However, where natural climate aberrations mask or amplify the impacts of anthropogenic climate change, identifying key detrimental changes is challenging. In these situations, long-term, systematic field studies allow the consequences of anthropogenically driven climate change to be distinguished from the expected fluctuations in natural resources. In this study, we describe fluctuations in encounter rates for humpback whales, *Megaptera novaeangliae*, between 2008 and 2018. Encounter rates were assessed during transect surveys of the Au'Au Channel, Maui, Hawaii. Initially, rates increased, tracking projected growth rates for this population segment. Rates reached a peak in 2013, then declined through 2018. Specifically, between 2013 and 2018, mother–calf encounter rates dropped by 76.5%, suggesting a rapid reduction in the reproductive rate of the newly designated Hawaii Distinct Population Segment of humpback whales during this time. As this decline coincided with changes in the Pacific decadal oscillation, the development of the NE Pacific marine heat wave and the evolution of the 2016 El Niño, this may be another example of the impact of this potent trifecta of climatic events within the North Pacific.

# 1. Introduction

As global atmospheric carbon dioxide levels pass the 400 ppm threshold [1], the biological impacts of climate change on marine systems are becoming increasingly widespread [2]. Within the marine realm, changes in ocean temperature and ocean chemistry are generally recognized as the primary direct consequences of climate change [3,4]; however, mounting evidence indicates that the fundamental processes which support marine systems are also being impacted [2,5–7]. For some marine organisms and ecosystems, these changes may signal the onset of a downward spiral from which recovery is unlikely [8–10]. For other marine fauna, the eventual outcome of these changes may be harder to predict [11,12] or could be beneficial [13–15]. Accurately predicting possible outcomes is challenging, especially where naturally occurring climate anomalies and fluctuations may act synergistically, potentially amplifying or masking perturbations associated with anthropogenically driven climate change (e.g. [16,17]). Additionally, while oceanographic conditions have been closely monitored in recent years, the required complementary long-term datasets documenting marine biological resources are comparatively sparse [2,18]. One possible solution is to mine past studies and assemble data that can be used to investigate the links between marine resources and environmental forcing. This key information can then be applied to ensure that management strategies accurately target the most detrimental impacts of climate change.

Within the North Pacific system, the last decade may be characterized as a period of pronounced climate variability. At high latitudes, mean sea surface temperatures (SSTs) have been rising steadily (estimated rate $0.7°C$ decade$^{-1}$; [19]) while local increases in SSTs have been even more pronounced: For example, between 2015 and 2017, seasonal summer temperature anomalies of 4 to $7°C$ were reported for the Barents and Chukchi Seas [19]. As sea temperatures rise, Arctic sea-ice levels are falling and a profoundly different climate regime is emerging across the Pacific Arctic [7].

At lower latitudes, increases in mean SSTs in the North Pacific have been more moderate ($0.12°C$ decade$^{-1}$; [20]); however, any potential stability has been eclipsed by a trifecta of other climate anomalies. In the summer of 2014, the Pacific decadal oscillation (PDO), a basin-wide system that acts at a multi-decadal level [21], flipped from a strong, consistent negative phase to a pronounced positive phase [22,23]. In its negative phase, the PDO is characterized by cooler water temperatures across the Central Pacific and strong upwellings in coastal waters along the Eastern North Pacific [24]. During a positive phase, SSTs rise and coastal upwellings weaken. Coinciding with this transition, an additional oceanic anomaly, comprising a massive lens of warm water, began to develop in the NE Pacific. Originally, the anomaly was centred in the offshore waters of the Gulf of Alaska; however, during the summer of 2014, it began to move east, quickly spreading along the shelf of North America and coastal Alaska, and acquiring the widely used nickname 'the blob' [25]. By the winter of 2014, associated SST anomalies of greater than $+3°C$ were reported [17,26]. The summer of 2015 then saw rising SSTs in the West-Central Pacific, signs associated with the beginning of a strong El Niño/southern oscillation (ENSO) event [27]. The typical hallmarks of these 2–3-year systems also include warming SSTs and reduced coastal upwellings; potentially, these effects further amplified the ongoing anomalies already playing out in Central North Pacific.

To date, a wide range of mass mortality events and other biological disruptions across the North Pacific and the Gulf of Alaska have been causally connected to these unusual conditions [17,26]. The warming of coastal waters associated with all three of these anomalies led to increases in rainfall and freshwater coastal run-off [28], and reduced surface winds [25]. As a result, stratification increased, minimizing coastal upwellings. Nutrient transport into the mixed layer was then suppressed, leading to extremely low productivity, as evidenced by low chlorophyll levels beginning in the winter of 2014 [28]. Compounding the inherent challenges associated with low productivity levels, horizontal advection, whereby cool water species migrate north and warm water species expand their range, potentially triggered a range of cascading impacts [25]. So far, casualties associated with these disruptions include a mass die-off of common murres, *Uria aalge*, along with the northern coastline of the Gulf and widespread mortalities in tufted puffins (*Fratercula cirrhata*) in the Bering Sea [29]. A large-scale mortality event for Cassin's auklets (*Ptychoramphus aleuticus*) along the Pacific Northwest coastline has also been attributed to this combination of unusual conditions [30].

For migratory mysticetes such as the humpback whale, *Megaptera novaeangliae*, fitness and success is entirely dependent on the availability of adequate prey resources on high-latitude feeding grounds. As capital breeders, migratory mysticetes exploit high-latitude prey resources during summertime feeding seasons, then depend entirely on stored energy reserves to support seasonal migration and wintertime breeding activities in low-latitude regions [31]. For female mysticetes, the successful completion of each stage of reproduction is contingent upon the adequate availability of stored energy reserves. Prior to pregnancy, an increase in stored energy reserves has been detected in several

migratory mysticetes (e.g. grey whales, *Eschrichtius robustus* [32], fin whales, *Balaenoptera physalus* [33] and North Atlantic right whales, *Eubalaena glacialis* [34]). Based on comparative studies of other large mammals, potentially this triggers ovulation via the release of leptin from adipose cells. Leptin then orchestrates the secretion of gonodotrophin-releasing hormone and luteinizing hormone from the hypothalamus and the pituitary, which in turn stimulate ovulation [35]. Continuing through the post-conception period, accumulated energy reserves support fetal growth and development; maternal mysticetes typically use up to 25% of these stores during this period [36]. Subsequently, during the postnatal period, remaining reserves are mobilized to meet the demands of early lactation. Changes in body shape indicate that maternal females typically use a further 20–35% of their stored reserves during this time [34,37].

Potentially reflecting these elevated energetic requirements, multiple studies have demonstrated clear connections between mysticete reproductive rates, nutritional resources and oceanic conditions. For example, in grey whales, reproductive rates increased following seasons in which sea-ice conditions extended temporal access to preferred feeding grounds in the Bering Sea [38,39]. Once climatic conditions changed and access to feeding regions was limited, reproductive rates declined [38]. Similarly, in North Atlantic right whales, notable increases in reproductive rates during the 1990s were closely matched to increases in the availability of their preferred prey, *Calanus finmarchicus*. These prey increases were driven by favourable oceanic conditions in the Gulf of Maine, which were in turn related to climatic anomalies in the near Arctic [40]. When a distinct climatic shift altered oceanic conditions in the Gulf of Maine, reducing the availability of *C. finmarchicus*, North Atlantic right whale reproductive rates also went into decline [41,42]. Additional studies describe both short- and long-term periodicity in these fluctuations, ranging from 1 to 6 years previous depending on the life cycle of the prey species [43]. Taken cumulatively, these studies elucidate the mechanisms through which mysticete reproductive rates respond to prey availability and oceanic conditions.

In this study, we provide details of a decade-long study documenting fluctuations in the reproductive rates of humpback whales in the waters around Maui, Hawaii. These whales compose the Hawaii distinct population segment (DPS) of humpback whales, a new definition that broadly describes the portion of North Pacific humpback whales that use Hawaiian waters as a breeding ground [44]. Current estimates suggest the Hawaii DPS comprises around 60% of the larger North Pacific population [44,45]. Our study was conducted over an 11-year period from 2008 to 2018, in the Au'Au Channel between West Maui and Lanai. This region comprises primary nursery habitat and is used by approximately 85% of humpback whale mother and calf pairs in Hawaiian waters over the winter breeding season [46–49]. Data from the first time period (2008–2010) stem from a previously published study [47] but are used here to extend the time series, as advocated by Doney *et al*. [2].

At the outset, the initial goal of this study was to detect seasonal changes in habitat use within the study area. As unforeseen fluctuations in encounter rates became increasingly apparent, the study was extended to capture changes in comparative encounter rates, and between 2013 and 2018, we documented a clearly evident decline in the numbers of mother–calf pairs encountered within the study area. We review a range of possible scenarios that could account for these observations. These include changing patterns of habitat use, or the possibility that this population segment may have reached, or even surpassed its carrying capacity. Finally, we compare the changes in encounter rates to oceanographic indicators. Using models that incorporate key oceanographic indices to capture the recent climate anomalies in the North Pacific, we examine potential linkages between these recent events and the fluctuations in reproductive rates in the Hawaii DPS documented in this study.

# 2. Material and methods

## 2.1. Study site

The study was conducted in shoreline to mid-channel waters, along with the eastern portion of the Au'Au Channel, West Maui (approx. 20°52′ N, 156°40′ W). The Au'Au Channel lies between the islands of Maui and Lanai and features gently sloping shoreline gradients, maximum water depths of approximately 150 m and complex mid-channel topography that includes sea mounts and ridgelines, interspersed between steep-sided sandy basins [50]. The study area (figure 1) was designed to include the range of habitats available in this area, extending from the Maui shoreline to either the mid- or deepest point of the channel at each minute of latitude, whichever lay furthest from the Maui shoreline (these locations fell between 8 and 10 km offshore). Northern and southern limits were set within the lee provided by the

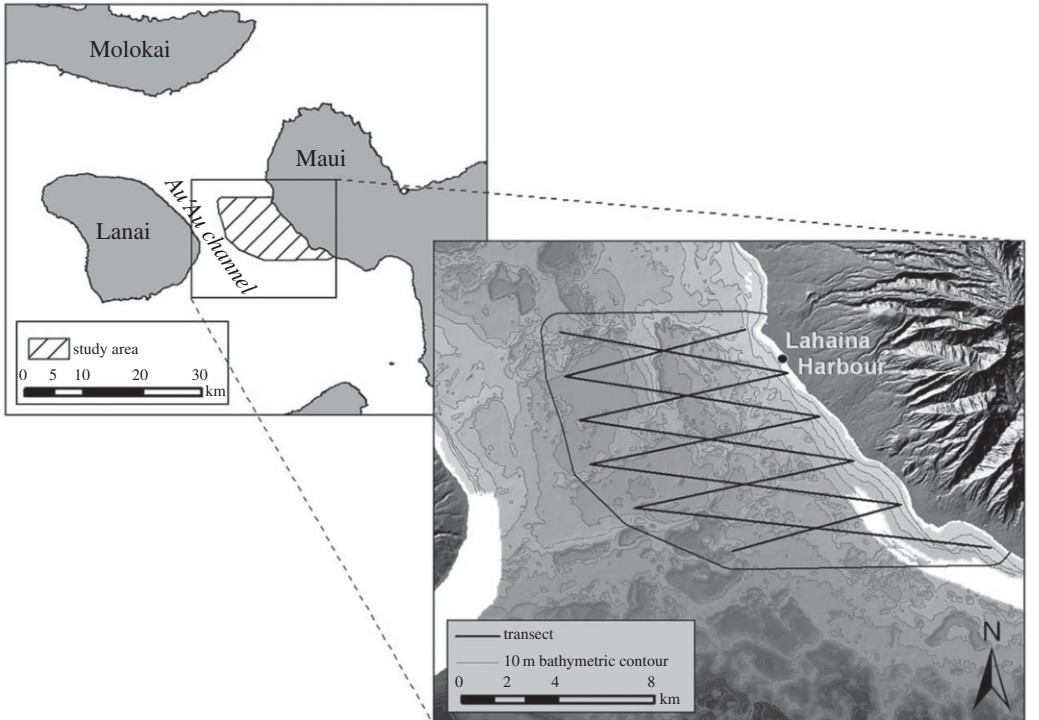

**Figure 1.** The study area. Figure prepared by Kristen LaBonte.

West Maui Mountains, thereby minimizing local variations in sightability and sea state across the study area. Lahaina Small Boat Harbor lies within the study area; this is a key hub for local whale watching and other ocean-tourism-related activities. The study area in total covered 124.5 $km^{-2}$.

## 2.2. Timing

The study spans the wintertime (January through March) season, from 2008 to 2018. The precise timing and extent of the surveys varied between years (table 1; electronic supplementary material, table S3). In 2008–2010, all surveys were conducted in the latter portion of the season (mid-March onwards). These results have been published previously [47] and were used to establish maternal patterns of habitat use in the region. Over 2013 and 2014, surveys were conducted during 3 two-week periods, in mid-January, mid-February and mid-March. Survey protocols used in 2008 to 2010 were repeated in 2013 and 2014 to allow comparisons of the results. The aim of these surveys was to capture any seasonal or temporal variability in patterns of habitat use. Surveys were reinstated in 2016, but limited to late season (mid-March) due to poor weather. In 2017 and 2018, surveys were once again conducted during 3 two-week periods, in mid-January, mid-February and mid-March, again using the 2008–2010 survey protocols. All surveys were restricted to favourable sea conditions (Beaufort less than or equal to 3) in order to ensure consistency in weather/sightability conditions.

## 2.3. Survey design

A system of parallel waypoints at 1 min of latitude intervals across the study area was established, as shown in figure 1 (see electronic supplementary material, table S1 for waypoints list). Inner waypoints were located between 250 and 500 m from the nearest shoreline depending on local topography; outer waypoints were located at the deepest or mid-point of the channel, whichever lay further offshore. Surveys were conducted across the study area along equally spaced zig-zag sampling transects set at 2° intervals between these waypoints. This ensures an equal probability of coverage across the site [51].

Daily starting points were chosen based on prevailing weather, in order to ensure consistent sighting conditions while still maintaining proportional coverage of different habitat types across the study area. Following the protocols established by Strindberg & Buckland [51], as long as only complete transects are included, completed transect legs between waypoints comprise independent samples. While more stringent methods are required for establishing abundance estimates, this survey method provided

**Table 1.** Survey dates for transect surveys conducted in the Au'Au Channel, Maui, 2008–2018.

| season | year | timing of surveys (start date – end date) | Mean Julian date (based on 1 Jan) |
|---|---|---|---|
| early | 2013 – 2014 | 18 Jan – 5 Feb | 24 |
| | 2017 – 2018 | 15 Jan – 6 Feb | 25 |
| mid | 2013 – 2014 | 19 Jan – 2 March | 55 |
| | 2017 – 2018 | 15 Feb – 25 Feb | 52 |
| late | 2008 – 2010 | 16 March – 29 March | 81 |
| | 2013 – 2014 | 18 March – 27 March | 81 |
| | 2016 | 19 March – 22 March | 80 |
| | 2017 – 2018 | 19 March – 22 March | 80 |

comparative encounter rates within the study area, and as identical methods dating back to 2008 were applied, encounter rates could be compared between different time intervals.

Survey effort varied between years over the course of the study (see electronic supplementary material, table S3 for full details), reflecting the different goals of the study during each time period. Within the first survey period (2008–2010), the primary goal of the study was to establish habitat use patterns within the study area, and in order to provide an adequate sample size to ensure appropriate power during data analysis, multiple surveys of each transect line were conducted (see Cartwright et al. [47]). During 2013–2014, the primary goal of this study was to detect seasonal changes in habitat use. To meet the requirements of this study, surveys were set up to ensure consistent coverage within the study area during early, mid- and late season over a two-year period. Over the span of the two seasons, the entire set of zig-zag transect lines (86.7 km) was covered within each of 3 two-week windows. Cumulatively, this provided three replicates of the entire set of transect lines, over a 2-year time frame (2013–2014). In 2016, the surveys were reinstated with the primary goal being to quantify temporal changes in sighting rates between years. Essential to this goal was the maintenance of consistent methods, so that the data could be compared with baseline data compiled previously. Based on distances covered within each season during the 2013–2014 surveys, a goal of approximately 50 km coverage along the original lines was chosen, as this would provide comparable data to that obtained during the 2013–2014 surveys (see electronic supplementary material, table S3). Only completed transect lines were included. This ensures that sightings are proportional to the available habitat in the area, as advocated by Strindberg & Buckland [51], but led to a small amount of variability in total distance travelled. The two-week time windows used in 2013–2014 were adopted as target time periods. To balance the competing goals of maximizing the sample size during this period of much lower numbers while still replicating the basic survey design established previously, each zig-zag transect was surveyed at least once. If the goal of covering 50 km had not been met, a single additional line was covered, but no line was surveyed more than two times in any sample period.

Throughout the full extent of the study, the same two survey vessels were used (a 6 m and an 8 m powerboat). Vessels travelled at approximately 9 km h$^{-1}$ (five knots) along the survey lines, and the same research team members supervised the collection of field data. Two experienced and fully trained designated naked-eye observers scanned on opposite sides of the vessel; any sightings within 90° on either side of the forward bow and within an estimated 1 km to either side of the survey line were recorded. Locations of sighted whales were recorded after the whale(s) left the surface, as latitude and longitude waypoints, using handheld GPS units in the 2008–2010 study and on GPS enabled Ipads from 2013 onwards. Generally, as humpback whales dive they leave a footprint (a vortex of flattened water) that persists at the surface, so wherever possible, this was used as a marker. Group composition was established following protocols described in Cartwright et al. [47]. In brief, calves were identified based on comparative body size of less than one-third of the maternal female's (mother's) body length, the mother was recognized by her close association with the calf. All other whales within two body lengths of the mother or other whales in the group and moving in a coordinated pattern were assumed to be associated and designated as escorts to the mother and calf group. These are presumed to be male whales, associated in pursuit of potentially seeking mating opportunities with receptive females [52,53]. All other individuals were identified as adults of unknown gender.

Groups containing calves spend more time at the surface, so detection probability, g(0), was probably slightly higher for mother–calf pairs versus adult animals (based on surface versus submerged time estimates in Mobley *et al.* [54], travelling at 9.4 km h$^{-1}$, mean g(0) = 0.313 for adult groups versus 0.360 for mother–calf groups). Detection probability would also vary slightly with group size; consequently, the results presented here reflect relative, not absolute densities of whales in the region. Still, as effective strip width for humpback whales on boat based surveys in Californian waters with a set speed of five knots has been previously estimated as 3.2 km [55], we assume that sightability within a 1 km strip width within Hawaiian waters would approach 100%. Fluke photo IDs (as per [56]) and surface images documenting body markings, lesions and other scars were compiled for all sighted animals and used post hoc to eliminate any chance of pseudo replication over the course of the day, between survey vessels or within regions of overlap at the beginning or the end of any successive transects. Any whales that were identified as resights within the same day, either in the field during data collection or during post hoc analysis, were recorded only once, at the first encounter. Resights between different days were rare (detected in less than 2% of sightings) and these were included in the dataset, as the focal estimated rate in this study was not an abundance estimate, rather the goal was to estimate the daily encounter rate per kilometre travelled.

In 2018, in response to exceptionally low mother–calf group encounter rates within the study area and anecdotal reports from local whale-watching vessels of higher numbers of whales in mid-channel waters, a small set of ad hoc additional surveys were conducted running along transects between the outer waypoints of the study area (see electronic supplementary material, table S4). Encounter rates for these surveys were not incorporated into the analysis of habitat use nor included in yearly encounter rates for the focal study area. They are provided here simply as an anecdotal set of observations that might potentially inform future in-depth studies in this area. This outer transect bisected the deeper, mid-channel waters of the Au'Au Channel. Survey protocols were consistent with those used in the main study area; a complete mid-channel transect between odd-numbered transects (1–11) was conducted during the early, mid and late season time windows in 2018.

## 2.4. Data analysis

### 2.4.1. Spatial analysis

A geographic information system (GIS) model was constructed using ArcGIS 10.5 (Environmental Systems Research Institute (ESRI)). A base map was obtained from ESRI (https://services.arcgisonline.com/ArcGIS/rest/services/Ocean/World_Ocean_Base/MapServer), and coastline data came from the Hawaii Data Clearinghouse. Water depth was obtained from the Main Hawaiian Islands Multibeam Synthesis website and incorporated as a 50 m bathymetric grid (http://www.soest.hawaii.edu/HMRG/Multibeam/index.php). A 750 m buffer constructed around the survey line provided coverage of 86% of the study area without overlap between mid-sections of adjacent transects. As sightings within an estimated 1 km had originally been recorded, this also reduced any potential edge effect, ensuring that all sightings across the width of the transect strip were captured. Sightings that fell beyond the buffer were discarded, as were sightings from incomplete transects. Although this did reduce the sample size slightly, Strindberg & Buckland [51] advocate these steps as a method of maintaining equal probability coverage across the survey area. For each encounter, water depths were compiled using the extract values to points function within the spatial analyst toolbox and distance from shore was obtained using the near function under the proximity tool within the analysis toolbox.

### 2.4.2. Encounter rates

Encounter rates were calculated for all surveys, as the number of individual whales encountered (i.e. the total number of whales in the group, including any calves) and the numbers of groups encountered per kilometre travelled along the transect lines. Encounter rates were classified by year of sighting and sub-classed by season (early, mid-, late), and by social composition (mother–calf (MC) versus adult (A) groups), based on the presence or absence of a calf in the group.

### 2.4.3. Oceanographic conditions

Satellite-derived SSTs for the study area were accessed via https://coastwatch.pfeg.noaa.gov/erddap. The Multi-scale Ultra-high Resolution (MUR) SST Analysis fv04.1, Global, 0.01°, 2002–present dataset

(available at https://podaac.jpl.nasa.gov/Multi-scale_Ultra-high_Resolution_MUR-SST) was used, as this provided high-resolution SST data for the study area over the entire time span of the study. First, monthly mean composite SSTs were acquired for the Au'Au Channel (20.7° N to 21.0° N; −156.9° E to −156.6° E) and used to determine the variability in local wintertime (January to March) SSTs over the course of the study. Subsequently, daily SSTs were compiled for the two-week window within which the surveys had been conducted each year (according to dates shown in table 1) and used to calculate mean estimates of SSTs for each unique survey period.

Monthly index values for the PDO were obtained at http://research.jisao.washington.edu/pdo/PDO.latest.txt; this index provides standardized monthly values for the PDO, derived as the leading principal component of monthly SST anomalies in the North Pacific, pole-ward of 20° N [21]. El Niño southern oscillation (ENSO) data were obtained from NOAA (National Ocean and Atmospheric Administration) at http://origin.cpc.ncep.noaa.gov/products/analysis_monitoring/ensostuff/ONI_v5.php. This index, referred to as the ONI index (ONI: Oceanic Niño Index), is the operational definition used by NOAA to predict the likely development and persistence of El Niño weather events. Additionally, we compiled indices from the TNI index database (Trans-Niño Index; https://www.esrl.noaa.gov/psd/data/correlation/tni.data). This index measures the gradient in SST anomalies between the Central and Eastern Equatorial Pacific and is used specifically to identify Central Pacific El Niño events [57,58]. First, these indices were reviewed to establish the basic trends in oceanic conditions over the duration of the study. Next, reflecting the findings of Seyboth *et al.* [59] and Bengtson Nash *et al.* [60], indices were advanced by 1 and then 2 years; monthly values from May to October were compiled to provide a mean composite annual value that reflected North Pacific oceanic conditions in previous feeding seasons (i.e. for example, encounter rates for 2008 were associated with oceanic indices in 2007, then the 2006 feeding season). These advanced indices were then used in the models constructed (see below).

### 2.4.4. Statistical analysis

Statistical analyses were conducted using SPSS v. 24, and R v. 3.4.0 [61]). Where standard parametric and non-parametric inferential tests were used, significance was set at 0.05 and Bonferroni corrections included where multiple identical tests were conducted. Where models were constructed, model performance was evaluated based on ΔAICc values and deviance explained, as statistical tests and associated *p*-values may be unreliable when used with smaller datasets, such as this [62].

Changes in habitat use within the study area between years, social groups and seasons were investigated using survey data collected throughout the season in 2013–2014. Survey data from 2017 to 2018 were also incorporated as these had been collected following the same protocols. Variables were incorporated into a multi-variate ANOVA test (MANOVA); depth of water and distance from shore for each group were incorporated as dependent variables and season (early, mid- and late season), study period (2013–2014 and 2017–2018) and group composition (mother–calf (MC) versus adult groups (A)) were used as fixed factors. After confirming that all assumptions of the multi-variate MANOVA were met, the interpretation was based on Wilks' lambda statistics. To detect any long-term trends over the full extent of the study, a second analysis was conducted using group locations from 2008 to 2018. It should be noted this analysis included only late season sightings, so fixed factors were limited to study period (2008–2010, 2013–2014 and 2017–2018) and group composition.

Variations in encounter rates were investigated using a series of generalized additive models (GAMs; [63]), constructed using the 'mgcv' package for R [64]. In the first sequence of models, temporal trends were investigated. Separate models were constructed for numbers of individual whales encountered and for mother–calf groups; encounter rates were used as the response variable while season and year were used as explanatory variables. Season was included as a factor. Based on the examination of exploratory scatter plots, the year was included as a smoothed variable to allow detection of nonlinear trends. Thin plate penalized regression splines were used (this is the default setting in 'mgcv') and as the sample dataset is small, the number of knots was constrained to 4, as advocated for a sample of this size [65].

In a second set of models, the impact of key oceanic parameters on variations in encounter rates was explored. Encounter rates for individual whales, mother–calf groups and adult-only groups comprised the response variables, with separate models constructed for each variable. Local SSTs, PDO, ONI and TNI indices were incorporated as explanatory variables. For the local SST, mean estimates for the specific two-week window in which the surveys were conducted (described in the Material and methods section above) were used. For the three oceanic indices (PDO, ONI and TNI), the composite mean summertime values (May to October) were compiled. Initially, these summertime composite

values were then advanced by 1 year, so that they reflected conditions in the previous feeding season. In a follow-up analysis, these data were advanced by two seasons, so encounter rates for 2008 were compared with composite mean summertime values for 2007 and then 2006 and so forth. Prior to testing, the data were screened for normality, the detection of outliers and for collinearity among the explanatory variables; the data were normal and no outliers were present. Indications of collinearity were limited to a moderate, negative correlation between the TNI indices and local SSTs in Hawaii ($r = -0.753$); however, the significance of this outcome did not withstand correction for multiple testing ($p = 0.031$, $\alpha/k = 0.0125$). All other $r$ values fell below 0.6. As in the first set of models, continuous variables were smoothed to allow for the detection of nonlinear trends, the default setting of thin plate penalized regression splines was used and the number of knots was constrained to 4 for each model [65]. Owing to the limited sample size, only single factor models were constructed and interaction terms were not included [65].

# 3. Results

Across the 11-year period from 2008 to 2018, a total distance of 1334.2 km was covered and the locations of 366 groups that included 875 whales were recorded. Between-year differences in the mean Julian dates for surveys were less than or equal to 3 days (table 1). Distance surveyed varied between years, as shown in table 3; electronic supplementary material, table S3. In 2008–2010, surveys covered a distance of 731 km but were limited to late season and a total of 148 groups were sighted. Surveys conducted at two-week intervals throughout 2013–2014 covered a total distance of 260.2 km and 149 groups were sighted. During 2017–2018, surveys were again conducted at two-week intervals throughout the season. A total distance of 292.5 km was surveyed and 65 groups were encountered.

## 3.1. Fine-scale patterns of habitat use

Comparing sightings recorded throughout the season in 2013–2014 and 2017–2018, group locations (represented by depth of water and distance from shore) varied according to season but not with the survey time period or group composition (MANOVA for season: $F4 = 3.271$, $p = 0.012$; for survey time period: $F2 = 1.022$, $p = 0.362$; and for group composition: $F4 = 2.163$, $p = 0.118$). Within the effect of season, post hoc testing indicated that depth of water was not significantly different between seasons ($F2 = 1.572$, $p = 0.210$), but the distance from shore was significantly different ($F2 = 6.517$, $p = 0.002$). Groups were encountered significantly closer to shore during mid-season versus early season (using pooled data for all social groups; mean distance from shore = 5.8 km in early season and 4.3 km in mid-season, $\alpha/k = 0.0166$; post hoc Tukey; $p = 0.012$), but differences between early versus late season ($p = 0.033$) and mid- versus late season ($p = 0.990$) were not significant. When two-way and three-way interactions were examined, the only significant interaction included all three factors, but this did not withstand corrections for multiple testing ($\alpha/k = 0.0125$; $F4 = 2.637$, $p = 0.034$). Repeating the analysis over the full extent of the study (2008–2018; late season data only), there were no significant differences in group locations between different survey time periods ($F4 = 1.604$, $p = 0.173$), according to group composition ($F = 0.891$, $p = 0.412$) or associated with the interaction between these factors ($F = 1.478$, $p = 0.208$) (table 2).

## 3.2. Encounter rates

### 3.2.1. All groups, individual counts

Assessing encounter rates for individual whales, trends over the course of the study were nonlinear. Late season sightings encompassed the longest time range, spanning from 2008 to 2018. Over this 11-year period, encounter rates increased initially, from 0.42 whales km$^{-1}$ in 2008 to a peak of 1.12 whales km$^{-1}$ in 2013, and then declined to a low of 0.14 whales km$^{-1}$ by 2018 (figure 2; electronic supplementary material, table S3). However, it should be noted, these estimates reflect late season encounter rates and trends may be impacted by changing seasonal peaks. Comparing data collected in 2013, 2014, 2017 and 2018 may be more robust as surveys were conducted throughout the season. Comparing mean encounter rates for the entire season across two-year brackets (2013–2014 versus 2017–2018), encounter rates fell by 63% (from 1.39 whales km$^{-1}$ in 2013–2014 to 0.51 whales km$^{-1}$ in 2017–2018). Comparing mean encounter rates for individual years (2013, 2014,

**Table 2.** Locations of humpback whales classified by social composition in the Au'Au Channel, Maui, between 2008 and 2018. When all groups were pooled, mean distance to shore fell significantly between early and mid-season (a versus b; Wilks' lambda statistic (MANOVA): $F4 = 3.271$, $p = 0.012$; post hoc Tukey: $p = 0.012$, $\alpha/k = 0.0166$).

| season | year | water depth (mean (s.d.) m) | | distance from shore (mean (s.d.) km) | |
|---|---|---|---|---|---|
| | | mother – calf groups | adult groups | mother – calf groups | adult groups |
| early | 2013 – 2014 | 59.0 (13.4) | 61.7 (9.7) | 4.68 (2.31)[a] | 6.03 (2.04)[a] |
| | 2017 – 2018 | 72.6 (1.95) | 59.4 (12.6) | 6.97 (0.95)[a] | 5.74 (1.83)[a] |
| mid | 2013 – 2014 | 61.5 (12.4) | 59.8 (13.9) | 4.52 (1.86)[b] | 4.89 (1.88)[b] |
| | 2017 – 2018 | 50.5 (15.7) | 62.0 (10.1) | 2.66 (2.26)[b] | 5.05 (2.12)[b] |
| late | 2008 – 2010 | 58.9 (14.3) | 62.8 (10.91) | 4.65 (2.27) | 5.03 (1.97) |
| | 2013 – 2014 | 53.0 (14.2) | 57.1 (18.5) | 3.59 (2.18) | 5.57 (2.67) |
| | 2017 – 2018 | 66.3 (9.6) | 61.8 (15.8) | 6.16 (2.08) | 5.73 (1.89) |

**Table 3.** Encounter rates for humpback whale groups classified by group composition, sighted during transect surveys conducted in the Au'Au Channel, Maui, 2008 – 2018.

| year | all groups | calf groups | % calf groups | distance surveyed (km) | encounter rate (mother – calf groups km$^{-1}$) | encounter rate (adult groups km$^{-1}$) |
|---|---|---|---|---|---|---|
| brackets | | | | | | |
| 2013 – 2014 | 149 | 89 | 59.7 | 260.2 | 0.34 | 0.23 |
| 2017 – 2018 | 65 | 23 | 35.5 | 292.5 | 0.08 | 0.14 |
| entire seasons | | | | | | |
| 2013 | 115 | 68 | 59.1 | 168.0 | 0.40 | 0.28 |
| 2014 | 34 | 21 | 61.8 | 92.2 | 0.23 | 0.14 |
| 2017 | 41 | 17 | 41.5 | 146.7 | 0.12 | 0.16 |
| 2018 | 24 | 6 | 25.0 | 145 | 0.04 | 0.12 |
| late season only | | | | | | |
| 2008 | 44 | 27 | 61.4 | 270.5 | 0.10 | 0.06 |
| 2009 | 77 | 42 | 54.5 | 355.3 | 0.12 | 0.10 |
| 2010 | 27 | 17 | 63.0 | 105.3 | 0.16 | 0.09 |
| 2013 | 23 | 17 | 73.9 | 52.5 | 0.32 | 0.11 |
| 2014 | 12 | 9 | 75.0 | 34.2 | 0.26 | 0.09 |
| 2016 | 4 | 3 | 75.0 | 50.3 | 0.06 | 0.02 |
| 2017 | 7 | 6 | 85.7 | 44.9 | 0.13 | 0.02 |
| 2018 | 3 | 2 | 66.7 | 49.7 | 0.04 | 0.02 |

2017 and 2018), encounter rates fell from 1.66 whales km$^{-1}$ in 2013 to 0.90 whales km$^{-1}$ in 2014, 0.63 whales km$^{-1}$ in 2017 and 0.38 whales km$^{-1}$ in 2018. With respect to seasonality, encounter rates for all individual whales typically peaked during mid-season, with the single, but notable exception of 2018 when sighting rates peaked in the early portion of the season (figure 2).

### 3.2.2. Sightings of mother – calf versus adult groups

When classified by social composition, trends varied between different groups. For mother–calf groups (MC) over the longest time span (2008–2018), late season encounter rates increased from 0.10 MC groups km$^{-1}$ in 2008 to 0.32 MC groups km$^{-1}$ in 2013, then declined to a low of

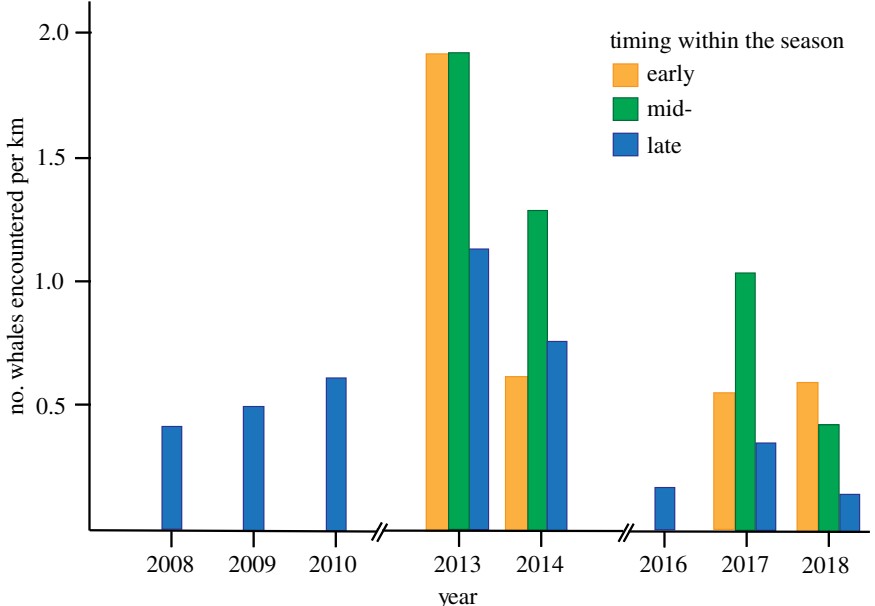

**Figure 2.** Encounter rates for humpback whales (counts of individuals) in the Au'Au Channel, Maui, between 2008 and 2018.

0.04 MC groups km$^{-1}$ in 2018. Similarly, late season encounter rates for adult groups (A) increased from 0.06 to 0.11 A groups km$^{-1}$ between 2008 and 2013, then declined to a low of 0.02 A groups km$^{-1}$ by 2018. Comparing mean encounter rates for the entire season across 2-year brackets (2013–2014 versus 2017–2018) encounter rates fell by 76.5% for MC groups (from 0.34 groups km$^{-1}$ in 2013–2014 to 0.08 groups km$^{-1}$ in 2017–2018) and by 39% for A groups (from 0.23 groups km$^{-1}$ in 2013–2014 to 0.14 groups km$^{-1}$ in 2017–2018). Comparing mean encounter rates for 2013, 2014, 2017 and 2018: for MC groups, mean annual encounter rates declined consistently between 2013 and 2018 (from 0.40 MC groups km$^{-1}$ in 2013 to 0.23 MC groups km$^{-1}$ in 2014 to 0.12 MC groups km$^{-1}$ in 2017 and 0.04 MC groups km$^{-1}$ in 2018). By contrast, adult encounter rates declined between 2013 and 2014 but then remained relatively consistent between 2014 and 2018 (0.28 A groups km$^{-1}$ in 2013, 0.14 A groups km$^{-1}$ in 2014, 0.16 A groups km$^{-1}$ in 2017 and 0.12 A groups km$^{-1}$ in 2018) (figure 3 and table 3).

### 3.2.3. Modelling results

GAMs constructed to further explore temporal and seasonal changes in encounter rates indicated that for encounter rates for all individuals, the best performing models were obtained using year as a smoothed predictor. Year explained 49.9% of the deviance for encounter rates for individual whales. The smoothed plot revealed a unimodal nonlinear trend, peaking in the winter of 2013. For encounter rates for mother–calf groups, again year performed best, explaining 64.6% of the deviance, with a similar unimodal trend over this period. Notably, season performed better in encounter rates for all individuals versus mother–calf groups (explaining 28.1 versus 7.5% of the deviance), inferring that individual encounter rates potentially fluctuated more over the season in comparison with mother–calf encounter rates. This supports the general trend evident in the data, whereby encounter rates for adult groups were notably low during late seasons from 2014 onwards (figure 4 and table 4).

## 3.3. Oceanographic conditions

### 3.3.1. Variation in local wintertime sea surface temperatures

Between 2008 and 2018, local wintertime (January to March) SSTs for the Au'Au Channel ranged from a minimum of 22.7°C during March 2009 to a maximum of 25.5°C in March 2017. There was no evidence of any linear trend (figure 5; Pearson's CC = 0.540, $p = 0.087$); however, when data were divided into years around the peak sighting year (2013), and SSTs from 2008 to 2013 were compared with SSTs from 2014 to

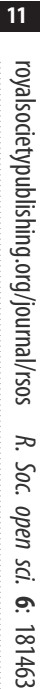

**Figure 3.** Encounter rates for (*a*) mother–calf groups and (*b*) adult groups during transect surveys of the Au'Au Channel, Maui, between 2008 and 2018.

2018, the difference between these two groups was significant (using a two sample *t*-test; $t = -2.952$, d.f. $= 9$, $p = 0.016$), with slightly higher mean temperatures in 2014 to 2018, compared with 2008 to 2013 ($0.6°C$).

### 3.3.2. Variations in system level climate anomalies

Trends in the three indices selected to represent the occurrence and strength of ocean anomalies in the North Pacific over the course of the study (the PDO, ONI and the TNI) indicated pronounced climatic changes during this time. At the beginning of the study period, the PDO index was in a strong negative phase. It diminished slightly between September 2009 and May 2010 then returned to a pronounced negative condition until the beginning of 2014, when it flipped to a positive integer. Positive PDO values were then persistently high throughout 2014, 2015 and 2016, but have fallen in the last two years. The most recent El Niño was signalled in the ONI index, which displayed a

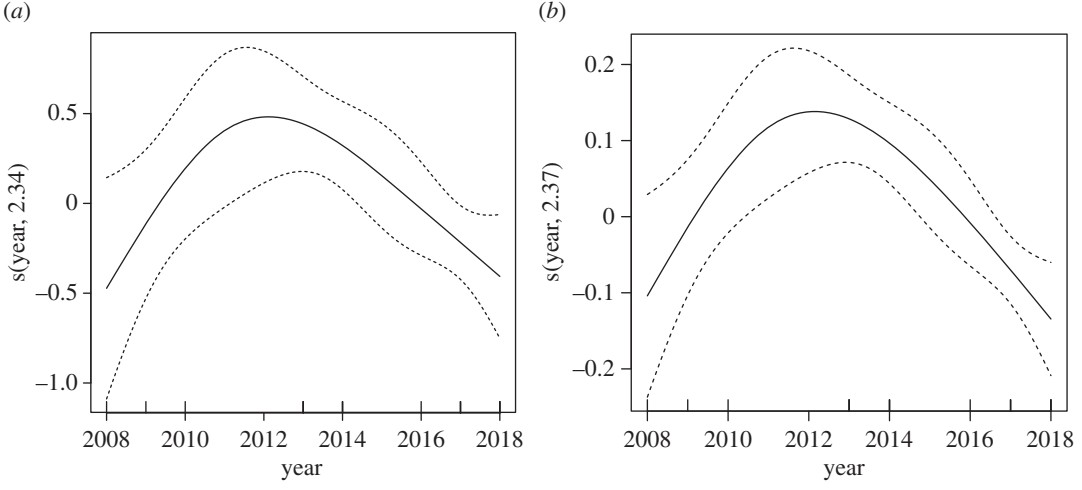

**Figure 4.** Nonlinear trends in encounter rates for (*a*) all individual whales and (*b*) mother–calf groups during transect surveys of the Au'Au Channel, Maui, between 2008 and 2018.

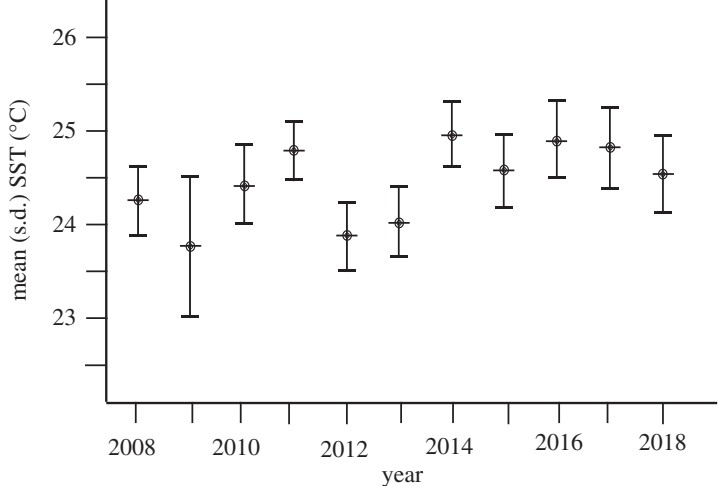

**Figure 5.** Mean composite sea surface temperatures for the winter season (January to March) for the Au'Au Channel, 2008–2018. While there was no evidence of any linear trend (Pearson's CC = 0.540, $p = 0.087$), when pre-2014 SSTs (from 2008 to 2013) are compared with post-2014 SSTs (from 2014 to 2018), the difference between these two groups was significant (0.6°C; using a two sample *t*-test; $t = -2.952$, d.f. = 9, $p = 0.016$).

**Table 4.** Generalized additive models describing temporal trends in encounter rates for humpback whales in the Au'Au Channel, Maui, Hawaii.

| explanatory variables | deviance explained (%) | AICc | ΔAICc |
|---|---|---|---|
| encounter rates: all individual whales | | | |
| s(year, 2.34) | 49.9 | 22.80 | 0.00 |
| factor (season) | 28.1 | 27.92 | 5.12 |
| encounter rates: mother–calf groups | | | |
| s(year, 2.37) | 64.6 | −29.33 | 0.00 |
| factor (season) | 7.5 | −11.71 | 14.61 |

pronounced peak in 2015. The TNI index, which predicts Central Pacific El Niño events, saw monthly values exceeding −2 during 2009 and 2010, and again in more recent years, beginning in the summer of 2013. This index typically changes prior to the onset of the broader El Niño event [57] (figure 6).

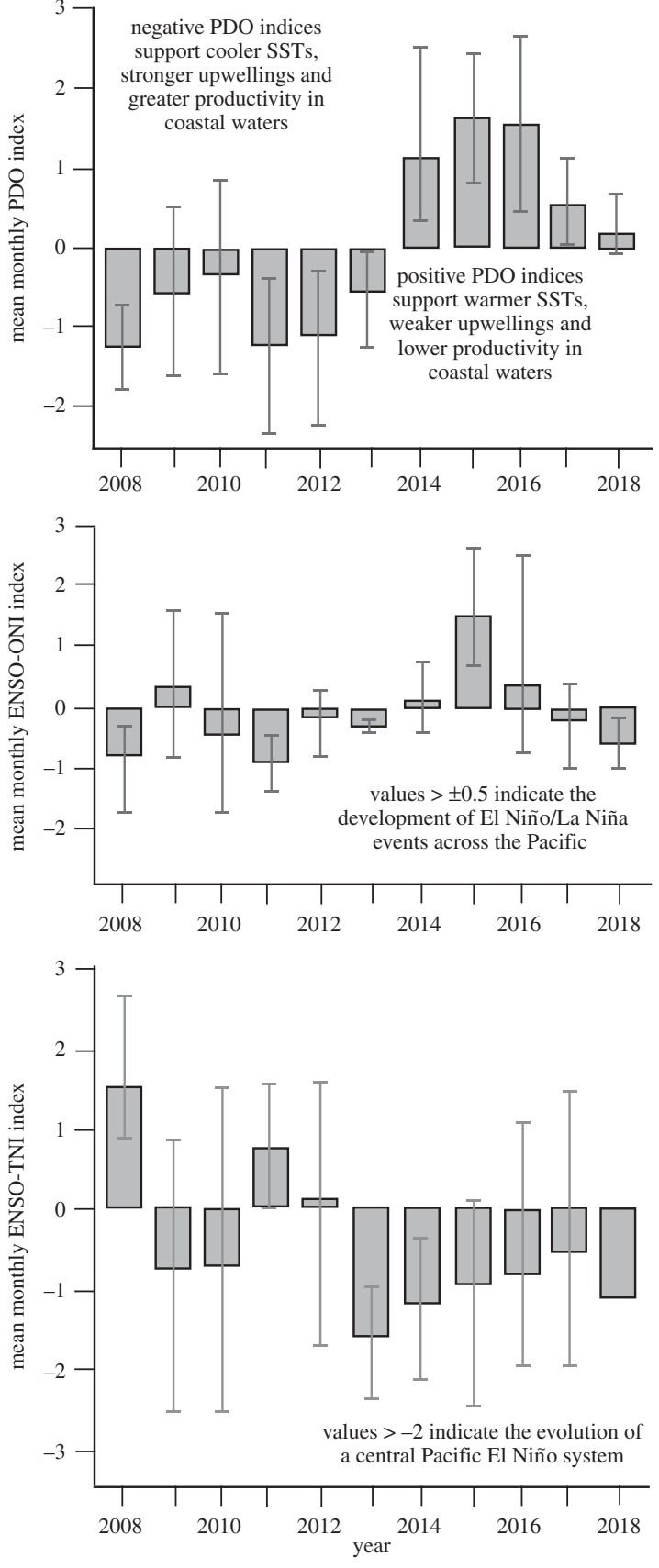

**Figure 6.** Mean annual values for key climate indices that influence oceanographic conditions in the North Pacific, between 2008 and 2018.

**Table 5.** Generalized additive models describing the influence of oceanic climate conditions in the North Pacific on encounter rates for humpback whales in the Au'Au Channel, Maui, Hawaii. SST, mean local sea surface temperature (derived during the dates of the study); PDO, Pacific decadal oscillation; ONI, Oceanic Niño Index; TNI, Trans-Niño Index (advanced as indicated).

| explanatory variables | indices advanced by 1 year | | | indices advanced by 2 years | | |
| --- | --- | --- | --- | --- | --- | --- |
| | deviance explained (%) | AICc | ΔAICc | deviance explained (%) | AICc | ΔAICc |
| encounter rates—individual whales | | | | | | |
| s(SST, ∼) | <0.01 | 29.20 | 3.31 | n.a. | n.a. | n.a. |
| s(PDO, 0.867) | 27.0 | 25.89 | 0.00 | 55.5 | 21.74 | 0.00 |
| s(ONI, 1.713) | 21.3 | 28.79 | 2.90 | 43.3 | 25.61 | 3.87 |
| s(TNI, 2.563) | 23.4 | 30.05 | 4.19 | 43.6 | 25.65 | 3.91 |
| encounter rates—mother–calf groups | | | | | | |
| s(SST, 2.58) | 21.3 | −13.07 | 6.35 | n.a. | n.a. | n.a. |
| s(PDO, 1.39) | 35.5 | −19.42 | 0.00 | 66.8 | 26.50 | 0.00 |
| s(ONI, 0) | <0.01 | −14.46 | 4.96 | 53.2 | 20.89 | 5.61 |
| s(TNI, 2.77) | 34.3 | −15.62 | 3.77 | 46.2 | 18.79 | 7.71 |
| encounter rates—adult-only groups | | | | | | |
| s(SST, ∼) | <0.01 | −21.68 | 0.27 | <0.01 | 21.47 | 0.21 |
| s(PDO, 0.38) | 6.21 | −21.70 | 0.25 | 0.85 | 21.64 | 0.04 |
| s(ONI, 1.59) | 18.2 | −21.95 | 0.00 | <0.01 | 21.68 | 0.00 |
| s(TNI, ∼) | <0.01 | −21.68 | 0.27 | 27.0 | 21.42 | 0.26 |

### 3.3.3. Modelling results

Results from the GAMs constructed to detect interplay between oceanic indicators and encounter rates indicated slightly different inferences for all individuals (including mother–calf pairs) versus mother–calf groups. Inferences also varied notably between mother–calf and adult-only groups.

Notably, in all models, the local mean SST performed poorly, explaining between 0 and 21% of the variability, inferring that encounter rates were not influenced by variation in the local SST. When oceanic indicators were advanced by 1 year, for encounter rates for individual whales, the model containing the PDO index performed the best but explained only 27% of the variability. The ENSO indices (ONI and TNI) performed similarly, explaining 21.3 and 23.4% of the deviance respectively. For encounter rates for MC groups with oceanic indicators advanced by 1 year, the PDO index explained 35.5% of the variability, the TNI index explained 34.5% of the deviance, but had a less persuasive AICc value, while the ONI index performed poorly. Notably, plots for the PDO smoothers for individual encounter rates and for mother–calf encounter rates approached an inverse, linear relationship (table 5 and figure 7a,b).

When oceanic indicators were advanced by two seasons (so 2008 encounter rates were tied to 2006 composite mean summertime values and so forth), for mother–calf groups, the performance of the models improved considerably: variability explained by the PDO increased to 66.8%. Likewise, the models for the TNI and ONI indices also improved to 53.2% and 46.2% respectively; however, plots were no longer linear, potentially suggesting a more complex association or possibly an interaction between the indices used. Unfortunately, the sample size in this study precludes testing these interactions. Encounter rates for all individual whales reflected the trends described for mother–calf pair. In contrast though, for adult-only groups, neither the 1-year and 2-year advanced indices performed well, with deviance explaining ranging from 0 (ONI) to 27% (TNI advanced by 2 years) (table 5). Potentially, this suggests that trends in encounter rates for adult-only groups are not closely aligned with these oceanic indices.

## 4. Discussion

This study describes fine-scale patterns of habitat use and changes in encounter rates for humpback whales in the waters of the Au'Au Channel, West Maui over the past 11 years. This region comprises

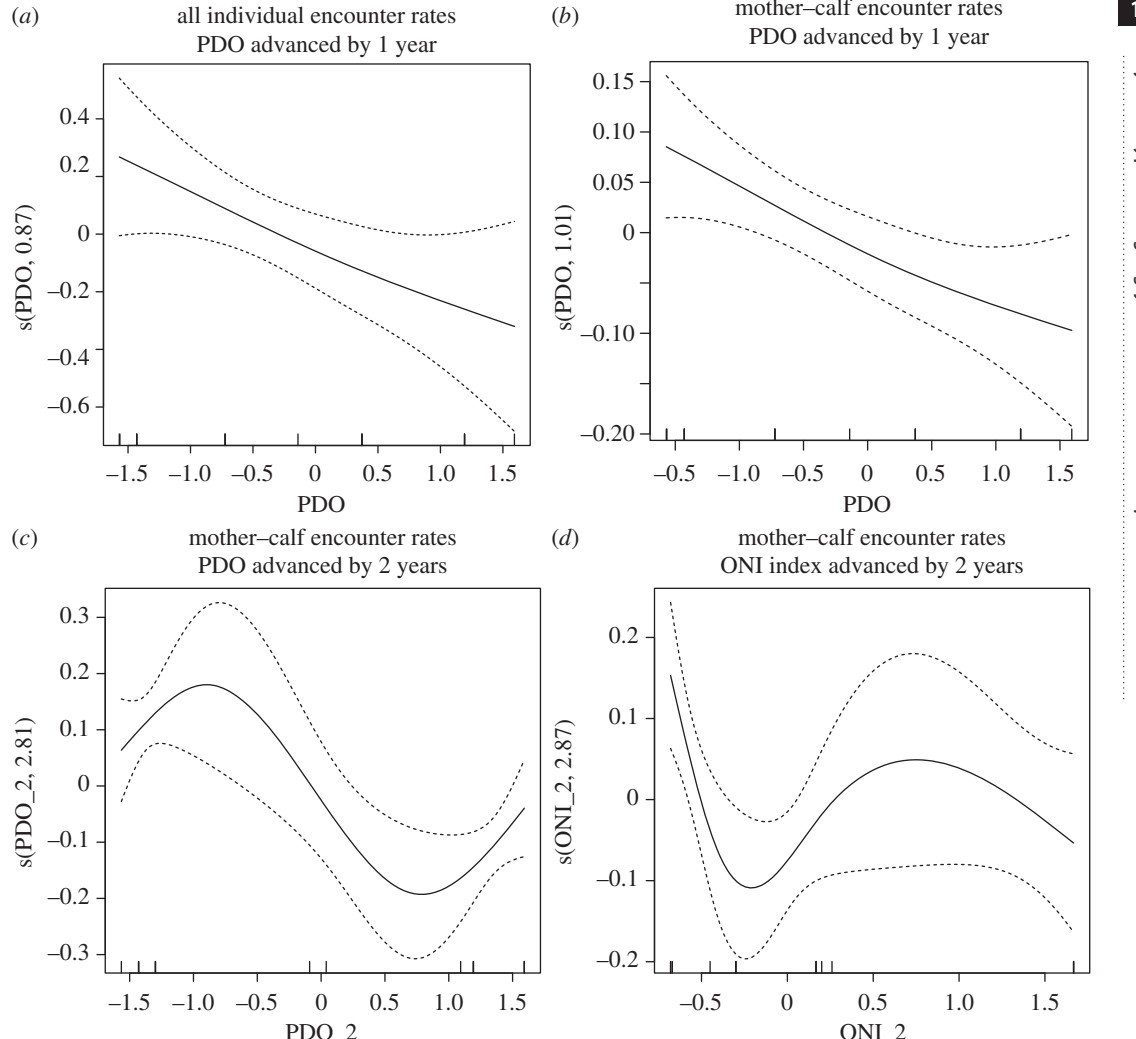

**Figure 7.** Smoothed trends in encounter rates as oceanic indices vary for all individual whales and mother–calf groups during transect surveys of the Au'Au Channel, Maui, between 2008 and 2018.

the favoured breeding area for the newly designated Hawaiian DPS [44]. Over the course of the study, while fine-scale changes in patterns of habitat use within the study area were minimal, we documented pronounced nonlinear fluctuations in encounter rates. Essentially, between 2008 and 2013 encounter rates for individual whales increased, potentially surpassing projected growth estimates for this population segment (5–6.6% per year; [45]). Individual encounter rates peaked in 2013 and then declined between 2013 and 2018. However, trends in these declines varied between different social groups. Encounter rates for adult groups dropped between 2013 and 2014, but then remained relatively consistent between 2014 and 2018. For mother–calf groups after 2013, encounter rates declined year-by-year through the end of the study period. Comparing mean encounter rates for two-year brackets (2013–2014 versus 2017–2018), encounter rates for mother–calf groups fell by 76.5%, and looking at changes between individual years, between 2013 and 2018, rates fell by 90.0% (encounter rates in 2013: 0.40 MC groups km$^{-1}$, versus 2018: 0.04 MC groups km$^{-1}$). These observations stand in stark contrast to recent reports of humpback whale baby booms documented recently in other regions [66].

## 4.1. Was there a decline in the reproductive rate of the Hawaii DPS?

The changes in encounter rates for mother–calf groups documented here potentially infer that the reproductive rate for the Hawaii DPS declined between 2013 and 2018. However, before reaching this conclusion, several key issues and questions need to be addressed. A pressing issue is to confirm that the current study was staged in an area where encounter rates for mother–calf pairs would accurately represent regional trends. The area where the study was conducted, namely the eastern portion of the

Au'Au Channel between the islands of Maui and Lanai, is currently perceived to be core mother−calf habitat within Hawaiian waters [46−49,67]. The most recent island-wide aerial surveys, though somewhat outdated, suggest that approximately 85% of mother−calf groups in the islands are found in the Au'Au Channel [46,54] and females exhibit a preference for this portion of the Au'Au Channel in the years when they calve [54]. Looking beyond this focal area, alternative studies report trends in mother−calf encounter rates that match the findings of this study: for example, a recent study that included the shorelines of Central and South Maui spanning the years 2014−2017 reported that proportions of mother−calf groups sighted fell from 54% of all group sightings in 2014 to 37% in 2017 [68]. By comparison, in this study, the proportion of mother−calf groups sighted fell from 62% of all groups in 2014 to 41% in 2017. The slight differences probably represent differences in coverage, as Currie & Stack [68] included a higher proportion of deeper, offshore waters in their study area, in comparison with this study. Moreover, looking across the island chain to the coastal waters along the Kohala coastline of the island of Hawaii, a region that is also recognized as favoured mother−calf habitat [69], a shore-based long-term study of sightings and occupancy rates in this area reports declines in the range of 75% for sightings of mother−calf pairs over the last 5 years (Hawaii Marine Mammal Consortium, S. Yin, personal communication).

While the alignment of observations from these three regional studies is notable, still, localized shifts in habitat use could account for the documented changes in encounters and sightings of mother−calf pairs. Within the study area surveyed here, preferred depths for mother−calf pairs remained consistent between late seasons from 2008 and 2018, while in the more short-term analysis, the only emerging trend was a potential relocation closer to shore during mid-season versus early and later time periods. During the 2018 season when encounter rates in the main study were exceptionally low, ad hoc observations revealed higher numbers of adult-only groups in mid-channel waters, but numbers of mother−calf pairs were consistent with sighting rates across the main study area (only 3 of 43 groups sighted included a mother−calf pair; see electronic supplementary material, table S4). Other regional studies referenced here ([68], unpublished HMMC study), included more extensive offshore/deeper waters within their focal study areas, but neither study reports any indication of changing patterns of localized habitat use.

A more large-scale relocation of mother−calf pairs within the island chain offers a further plausible explanation for changes in encounter rates in nearshore waters around Maui and the island of Hawaii. The Penguin Banks, a shallow, but the remote offshore bank that lies approximately 40 km south of the Island of Molokai, is recognized as an alternative area that is favoured by mother−calf pairs [46]. Unfortunately, reflecting the remote location and typically rough surface conditions, this region remains unsurveyed in recent years. Further afield still, within the Northwestern Hawaiian Island (NWHI) chain, recent acoustic studies have detected the presence of adult humpback whales [70]. Notably, as mother−calf pairs are acoustically cryptic [71], they would probably go undetected during acoustic monitoring. While previous researchers have suggested that protected waters within the NWHI chain were used as breeding grounds during much earlier periods (more than 300 years ago) when ocean temperatures may have been warmer [72], extracted satellite data compiled in this study indicates that water temperatures within the Au'Au Channel (at 22.7−25.5°C) currently remain well within the range used by humpback whales in breeding regions (21−28°C; [73]). Notwithstanding, further efforts to survey these more remote areas is clearly warranted at this time.

At this point, a pronounced habitat shift in mother−calf pairs within Hawaiian waters remains a possibility. However, recent sighting data from the high-latitude feeding grounds used by the Hawaii DPS provides further support for the assumption that the reproductive rate of this population segment declined between 2013 and 2018. Feeding grounds used by the Hawaii DPS include coastal regions of Northern British Columbia, Southeast Alaska, Prince William Sound, coastal regions of the Northern Gulf of Alaska and portions of Southwest Alaska [45]. Collectively, more than 80% of the Hawaii DPS head to these areas, and typically whales display a high degree of site fidelity between years [45,74]. Across these areas, sightings of mother−calf pairs declined between 2013 and 2018. In Glacier Bay, Southeast Alaska, sighting data extend back more than 30 years [74] and falling reproductive rates have recently been documented. Declines began in 2014; in 2016, park observers reported the lowest crude birth rate since studies began in 1986 and in 2017, the crude birth rate was the second lowest on record, with only two calves sighted through the entire summer [75]. Preliminary results from a survey covering a broader area of Southeast Alaska during the summer of 2017 documented only three calf sightings (SPLISH surveys; cited by Nielson et al. [75]) and further north in Prince William Sound, Southwest Alaska, no calves have been sighted since 2015 (O. Vonziegesar 2018, personal communication).

Looking ahead, long-term studies focused in high-latitude regions where high site fidelity allows for the reproductive rates of individual female whales to be documented (e.g. Glacier Bay) will provide valuable insight as these trends play out in future years. Within Hawaiian waters, more extensive studies that reach beyond the protected waters of the Au'Au Channel and other shoreline areas are necessary before an unequivocal conclusion can be reached. In the meantime though, the consistent reports of low rates of mother–calf sightings across both breeding and feeding regions certainly suggest that the reproductive rate of the Hawaii DPS declined steeply between 2013 and 2018.

## 4.2. Did this population segment reach carrying capacity?

The term carrying capacity is used in many different contexts across the biological literature [76]. In the context of this study, the term carrying capacity is being employed to refer to the size of a population when it reaches equilibrium [77]. At this point, the biotic potential of the population in terms of growth is checked by environmental resistance and the numbers within the population remain stable [77]. Current studies demonstrate that, in relation to this definition, carrying capacity is not fixed or static. Rather, carrying capacity varies both spatially and temporally, as habitat quality or the availability of the key resources fluctuate over time (e.g. [78–81]). Establishing carrying capacity is typically challenging. But as the impacts of surpassing carrying capacity are generally detrimental, both for the specific population and their habitat (see Chapman & Byron [76]), determining the status of a population with respect to its carrying capacity can be a useful step towards interpreting changes in vital rates and in developing appropriate management plans.

In large-bodied mammals with long lifespans, changes in density-dependent vital rates typically occur as populations approach carrying capacity [82]. At this point, reproductive rates generally slow down and may decline once carrying capacity has been reached [82]. To some degree, this trajectory describes the overall trend seen in this study: Beginning in 2008, sighting rates were in a period of rapid increase, potentially peaking in 2013 and then declining through 2018. It should be noted that surveys were not conducted in the 2 years prior to the peak recorded here (2011 and 2012); therefore, it is possible that the true peak could have been slightly earlier than this date. Notwithstanding, if this population segment reached, or even surpassed its carrying capacity during this period, then this provides a robust explanation that would account for nonlinear trends in sighting rates in mother–calf groups on the breeding grounds, and changes in the reproductive rate of this DPS.

However, a variety of observations suggest that the explanation may lie elsewhere. Firstly, rates of change in reproductive rates around carrying capacity are forecast to be approximately symmetrical (figure 2a, Fowler [82]). In this case, changes were asymmetrical; the rate of decline fluctuated, and ultimately far exceeded the rate of increase prior to the peak encounter rates recorded in 2013. Furthermore, Eberhardt & Siniff [83] describe a range of additional indicators that should be apparent if a population is approaching carrying capacity. These include increased aggression between mating-age adults, increased time spent foraging, increased rates of offspring mortality, lower rates of reproduction and signs of poor body condition, disease and malnutrition in the adult population [83]. Leading up to 2013, there were no reports of any increase in sightings of malnourished whales. Similarly, there was no spike in reported calf mortalities in the region, the reproductive rate appeared to be increasing and there were no reports of increased male aggression. If this population segment overshot carrying capacity prior to 2013, this could explain the steep decline in the reproductive rate that became apparent after 2013, but in this scenario too, signs of the population approaching carrying capacity should still have been evident prior to 2013.

Contrastingly, in recent months, the first reports of these indicator events and conditions have emerged in this subpopulation. On their feeding grounds, an increasing number of humpback whales exhibiting signs of malnutrition and high rates of parasitism have recently been sighted [75,84]. Reports from Alaska waters also indicate increasing numbers of whales persisting for longer periods on the feeding grounds prior to migration or returning earlier than usual to these areas; potentially, these whales are therefore engaged in searching for food over longer periods [84]. While clear evidence is admittedly scant at present, the low sighting rates for calves in Alaska waters even in comparison with the low encounter rates in Hawaiian waters could indicate increased calf mortality, and based on the observations of this study, as both mother–calf encounter rates and the percentage of mother-calf pairs sighted declined between 2013 and 2018, this suggests that the female reproductive rate for this DPS was in decline during this period.

All of these observations are described by Eberhardt & Siniff's [83] as declining condition indices. Observations of the indices can be used in a decision matrix compiled by Gerrodette & DeMaster [85]

to forecast and identify trends and changes in carrying capacity within a subpopulation, even when the actual carrying capacity has not been previously established, as is the case with this DPS. Within the decision matrix, the inferences of changes in condition indices specified by Eberhardt & Siniff [83] are combined with observed changes in abundance indices (suggested measures include numbers of offspring produced or encounter rates per kilometre). In this case, both the condition indices and abundance indices are in decline. Applying the decision matrix, this would indicate a lowering of the carrying capacity for this specific population segment (i.e. a reduction in the size of the population that can be supported within the current habitat used by the population segment [85]). Causes for such declines may vary widely. In some instances, changes can be specifically related to the availability of a single resource (e.g. [80]). In this case, given the increased reports of poor condition and malnutrition, nutritional stress emerges as a possible explanation for the changes documented within this DPS. Supporting this conclusion, the wide range of unusual mortality and mass casualty events in unrelated fauna reported across the Gulf of Alaska and connected marine systems has also been attributed to the reduced productivity and prey resources evident in this region [26,29,30].

## 4.3. Do fluctuations in prey resources align with recent changes?

Beginning in the winter of 2013, oceanic conditions within the North Pacific entered a period of unusual flux. Chronologically, the PDO began to shift in late 2013 and flipped to a positive phase in early 2014. During 2014, the TNI index, indicating the evolution of a Central Pacific El Niño event [55], reached a magnitude greater than 2, signalling the build-up of warm water in the Central North Pacific, and concurrently, the warm water anomaly in the North Pacific (known as 'the blob') spread into the coastal domains of Alaska and British Columbia [25]. In concert, all three of these anomalies would suppress coastal upwellings, draw down coastal productivity [28] and lead to pronounced horizontal advection of key prey species [25,26].

From a dietary perspective, humpback whales display a degree of plasticity that may potentially provide some resilience to large-scale oceanic shifts and associated changes in prey availability. In the California Current System, humpback whales are known to switch between a predominantly fish-based diet and a predominantly krill-based diet, as changes in the North Pacific Gyre Oscillation (NPGO) impact SSTs and alter upwellings that drive relative prey availability along the California–Oregon seaboard [86,87]. In Alaska waters, humpback whale prey preferences vary seasonally and regionally [84,88–91]. During the summer months in Southeast Alaska, mature euphausiids (*Thysanoessa* sp., *Euphausia pacifica*) are a preferred prey in some regions [88,92], while in other areas, such as Icy Strait and Glacier Bay, primary prey targets include small schooling fish such as capelin (*Mallotus villosus*), juvenile wall-eye pollock (*Gadus chalcogrammus*) and Pacific herring (*Clupea pallasii*). During fall, targeted prey vary regionally between herring and euphausiids. In winter, humpback whales in Southeast Alaska waters feed exclusively on herring [84] while in Southwest Alaska (Kodiak Island), local humpbacks also feed exclusively on fish, but will bypass juvenile pollock, *Gadus chalcogrammus*, in favour of schooling capelin, *Mallotus villosus* [91].

Although in theory this varied diet should provide some degree of resilience in Alaska humpbacks during periods when prey resources are disrupted, the climate anomalies that began in late 2013 have been causally linked to reduced availability across the range of these preferred prey. In coastal areas along the Gulf of Alaska and in Prince William Sound, increases in freshwater run-off have been linked to regional declines in herring [93]. In Southeast Alaska, the slowing of coastal upwellings led to changes in both quality and quantity of euphasiid prey [94], and in the open waters of the Gulf, horizontal advection disrupted regional assemblages of planktonic fauna. Specifically, smaller, southern species of copepods replaced the higher quality, larger species typically seen in the Gulf [22,95]. As copepods form the cornerstone of pelagic food webs in the Gulf, numerous higher trophic levels were impacted, including planktivorous fish populations [22,95].

In female mysticetes, as adequate energy stores are a requirement for both the initiation and successful maintenance of reproduction [33,35,37], reductions in the availability of preferred prey and associated nutritional stress may be mechanistically linked to a depression in reproductive rates. In the results obtained in this study, the performance of the GAM models suggested a degree of linkage between reproductive rates recorded in Hawaiian waters and conditions in the broader North Pacific marine system, as indicated by the oceanic indices incorporated into the analysis. Notably, the performance of the models improved when the oceanic indices were advanced by two seasons compared with a single season advance. This further supports the suggestion that the suppression of ovulation in response to reduced prey availability in the year prior to pregnancy may be the key

mechanism at play here. A suppression in ovulation rates would probably reduce the gamut of reproductive activities for female humpback whales, including the frequency of migration to the breeding grounds. Within breeding areas, females without young calves quickly leave the breeding grounds [96] and in the later portions of the season adult males typically seek mating opportunities with maternal females that may go into post-partum ovulation [97]. In periods of nutritional stress, it is easy to foresee that this would become a rare event. Reflecting these reduced mating opportunities, males would probably depart earlier for the feeding grounds, potentially accounting for the low numbers of adults seen in Maui waters during late seasons in recent years.

At this point, a final explanation for the fluctuating sighting rates for mother–calf humpback whale pairs in Hawaiian waters requires extensive further research, as many questions remain unresolved. As advocated by Doney et al. [2], a more definitive answer addressing the causes of the changes seen will require the availability of biological datasets that document changes in population dynamics and key life-history traits over the long-term time scale. Potentially, such details will facilitate a much clearer understanding of the complex interactions involved and allow for improved accuracy in predicting future fluctuations within this and other distinct population segments of humpback whales.

# 5. Conclusion

Beginning in the winter of 2014, the combination of recent climatic anomalies in the Eastern North Pacific, namely the regime shift of the PDO to a positive phase, the development and subsequent migration of the warm water anomaly known as the blob, and the evolution of a Central Pacific El Niño event combined to create oceanic conditions along the Alaska coastline that reduced productivity and impacted key humpback whale prey resources [17,22,28]. Highlighting the mechanistic links that exist between climate anomalies, physical oceanographic processes and biological resources, these climate anomalies currently provide a plausible explanation for the decline in reproductive rates within the Hawaii DPS documented in this study.

Understanding the interactions and connectivity between different climate anomalies is complex. Consequently, we would encourage experts in this field to review the data presented here, as this could provide a more nuanced analysis of the connections between fluctuating humpback whale reproductive rates within local population segments, regional prey availability and recent marine regime shifts. Future studies that document health indicators, such as body condition, especially in female humpback whales, will provide valuable insight into the resilience of specific population segments, such as the Hawaii DPS, as they negotiate the challenges presented as marine systems shift towards a new climate regime. A better understanding of anthropogenic energetic stressors, such as whale watching in both feeding and breeding regions, could elucidate means by which ameliorative management practices could help minimize maternal stress during this transition.

Humpback whales are widely recognized as a sentinel species [39,60]. Changes in a key life-history trait such as the reproductive rate provide an easily detected signal of other more subtle changes, such as alterations in prey quality, which in turn may ultimately carry much broader impact. Climatic models have confirmed that anthropogenic climate change exacerbated the formation of the recent warm water anomaly in the North Pacific and the associated warming of the Gulf of Alaska [29]. Events like this provide insight into the functioning and challenges of a warmer ocean and lessons learnt during events such as this may prove to be exceptionally valuable in the years that lie ahead.

In the meantime, the Hawaii DPS of humpback whales carries considerable ecosystem capital, not only in terms of ecosystem services [98,99], but also more directly, supporting extensive whale watching and associated tourism activities in both Alaska and Hawaii. For the purposes of down listing, this species was divided into separate stocks. NMFS established that the health of one stock should be considered independently from other groups [44]. Following this line of reasoning, as the changes described here meet the trigger requirements laid out in the post-delisting monitoring plan [100], we suggest that increased protection and a review of the status of this stock are clearly justified at this time.

Ethics. The study was conducted under scientific research permits nos. 10018 and 17845, granted by the National Marine Fisheries Service, Office of Protected Resources, Permits and Conservation Division and awarded to R. Cartwright. Annual Special Activity Permits to operate in Hawaii State waters were granted by the Hawaii Board of Land and Natural Resources. Full details of the precise research protocols used in this study were carefully reviewed by the Office of Protected Species, prior to issuance of the NMFS research permit to ensure that these methods comply with all ethical guidelines. Inherent in this review is the requirement that every effort be made to minimize any

impact on animals during research activities. As this detailed and extensive review had been conducted by experts in this field, further ethical review by the cooperating institution, California State University Channel Islands was not required. All research protocols additionally comply with the Endangered Species Act (1973) and the Marine Mammal Protection Act (1972) (https://apps.nmfs.noaa.gov/ docs_cfm/laws_and_regulations.cfm).

Data accessibility. The datasets supporting this study are provided in the electronic supplementary materials, accompanying this publication.

Authors' contributions. R.C. and A.V. designed the project, R.C., A.V., V.H., C.W., J.C. and D.C conducted extensive fieldwork, led student teams in the field and participated in data analysis. R.C., C.W. and A.V interpreted the data, R.C. wrote the manuscript with assistance from C.W. All authors approved the final version prior to publication.

Competing interests. The authors declare no competing interests.

Funding. This project was principally funded through annual donations to the Keiki Kohola Project, received from Mark Percival, Worcester UK. Donations from the Instructionally Related Activities fund at California State University Channel Islands supported student participation in the project. Cesere Fine Art Photography provided additional financial support; funds were also received from the British Broadcasting Corporation, Apple Computers Inc. and Whale Trust, Maui.

Acknowledgements. This project could not have been completed without the generous support and endless encouragement provided by Mark Percival, Worcester UK. Support during fieldwork was provided by Terence Mangold, Rob Hawes, Sarah Scrivano, Michaela Miller, Mark Danielson, Sergio Burgos, Rob Schneider, Tad Chamberlin, Nicola Dempster and Charlotte Dempster. We are grateful to these and the many other volunteers who have assisted the Keiki Kohola project over the years. Logistic support provided by Lee James and the captains and crew of Ultimate Whalewatch, Maui, HI also made a very valuable contribution to the completion of this work. Additionally, we sincerely appreciate the keen and enthusiastic participation of the faculty and students of California State University Channel Islands. Blake Gillespie provided extensive support during the initial implementation of this study, and throughout the intervening years the university has supported the continued involvement of students and faculty in this project. Finally, thanks go to Joe Mobley and Jan Straley for input and feedback provided during reviews of a previous version of this manuscript.

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
