## [Reviewer comments · Royal Society Open Science]

Review History

RSOS-181463.R0 (Original submission)

Review form: Reviewer 1 (Joe Mobley)

Is the manuscript scientifically sound in its present form?

Yes

Are the interpretations and conclusions justified by the results?

Yes

Is the language acceptable?

Yes

Is it clear how to access all supporting data?

Yes

Do you have any ethical concerns with this paper?

No

Have you any concerns about statistical analyses in this paper?

No

Recommendation?

Accept with minor revision (please list in comments)

Comments to the Author(s)

Comments to authors are attached separately (Appendix A).

Review form: Reviewer 2 (Jan Straley)

Is the manuscript scientifically sound in its present form?

Yes

Are the interpretations and conclusions justified by the results?

Yes

Is the language acceptable?

Yes

Is it clear how to access all supporting data?

Yes

Do you have any ethical concerns with this paper?

No

Have you any concerns about statistical analyses in this paper?

No

Recommendation?

Accept with minor revision (please list in comments)

Comments to the Author(s)

Dear Dr. Cartwright and coauthors,

You have written a fairly clear and concise paper on a timely topic. It needs some clarification; hence, I have recommended minor revisions and I encourage the journal to publish quickly.

A few line by line suggestions:

Page 4 Line 48 Describe what you mean by 'carrying capacity', because this term is hard to define and varies in the literature. Do you mean the population has surpassed 'K'? If so, describe what 'K is for this DPS. Or do you mean the ecosystem has shifted so fewer whales can be sustained?

Page 5 Lines 20-37 I suggest using an effort table to include years, 'season', days on the water of Beaufort 2 or less and nm surveyed. A table would be easier for the reader to see where the data gaps exist, if any.

Page 6 Line 3 Were the observers the same? Was there observer experienced? Convince the reader the data were collected by trained experienced observers and not an intern from Kansas.

Line 17 Describe what you mean by escorts.

Line 20 Were iPads used to record GPS in all study years?

Page 7 Line 19 explain ESRI

Page 8 Statistical Analysis and Environmental Parameters

How did you deal with the correlation among oceanographic events? Or were they considered as independent events? I suggest you seek the advice of an oceanographer who may have insights you have not considered.

Encounter rates

Page 9 Line 37 Results

As a reader I would like to see a comparison of effort between the two time periods. If it is presented anywhere it is not obvious. This will inform the reader if there is validity in documenting a downward trend if the type of effort is balanced among the early and later years. The number of days and nm transected would be easy to add to the other tables if Being a rate it is the number of groups with calves over some unit but that unit is unclear.

In the following sentence, the reader has no idea if the drop in encounter rates was due to effort In days surveyed or nm transected.

'Of these, 149 groups were encountered in 2013-2014 and 65 groups were encountered in 2017-2018, reflecting the changes in encounter rates between these time periods.'

Page 11 Line 6 This section made me dizzy. Could you present the two groups of years and then the individual years in a table? Lots of information but a bit confusing to follow. In thinking about how to make this easier to follow are the early, mid and late divisions really necessary? Could you pick the timing that best represents the years. While I don't mind the three seasons and find the differences interesting it is hard to follow. I suggest present the results as an overall grouping between the two sets of years and then add the necessary seasons to support your findings.

Page 12 Line 25 In the methods this is called Environmental Parameters....be consistent in the headers and layout between Methods and Results

Page 13

Line 49 Encounter rates of calves? M/C? Adults?

Line 52ish How does the timing work for the nutritional stress hypothesis? If declines were before 2015 could this indicate the warm water events impacted whales earlier?

Page 14 Line 42 Correction Gabriele not Gabrielle. Double check HMMC numbers from their 2018 season report.

Page 15 Line 45 GulfWatch Alaska has studied PWS humpback and herring since 2006. Results through 2009 are reported in a special issue from Deep Sea Research 2 Other surveys are reported in survey reports available from the AOOS website portal. This might be a good time to discuss the year prior to pregnancy as a year a female needs to be in top fitness. If warm water events first began impacting GOA productivity in 2013 the subsequent poor calving years should

have started to be seen in HI in 2015 and beyond. This is because poor body condition from nutritional stress on the feeding grounds in 2013 would have made 2014 a poor year to sustain a pregnancy in HI and AK and the resulting poor calving year would be first documented in HI the winter of 2015.

Page 16 Line 12ish Again think of the vagueness of the carrying capacity concept and explain your thinking in saying they really reached some point that caused a rapid decline in calves.

Page 17

Line 20 Krill is not the primary prey in SEAK

Line 45 Again consult with an oceanographer to fully understand how the three oceanographic events interplay among each other.

Page 18

Line 28 This is speculative and not based on actual data. Females MAY overwinter, a few have, in Alaska, but it is more likely they spend little time on the breeding grounds (or only go part way) and more time feeding because the migration is staggered hence giving the appearance of whales staying all winter in Alaska.

Line 42 ish Again carrying capacity needs to be clear as to what you mean by this term.

References

Straley, J.M., Deep-Sea Research Part II (2017), <http://dx.doi.org/10.1016/j.dsr2.2017.08.008>

Decision letter (RSOS-181463.R0)

17-Dec-2018

Dear Dr Cartwright,

The editors assigned to your paper ("Declining reproductive rates in Hawaii's humpback whales, *Megaptera novaeangliae*, in response to recent climate anomalies in the North Pacific") have now received comments from reviewers. We would like you to revise your paper in accordance with the referee and Associate Editor suggestions which can be found below (not including confidential reports to the Editor). Please note this decision does not guarantee eventual acceptance.

Please submit a copy of your revised paper before 09-Jan-2019. Please note that the revision deadline will expire at 00.00am on this date. If we do not hear from you within this time then it will be assumed that the paper has been withdrawn. In exceptional circumstances, extensions may be possible if agreed with the Editorial Office in advance. We do not allow multiple rounds of revision so we urge you to make every effort to fully address all of the comments at this stage. If deemed necessary by the Editors, your manuscript will be sent back to one or more of the original reviewers for assessment. If the original reviewers are not available, we may invite new reviewers.

To revise your manuscript, log into <http://mc.manuscriptcentral.com/rsos> and enter your Author Centre, where you will find your manuscript title listed under "Manuscripts with Decisions." Under "Actions," click on "Create a Revision." Your manuscript number has been

appended to denote a revision. Revise your manuscript and upload a new version through your Author Centre.

- Data accessibility

If you wish to submit your supporting data or code to Dryad (<http://datadryad.org/>), or modify your current submission to dryad, please use the following link:
<http://datadryad.org/submit?journalID=RSOS&manu=RSOS-181463>

- Competing interests

- Authors' contributions

- Acknowledgements

- Funding statement

on behalf of Dr Denise Greig (Associate Editor) and Kevin Padian (Subject Editor)
 openscience@royalsociety.org

Associate Editor's comments (Dr Denise Greig):

This is a nice dataset surveying whale presence on a nursery ground in the Hawaii from 2008 to 2018 and the impact of climate anomalies on whale numbers. The paper is timely given the basin wide impact of these oceanographic anomalies on multiple taxa.

Two supportive reviewers categorized their comments as minor revisions, but I think some of their requests are fairly extensive so I am calling the revisions major, recognizing that they will take some effort on your part to address. In additions to their comments, I just have a few minor questions

Page 2, line 15. I wonder if you mean “signal” rather than “sentinel”

Page 3, line 40. I wonder why you chose “adiposity” as opposed to “body condition”. Adiposity implies obesity in humans – I have not seen it defined in reference to whales.

Page 5, line 25. What is the time persiod for “the entire season”?

Page 5, lines 42-49. This was confusing to read with distance measured in degrees, minutes, and meters – can it be converted to meters and kilometers with the degrees and minutes in parentheses?

Page 6, line 24. What is $g(0)$? Detection probability? Or percent time on the surface?

Page 19, line 18, what is a teleconnection?

Tables. Please remove the vertical lines (and any of the horizontal lines that are not needed).

Tables 4 and 6, please describe the explanatory variables in the table title or footnote even if they are descibed in the text (i.e. what is $s(\text{year}, 2.34)$?, SST, PDO...)

Comments to Author:

Reviewers' Comments to Author:

Reviewer: 1

Comments to the Author(s)

Comments to authors are attached separately

Reviewer: 2

Comments to the Author(s)

Dear Dr. Cartwright and coauthors,

You have written a fairly clear and concise paper on a timely topic. It needs some clarification; hence, I have recommended minor revisions and I encourage the journal to publish quickly.

A few line by line suggestions:

Page 4 Line 48 Describe what you mean by 'carrying capacity', because this term is hard to define and varies in the literature. Do you mean the population has surpassed 'K'? If so, describe what 'K is for this DPS. Or do you mean the ecosystem has shifted so fewer whales can be sustained?

Page 5 Lines 20-37 I suggest using an effort table to include years, 'season', days on the water of Beaufort 2 or less and nm surveyed. A table would be easier for the reader to see where the data gaps exist, if any.

Page 6 Line 3 Were the observers the same? Was there observer experienced? Convince the reader the data were collected by trained experienced observers and not an intern from Kansas.

Line 17 Describe what you mean by escorts.

Line 20 Were iPads used to record GPS in all study years?

Page 7 Line 19 explain ESRI

Page 8 Statistical Analysis and Environmental Parameters

How did you deal with the correlation among oceanographic events? Or were they considered as independent events? I suggest you seek the advice of an oceanographer who may have insights you have not considered.

Encounter rates

Page 9 Line 37 Results

As a reader I would like to see a comparison of effort between the two time periods. If it is presented anywhere it is not obvious. This will inform the reader if there is validity in documenting a downward trend if the type of effort is balanced among the early and later years. The number of days and nm transected would be easy to add to the other tables if Being a rate it is the number of groups with calves over some unit but that unit is unclear. In the following sentence, the reader has no idea if the drop in encounter rates was due to effort In days surveyed or nm transected.
'Of these, 149 groups were encountered in 2013-2014 and 65 groups were encountered in 2017-2018, reflecting the changes in encounter rates between these time periods.'

Page 11 Line 6 This section made me dizzy. Could you present the two groups of years and then the individual years in a table? Lots of information but a bit confusing to follow. In thinking about how to make this easier to follow are the early, mid and late divisions really necessary? Could you pick the timing that best represents the years. While I don't mind the three seasons and find the differences interesting it is hard to follow. I suggest present the results as an overall grouping between the two sets of years and then add the necessary seasons to support your findings.

Page 12 Line 25 In the methods this is called Environmental Parameters....be consistent in the headers and layout between Methods and Results

Page 13

Line 49 Encounter rates of calves? M/C? Adults?

Line 52ish How does the timing work for the nutritional stress hypothesis? If declines were before 2015 could this indicate the warm water events impacted whales earlier?

Page 14 Line 42 Correction Gabriele not Gabrielle. Double check HMMC numbers from their 2018 season report.

Page 15 Line 45 GulfWatch Alaska has studied PWS humpback and herring since 2006. Results through 2009 are reported in a special issue from Deep Sea Research 2 Other surveys are reported in survey reports available from the AOOS website portal. This might be a good time to discuss the year prior to pregnancy as a year a female needs to be in top fitness. If warm water events first began impacting GOA productivity in 2013 the subsequent poor calving years should have started to be seen in HI in 2015 and beyond. This is because poor body condition from nutritional stress on the feeding grounds in 2013 would have made 2014 a poor year to sustain a pregnancy in HI and AK and the resulting poor calving year would be first documented in HI the winter of 2015.

Page 16 Line 12ish Again think of the vagueness of the carrying capacity concept and explain your thinking in saying they really reached some point that caused a rapid decline in calves.

Page 17

Line 20 Krill is not the primary prey in SEAK

Line 45 Again consult with an oceanographer to fully understand how the three oceanographic events interplay among each other.

Page 18

Line 28 This is speculative and not based on actual data. Females MAY overwinter, a few have, in Alaska, but it is more likely they spend little time on the breeding grounds (or only go part way) and more time feeding because the migration is staggered hence giving the appearance of whales staying all winter in Alaska.

Line 42 ish Again carrying capacity needs to be clear as to what you mean by this term.

References

Straley, J.M., Deep-Sea Research Part II (2017), <http://dx.doi.org/10.1016/j.dsr2.2017.08>.

Author's Response to Decision Letter for (RSOS-181463.R0)

See Appendices B & C.

RSOS-181463.R1 (Revision)

Review form: Reviewer 1 (Joe Mobley)

Is the manuscript scientifically sound in its present form?

Yes

Are the interpretations and conclusions justified by the results?

Yes

Is the language acceptable?

Yes

Is it clear how to access all supporting data?

Yes

Do you have any ethical concerns with this paper?

No

Have you any concerns about statistical analyses in this paper?

No

Recommendation?

Accept as is

Comments to the Author(s)

I've reviewed the revised manuscript in light of the previous reviewer comments and the authors' response. This is a much improved manuscript that is worthy of publication nearly as is. The only thing in need of revision is a typo on lines 129 and 133, should be "mid-March" instead of "mi-March."

Review form: Reviewer 2 (Jan Straley)

Is the manuscript scientifically sound in its present form?

Yes

Are the interpretations and conclusions justified by the results?

Yes

Is the language acceptable?

Yes

Is it clear how to access all supporting data?

Yes

Do you have any ethical concerns with this paper?

No

Have you any concerns about statistical analyses in this paper?

No

Recommendation?

Accept as is

Comments to the Author(s)

Dr.Cartwright and co authors,

You addressed all reviews concerns and the manuscript improved immensely. Great work. I recommend publication as is.

Decision letter (RSOS-181463.R1)

14-Feb-2019

Dear Dr Cartwright:

On behalf of the Editors, I am pleased to inform you that your Manuscript RSOS-181463.R1 entitled "Declining reproductive rates in Hawaii's humpback whales, *Megaptera novaeangliae*, in response to recent climate anomalies in the North Pacific" has been accepted for publication in Royal Society Open Science subject to minor revision in accordance with the referee suggestions. Please find the referees' comments at the end of this email.

The reviewers and Subject Editor have recommended publication, but also suggest some minor revisions to your manuscript. Therefore, I invite you to respond to the comments and revise your manuscript.

- Ethics statement

- Data accessibility

If you wish to submit your supporting data or code to Dryad (<http://datadryad.org/>), or modify your current submission to dryad, please use the following link:
<http://datadryad.org/submit?journalID=RSOS&manu=RSOS-181463.R1>

- Competing interests

- Authors' contributions

- Acknowledgements

- Funding statement

Because the schedule for publication is very tight, it is a condition of publication that you submit the revised version of your manuscript before 23-Feb-2019. Please note that the revision deadline will expire at 00.00am on this date. If you do not think you will be able to meet this date please let me know immediately.

- 1) A text file of the manuscript (tex, txt, rtf, docx or doc), references, tables (including captions) and figure captions. Do not upload a PDF as your "Main Document".
- 2) A separate electronic file of each figure (EPS or print-quality PDF preferred (either format should be produced directly from original creation package), or original software format)

- 3) Included a 100 word media summary of your paper when requested at submission. Please ensure you have entered correct contact details (email, institution and telephone) in your user account
- 4) Included the raw data to support the claims made in your paper. You can either include your data as electronic supplementary material or upload to a repository and include the relevant doi within your manuscript
- 5) All supplementary materials accompanying an accepted article will be treated as in their final form. Note that the Royal Society will neither edit nor typeset supplementary material and it will be hosted as provided. Please ensure that the supplementary material includes the paper details where possible (authors, article title, journal name).

on behalf of Dr Denise Greig (Associate Editor) and Kevin Padian (Subject Editor)
openscience@royalsociety.org

Associate Editor Comments to Author (Dr Denise Greig):

Really well done response to reviewers! Thank you for all the information.

I would change this sentence in the abstract to keep all the verbs in agreement

From "Rates initially increased, tracking projected growth rates for this population segment, reaching a peak in 2013, and then declining through 2018."

To "Rates initially increased, tracking projected growth rates for this population segment, reached a peak in 2013, and then declined through 2018."

P 5, line 125. Thank you for letting me know the timing of a normal season in your response, but I think you need to add it to the text, maybe after this sentence, "The study spans the January through March season from 2008 to 2018, however timing..."

Line 129. Survey protocols used in 2008 to 2010 were repeated in 2013 and 2014 to allow comparisons of the results.

Line 132. In 2017 and 2018, surveys were once again conducted during three two week periods, in mid-January, mid-February and mi-March, again using the 2008-2010 survey protocols.

Line 133. All surveys were restricted to favorable sea conditions (Beaufort <3) in order to ensure consistency in weather / sightability conditions. Shouldn't this be <4 or less than or equal to 3? I understood from your response that surveys were conducted in Beaufort = 3.

Line 305. In a second set of models, the impact of key oceanic parameters (impact on what? On variation in whale encounter rates?) was explored. Encounter rates for individual whales, mother-calf groups and adult only groups comprised the response variables, with separate models constructed for each variable.

Reviewer comments to Author:

Reviewer: 1

Comments to the Author(s)

I've reviewed the revised manuscript in light of the previous reviewer comments and the authors' response. This is a much improved manuscript that is worthy of publication nearly as is. The only thing in need of revision is a typo on lines 129 and 133, should be "mid-March" instead of "mi-March."

Reviewer: 2

Comments to the Author(s)

Dr. Cartwright and co authors,

You addressed all reviews concerns and the manuscript improved immensely. Great work. I recommend publication as is.

Author's Response to Decision Letter for (RSOS-181463.R1)

See Appendix D.

Decision letter (RSOS-181463.R2)

20-Feb-2019

Dear Dr Cartwright,

I am pleased to inform you that your manuscript entitled "Fluctuating reproductive rates in Hawaii's humpback whales, *Megaptera novaeangliae*, reflect recent climate anomalies in the North Pacific" is now accepted for publication in Royal Society Open Science.

Royal Society Open Science operates under a continuous publication model

(<http://bit.ly/cpFAQ>). Your article will be published straight into the next open issue and this will be the final version of the paper. As such, it can be cited immediately by other researchers. As the issue version of your paper will be the only version to be published I would advise you to check your proofs thoroughly as changes cannot be made once the paper is published.

on behalf of Dr Denise Greig (Associate Editor) and Kevin Padian (Subject Editor)
openscience@royalsociety.org

Appendix A

Review of Cartwright et al.: “Declining reproductive rates in Hawaii’s humpback whales, *Megaptera novaeangliae*, in response to recent climate anomalies in the North Pacific” (manuscript ID: RSOS-181463)

Reviewer: Joe Mobley, PhD

This is an important and timely contribution both at a regional level showing compelling evidence of a potential downturn in reproductive rates of humpback whales in the Hawaii wintering region, as well as the global level by associating such changes in response to broad-level climate change. That being said, there were a number of small issues, none of them lethal, that if left uncorrected might detract from the important story the results describe.

The larger of the small issues was that of variable effort within and across years; e.g., seemingly random repetitions of transect lines; post-hoc insertions of additional lines for no apparent useful purpose. For readers like myself, who are familiar with distance sampling theory, such details snag one’s attention as red flags. However, there’s little doubt in my mind that the strong downward trend they describe is true; my concern is that these anomalies needlessly draw attention from that point. My specific comments follow:

p. 4 line 42-43: “Overall, we found very little evidence of any changes in patterns of habitat use.”

Comment: This is too sweeping a conclusion for the small modification made in 2018 (see related comments below). I recommend that any claims of “no change in patterns of habitat use” be dropped.

p. 5 lines 21-22 – “The study spans the years 2008 to 2018. In 2008 to 2010, all surveys were conducted in the latter portion of the season (mid-March onwards).”

Comment: A reference to Table 1 would be helpful here given the complexity of the timing of surveys across years.

p. 5 lines 32-33: "In 2018, additional surveys were added in mid-channel waters to capture any local relocation."

Comment: The utility of adding these lines escaped me. They added so little in the range of effort and involved so few additional sightings that I feel the manuscript would be better off not including them at all. Certainly the additional sightings they yielded were not enough to make any statistically meaningful comparisons. My recommendation is to drop these data and focus on the main results.

p. 5 lines 34 &35 "restricting to BF 2 or better"

Comment: As someone who works frequently in the same region (Auau Channel) I have to question the veracity of this claim. The presence of any whitecaps at all quickly changes the Beaufort sea state from 2 to 3 and, in my experience, this happens frequently.

p. 5 lines 54-57: "Each survey line was covered at least once but no more than twice within each set of surveys and surveys were continued until either approximately 50 km had been covered, or the locations >50 groups had been recorded."

Comment: This detail bothered me most of all. I'm used to following clear rules in the placement and execution of track lines and this makes it sound almost whimsical. Whatever the authors can do to reveal any explicit rules used in replicating lines would be helpful. Also, they need to clearly address the issue of replicated sightings and their treatment of these (see below) since replicating lines would obviously increase the probability of doing so.

p. 6 line 24: $g(o)$ should be $g(0)$, i.e., a zero not the letter o. Also their reference to Mobley, Bauer & Herman—(submerged vs surface times) is incorrect; should be Mobley (2001) sanctuary report.

p. 6 line 31-3: "Fluke photo ID's (as per [53]) and surface images documenting body markings, lesions and other scars were compiled for all sighted animals and used post-hoc to eliminate any chance of pseudo replication over the course of the day, between survey vessels or within regions of over-lap at the beginning or the end of any successive transects."

Comment: Need more detail about this correction for resights. What percentage of sightings were resights, and how were they treated; i.e., were they simply eliminated from the ER estimates or something else?

p. 7 line 20-21: "As sightings within an estimated 1 km had originally been recorded, this also reduced any potential edge effect."

Comment: What is an "edge effect?"

p. 12 line 10: "Mid-channel surveys (2018 season only) covered a total distance of 38.4 km"

Comment: This short distance just reinforces my comment for p. 5 lines 32-33 noted above.

p. 12 lines 32-33: "using the onset of climatic anomalies as a splitting point"

Comment: What specific anomalies are being referred to here?

p. 13 line 14-15: "For encounter rates for individual whales..."

Comment: Does "individual whales" include all whales? Or does it exclude cows with calves, the comparison group in this case? Please clarify.

p. 14 lines 51-54: "Surveys conducted in mid channel waters in 2018 when encounter rates in the main study are were exceptionally low did not provide any evidence of higher than expected numbers of cow-calf pairs in that adjacent area"

Comment: As noted above, the small numbers of animals seen in this one-season modification were not sufficient to support a "no-change" inference.

p. 15 line 42: "SPLISH" should be SPLASH

p. 16 lines 25-28: "In this case, changes were asymmetrical; the rate of decline fluctuated, and ultimately far exceeded the rate of increase prior to carrying capacity (K)."

Comment: This sentence implies that carrying capacity (K) has been established for the present DPS when such is not the case. Please modify the sentence to better clarify that point.

p. 17 lines 19-22: In the California Current System, humpback whales will switch between a predominantly fish based diet and a predominantly krill based diet..."

Comment: "Will" implies that we're seeing into the future here? Maybe change "will" to "are known to."

Overall, I feel that these are important results that deserve to be broadly shared. Once these requested changes are addressed, then I recommend that the revised manuscript be accepted for publication.

Mahalo,

Joe Mobley

Appendix B

Changes made:

1. In response to notes from the Associate editor:

Page 2, line 15. I wonder if you mean “signal” rather than “sentinel”

- Suggested word change clarifies the meaning of this sentence. Sentinel changed to signal

Page 3, line 40. I wonder why you chose “adiposity” as opposed to “body condition”. Adiposity implies obesity in humans – I have not seen it defined in reference to whales.

- I found this term used in several of the physiology papers referenced. In general, authors were referring to the process of adaptive obesity (see Nash et al. 2017). This is seen in humpback whales, as they consume high volumes of prey during the foraging season and lay down energy reserves for use during migration and breeding when they will be fasting. Effectively, like many hibernators, they are taking in more calories than expended and from an adaptive perspective, the larger these reserves are the better the animal will fare. The current literature suggests that female ovulation may be reliant on these fat reserves, and throughout the reproductive cycle, bigger females do better, raise larger calves, even swimming and mobility is less costly. Notably, females lose around 30-35% of their body mass over the course of gestation and lactation in the breeding areas, so certainly, there are rapid swings in body condition among these females. Also, studies have shown that it is an increase in the size of adipocyte cells, and not adipogenesis that occurs. All this being said, I think that adiposity might be an applicable term in line 40. However, in order to ensure that the text is accessible, I have changed this term to stored energy reserves and I have reviewed subsequent uses. In places where the inferred meaning is that the whales increase in size, but not to the point of obesity, the more general term body condition has been used. Additionally, the term adipose stores has been changed to energy reserves in subsequent comments on lines 29 to 33, as this then clarifies the meaning of these statements. (Apologies for the long answer, our current studies focus on body condition in females, so I might be overly focused on this issue right now).

Page 5, line 25. What is the time period for “the entire season”?

- Clarification added – the season runs from January through March, with mother-calf pairs seen generally from mid-January through to late March. Additional text has been added to clarify this.

Page 5, lines 42-49. This was confusing to read with distance measured in degrees, minutes, and meters – can it be converted to meters and kilometers with the degrees and minutes in parentheses?

- It’s a little tricky to do this as the length of 1 minute of latitude is more of a map reference than a distance per-say. A minute of latitude at this location is approx. 0.9 km – though this varies with latitude. In order to clarify how this was set up though I have added a note to see the figure that illustrates this and included details on the overall coverage of the study area. I’ve also added

additional details in the study site section to further explain the location and size of the study area.

Page 6, line 24. What is g(0)? Detection probability? Or percent time on the surface?

- g (0) is detection probability, I've added the explanation of this abbreviation in the lines above where the term is used for the first time.

Page 19, line 18, what is a teleconnection?

- This term is widely used in oceanographic literature – I had not met the term before though. Essentially it refers to connections between meteorological or other environmental phenomena and its used in situations where two distant components are linked. Sometimes it may be causal, or it may be a correlation too. I replaced this term with a new word “connectivity” as shown below to clarify the meaning here.

“Understanding the interactions and connectivity between different climate anomalies is certainly complex”

Tables. Please remove the vertical lines (and any of the horizontal lines that are not needed).

- Tables have been corrected - vertical and unnecessary horizontal lines have been removed

Tables 4 and 6, please describe the explanatory variables in the table title or footnote even if they are described in the text (i.e. what is s(year, 2.34)?, SST, PDO...)

- Full explanations of abbreviations have been added to the table titles and footnotes

2. Responses to reviewer #2

Reviewer: Joe Mobley, PhD

In response to this comment

“The larger of the small issues of concern was that of variable effort within and across years; e.g., seemingly random repetitions of transect lines; post-hoc insertions of additional lines for no apparent useful purpose. For readers like myself, who are familiar with distance sampling theory, such details snag one’s attention as red flags. However, there’s little doubt in my mind that the strong downward trend they describe is true; my concern is that these anomalies needlessly draw attention from that point”.

Addressing the reviewer's comments from a broader perspective, the point made here is well-taken. This study assembles details from a range of surveys that document encounter rates within a previously favored habitat over an extended time-period. Cumulatively, these surveys certainly do not meet the requirements necessary to determine actual abundance estimates. Rather, as advocated by Ducklow et al 2009, the data set presented here was assembled to provide a comparative indication of changes and trends in encounter rates within the confines of this study area. As such, this study is presented here as a first step, offering an up to date indication that notable changes in abundance and key population parameters may be on-going in this region. When the study was originally envisaged in 2008, design details were chosen with reference to the work of Strindberg and Buckland. In their methodology, single transect lines represent independent samples that infer distribution within a region in proportion to the habitat available. This method specifically takes into account issues such as gradients in abundance in relation to key habitat features such as coastlines or underwater topography, ultimately providing encounter rates that reflect distribution gradients and provide a mean encounter rate for the study area. During this study, as changes in encounter rates became apparent, the goal at this stage was to keep the methods consistent and provide reliable details that may indicate on-going changes and trends in encounter rates in this region. Hopefully, in so doing, this study will serve as a starting place and can help justify more detailed assessments of the precise nature of any changes in this region. As mentioned, comments added and detailed below will hopefully highlight this goal and explain the constraints and limitations of the current study to the reader.

Additionally, comments within the manuscript that address habitat use have been clearly amended to show that these statements only relate to fine scale patterns within the study area. As suggested, the limited, single year mid-channel surveys have been removed from the main body of the manuscript and comments have been limited to a short note referencing the fact that some effort was made in this region was added to the discussion. The table is now located in the supplemental tables and clearly described as an ad-hoc set of observations. However, as these observations may help to highlight and direct the need for continuing research, we believe that these details should remain in the manuscript in this limited form at this stage.

Line by line responses are as follows:

p. 4 line 42-43: "Overall, we found very little evidence of any changes in patterns of habitat use."

Reviewer comment: This is too sweeping a conclusion for the small modification made in 2018 (see related comments below). I recommend that any claims of “no change in patterns of habitat use” be dropped.

- The suggested change has been made and the comments regarding changes in habitat use has been edited throughout the text to reflect the fact that this data set is confined to a small study area along the Maui shoreline. This data set only reflects habitat uses within this specific area, as the reviewer points out, this dataset does not address the possibility of changes in habitat use within the region. At this time, this issue of changing habitat use remains unresolved and a comment to this effect has been included on page 17 line 8.

p. 5 lines 21-22 – “The study spans the years 2008 to 2018. In 2008 to 2010, all surveys were conducted in the latter portion of the season (mid-March onwards).”

Comment: A reference to Table 1 would be helpful here given the complexity of the timing of surveys across years.

Reference added as suggested

p. 5 lines 32-33: “In 2018, additional surveys were added in mid-channel waters to capture any local relocation.”

Comment: The utility of adding these lines escaped me. They added so little in the range of effort and involved so few additional sightings that I feel the manuscript would be better off not including them at all. Certainly the additional sightings they yielded were not enough to make any statistically meaningful comparisons. My recommendation is to drop these data and focus on the main results.

Details and results of the mid-channel surveys have been removed throughout the manuscript. As explained in the cover letter, they are now included as ad-hoc observations and provided in the supplemental materials merely to serve as reference point for future studies.

p. 5 lines 34 &35 “restricting to BF 2 or better”

Comment: As someone who works frequently in the same region (Auau Channel) I have to question the veracity of this claim. The presence of any whitecaps at all quickly changes the Beaufort sea state from 2 to 3 and, in my experience, this happens frequently.

- The upper limit of the Beaufort reference has been changed from 2 to 3. One of the components of the design of this study, was that we chose a study area that typically fell within the lee of the West Maui mountains. Within this study area, we therefore had sufficient leeway to choose transects and survey routes on a daily basis that offered the best prevailing weather conditions. As the reviewer points out, this is crucial to ensure consistent sightability. If we were on the water and the weather changed we halted the survey, noted the end point and resumed that transect on a different day when the weather was sufficiently calm to ensure consistent sightability. However after careful review of the precise details of the Beaufort, I would agree that it is more accurate to state that we halted the surveys if the winds exceeded Beaufort 3.

p. 5 lines 54-57: "Each survey line was covered at least once but no more than twice within each set of surveys and surveys were continued until either approximately 50 km had been covered, or the locations >50 groups had been recorded."

Comment: This detail bothered me most of all. I'm used to following clear rules in the placement and execution of track lines and this makes it sound almost whimsical. Whatever the authors can do to reveal any explicit rules used in replicating lines would be helpful. Also, they need to clearly address the issue of replicated sightings and their treatment of these (see below) since replicating lines would obviously increase the probability of doing so.

- This response addresses the survey design and methods comment specifically – repeat sightings is addressed below.

As the reviewer points out, the protocols used here don't meet the stringent requirements for distance sampling and accurate abundance estimates. Rather, the data set presented here is a cumulative record of encounter rates in a specific region, potentially indicating changes in relative abundance. As outlined in the new text, the goals of the three different time periods vary to some degree. In 2008, when we began assessing habitat use in this region, our original survey design followed protocols described by Buckland 2001 and Strindberg and Buckland 2004. In 2013-2014, we refined the study in order to detect seasonal patterns of habitat use. In the final period, rather than adopt new sampling protocols, we aimed to conduct surveys that would be comparable to previous data sets.

The goal of 50 groups was set up to provide a sample size that would ensure adequate power in the analysis, though, as numbers have fallen, we haven't reached this upper limit since 2013. In the 2013-

2014 period of data collection, we set this as an upper limit as we wanted to include data from two seasons. The goal of the study then was to determine habitat use and the sample size was based on a power analysis. In 2013, we did not envisage the decline in numbers seen.

With regards to replication of specific tracks, in 2013-14 we conducted three replicate surveys, in early, mid and late season, to look for seasonal trends in distribution. In 2017-18, we wanted to repeat the 2013 protocol as far as possible so that the two data sets could be compared. Although the transect comprises a single route, the legs of the transects within the zig-zag pattern were regarded as independent. Strindberg and Buckland 2004 do suggest repeating these legs to maximize coverage within a specific region and essentially, they state that each is an independent sample that provides encounter rates relative to the habitat that falls within the study area. With the construction described here, any density gradient in the study area is taken into account and ultimately the data provide mean encounter rates for the specific habitat included in the study area.

As a result of this, the current certainly is constrained to providing quantitative details of changes in relative encounter rates in this region. The study does not lend itself to any estimate of overall abundance as could be derived from more complex distance sampling protocols. This limitation is now clearly mentioned in the text.

p. 6 line 24: $g(o)$ should be $g(0)$, i.e., a zero not the letter o. Also their reference to Mobley, Bauer & Herman—(submerged vs surface times) is incorrect; should be Mobley (2001) sanctuary report.

- these changes have been made

p. 6 line 31-3: "Fluke photo ID's (as per [53]) and surface images documenting body markings, lesions and other scars were compiled for all sighted animals and used post-hoc to eliminate any chance of pseudo replication over the course of the day, between survey vessels or within regions of over-lap at the beginning or the end of any successive transects."

Comment: Need more detail about this correction for resights. What percentage of sightings were resights, and how were they treated; i.e., were they simply eliminated from the ER estimates or something else?

- As the estimates here were daily encounter rates, rather than abundance rates, searches were made only for within-day replicates. These were frequently detected in the field and when detected there, the whale(s) were only counted on the first encounter. Additional text has been added to clarify this.

p. 7 line 20-21: “As sightings within an estimated 1 km had originally been recorded, this also reduced any potential edge effect.”

Comment: What is an “edge effect?”

-This term, as used by Strindberg and Buckland, refers to the heterogeneity in sightings across the width of a transect – it’s the idea that there could be some fall-off in sightability towards the edge of the strip width. An extra comment has been added to clarify this.

As sightings within an estimated 1 km had originally been recorded, this also reduced any potential edge effect, ensuring that all sightings across the width of the transect strip were captured.

p. 12 line 10: “Mid-channel surveys (2018 season only) covered a total distance of 38.4 km” Comment: This short distance just reinforces my comment for p. 5 lines 32-33 noted above.

- Point taken, these surveys have been removed, save for a short anecdotal comment regarding ad-hoc observations in the discussion. Table 5 has also been relocated to the supplemental section.

p. 12 lines 32-33: “using the onset of climatic anomalies as a splitting point” Comment: What specific anomalies are being referred to here?

- I was referring principally to the flip in the PDO, however to clarify this, I have re-stated this split point as years of increasing numbers to the peak of sightings vs years after the peak encounter rate.

p. 13 line 14-15: “For encounter rates for individual whales...”

Comment: Does “individual whales” include all whales? Or does it exclude cows with calves, the comparison group in this case? Please clarify.

- Text has been changed to include this clarification – the term individuals included all whale sighted including mother and calf pairs.

Results from the GAMs constructed to detect interplay between oceanic indicators and encounter rates indicated slightly different inferences for all individuals including mother-calf pairs vs. mother-calf pairs only, counted as groups

p. 14 lines 51-54: "Surveys conducted in mid channel waters in 2018 when encounter rates in the main study are were exceptionally low did not provide any evidence of higher than expected numbers of cow-calf pairs in that adjacent area"

Comment: As noted above, the small numbers of animals seen in this one-season modification were not sufficient to support a "no-change" inference.

As above, this has been addressed

p. 15 line 42: "SPLISH" should be SPLASH

- This comments refers to the South-east Alaska surveys, actually known as SPLISH – see Neilson et al 2018.

p. 16 lines 25-28: "In this case, changes were asymmetrical; the rate of decline fluctuated, and ultimately far exceeded the rate of increase prior to carrying capacity (K)."

Comment: This sentence implies that carrying capacity (K) has been established for the present DPS when such is not the case. Please modify the sentence to better clarify that point.

This point has been clarified and amended. Indeed, we don't know the carrying capacity of the DPS. This section now discusses this in full detail and refers to indicators that can be used to assess the status of a population relative to its carrying capacity, based on things like vital rates and condition indices. A decision matrix is then used to assess the inferences of current observations in relation to carry capacity.

p. 17 lines 19-22: In the California Current System, humpback whales will switch between a predominantly fish based diet and a predominantly krill based diet..."

Comment: "Will" implies that we're seeing into the future here? Maybe change "will" to "are known to."

- Agreed, this statement has been changed as suggested

Responses to reviewer: 2

Page 4 Line 48 Describe what you mean by 'carrying capacity', because this term is hard to define and varies in the literature. Do you mean the population has surpassed 'K'? If so, describe what 'K is for this DPS. Or do you mean the ecosystem has shifted so fewer whales can be sustained?

The issue of carrying capacity is certainly complex, as the reviewer points out and as suggested, further explanation and a clear definition have been added to the manuscript.

Within the introduction, the comment has been amended to reflect the possibility that the population may have surpassed its carrying capacity. However, in the introduction, our goal was simply to highlight to the reader that a range of possible explanations needs to be considered. Therefore, in the interests of maintaining the flow of the paper at this point, the detailed discussion of this issue is located within the discussion. The section in the discussion where carrying capacity is discussed has been changed extensively. This section now opens with a clear definition of the definition being used here. The discussion of the status of this population in relation to carrying capacity is validated by references to theoretical studies that provide clarification and context. Additionally, through the inclusion of more recent, up to date studies reporting health and condition indicators for this DPS, the application of a decision matrix that combines condition and abundance estimates provides clear direction for reaching conclusions on the issues surrounding carrying capacity for this population segment. With these edits in place, this section now leads to a supported conclusion that the carrying capacity has shifted, and may now have reset to a lower point for this population segment.

Page 5 Lines 20-37 I suggest using an effort table to include years, 'season', days on the water of Beaufort 2 or less and nm surveyed. A table would be easier for the reader to see where the data gaps exist, if any.

-A reference to Table 1 and Table S1 is now included here. Table 1 includes years, season and Julian dates for the studies. Supplemental Table 1 includes distances travelled during each survey period. Details of distances are included in Table 3 and are also laid out in the introductory text at the start of the results section. Because the dataset presented here combines data from previous studies, there are pronounced differences in effort between years, but hopefully the survey design section now explains these in full.

Also the dataset is not continuous, so to emphasize this I have also inserted line breaks on the x- axes on the key figures in the results section to highlight to the reader where these the breaks occur.

Page 6 Line 3 Were the observers the same? Was there observer experienced? Convince the reader the data were collected by trained experienced observers and not an intern from Kansas.

Fully trained personnel collected the data. Comments have been added to the text to clarify this.

Line 17 Describe what you mean by escorts.

Extra details explaining this term have been added to the text.

Line 20 Were iPads used to record GPS in all study years

Good point – in the 2008-2010 study we used handheld GPS units for data collection. From 2013 onwards we used GPS enabled ipads.

Page 7 Line 19 explain ESRI

The full name of this organization (Environmental Systems Research Institute) is included in the previous sentence but the abbreviation was not added. This is now in place.

Page 8 Statistical Analysis and Environmental Parameters

How did you deal with the correlation among oceanographic events? Or were they considered as independent events? I suggest you seek the advice of an oceanographer who may have insights you have not considered.

A pre-test review of the variables conducted to identify significant correlations within the indices or between the indices and local SST in Hawaii, revealed just a single, moderate negative correlation between the TNI indices and the local SST in Hawaii (Pearson correlation coefficient $r = -0.753$, $p = 0.031$). This does not stand up to the altered requirements for significance in light of multiple testing ($\alpha/k = 0.0125$). Consequently, all variables were treated independently in the statistical analysis. Within the analysis, the sample size was insufficient to support models that included interaction functions between the explanatory variables; however, with a larger data set this would be useful next step. Potential connections are taken in consideration in the discussion, however at the advice of an oceanographer, the discussion section has been radically simplified in respect to the discussion of recent oceanographic events in the North Pacific.

Encounter rates

Page 9 Line 37 Results

As a reader I would like to see a comparison of effort between the two time periods. If it is presented anywhere it is not obvious. This will inform the reader if there is validity in documenting a downward trend if the type of effort is balanced among the early and later years. The number of days and nm transected would be easy to add to the other tables if Being a rate it is the number of groups with calves over some unit but that unit is unclear.

In the following sentence, the reader has no idea if the drop in encounter rates was due to effort In days surveyed or nm transected.

'Of these, 149 groups were encountered in 2013-2014 and 65 groups were encountered in 2017-2018, reflecting the changes in encounter rates between these time periods.'

The section in question here appears immediately under the heading Results and in this text, the exact details of distances and numbers of groups encountered are included.

These data are also included in supplemental table 3 and in table 3, however the surveys were not carried out continuously on specific days, rather, the surveys were interspersed among other activities, so the entire day wasn't spent on the survey. Therefore, the number of days spent on surveys is a bit mis-leading and would perhaps overstate the effort. In lieu, the details of distance covered will hopefully give readers a realistic perception of effort.

Page 11 Line 6 This section made me dizzy. Could you present the two groups of years and then the individual years in a table? Lots of information but a bit confusing to follow. In thinking about how to make this easier to follow are the early, mid and late divisions really necessary? Could you pick the timing that best represents the years. While I don't mind the three seasons and find the differences interesting it is hard to follow. I suggest present the results as an overall grouping between the two sets of years and then add the necessary seasons to support your findings.

In the section that the reviewer is referring to, the focus of the study in its initial inception was to detect seasonal changes in habitat use in the study area. As these seasonal differences are now part of the shifts we see in the region, we do feel that it is appropriate to include the seasonal categories.

However In table 3, the data is presented as the reviewer suggests here, with compiled data for the different time periods .The data is compiled for the entire season, to allow the reader to compare the data between years and time periods and then data for all late seasons is provided in a single section, so that apples are compared just to apples.... This data is also displayed in figure 3. In response to these comments, I have also cleaned up the narrative section that describes these data.

Page 12 Line 25 In the methods this is called Environmental Parameters....be consistent in the headers and layout between Methods and Results

This has been corrected and the heading Oceanographic conditions is now consistently used.

Page 13

Line 49 Encounter rates of calves? M/C? Adults?

This text has been clarified for meaning – these were encounter rates for all individuals

Line 52ish How does the timing work for the nutritional stress hypothesis? If declines were before 2015 could this indicate the warm water events impacted whales earlier?

Yes, the first signs of decline are prior to 2015 and this ties in still to the changes in the oceanographic conditions in the North Pacific. Oceanographers describe the beginning of the warming trend dating back to early 2013. However, the differences between 2013 and 2014 could be part of a typical cyclic trend - this is discussed in detail on line 620 in the revised version

Page 14 Line 42 Correction Gabriele not Gabrielle. Double check HMMC numbers from their 2018 season report.

Name correctly spelt. The numbers have been double checked and provided to HMCC for verification

Page 15 Line 45 GulfWatch Alaska has studied PWS humpback and herring since 2006. Results through 2009 are reported in a special issue from Deep Sea Research 2 Other surveys are reported in survey reports available from the AOOS website portal. This might be a good time to discuss the year prior to

pregnancy as a year a female needs to be in top fitness. If warm water events first began impacting GOA productivity in 2013 the subsequent poor calving years should have started to be seen in HI in 2015 and beyond. This is because poor body condition from nutritional stress on the feeding grounds in 2013 would have made 2014 a poor year to sustain a pregnancy in HI and AK and the resulting poor calving year would be first documented in HI the winter of 2015.

I believe this is the case and the additional data analysis included in this revision – conducted following the workshop in Oahu, supports this suggestion. When data describing oceanographic conditions are lagged by 2 years, the deviance explained increases two fold – from 35% deviance explained to almost 70%. The implication of this is that feeding conditions 2 seasons prior better predict variability in calf numbers in Hawaii waters. The manuscript has been edited to include these details.

Details of herring abundance in recent years were obtained from a local fisheries professional in Alaska, and used to inform the discussions within the text, regarding prey abundance. However, from a larger perspective, the scope of the discussion of the impact of the recent climate anomalies has been constrained to the established details (see below).

Page 16 Line 12ish Again think of the vagueness of the carrying capacity concept and explain your thinking in saying they really reached some point that caused a rapid decline in calves.

The reference to carrying capacity has been taken out of this comment, as this issue now covered in the more extensive discussion included in the relevant section.

Page 17

Line 20 Krill is not the primary prey in SEAK

This section has been re-written and now explains regional, seasonal and temporal prey preferences more accurately and in more detail:

In Alaska waters, humpbacks are known to feed on a similar variety of prey [79–81] however prey preferences vary regionally, seasonally and possibly at the individual level (Witteven et al. 2011 and Straley et al. 2017). In some regions of South-east Alaska and Northern British Columbia, mature euphausiids (*Thysanoessa sp*, *Euphausia pacifica*) comprise a targeted prey during the summer months, [79,82]. In other areas, such as Icy Strait and Glacier Bay, small schooling fish such as capelin (*Mallotus villosus*), juvenile wall-eye Pollock (*Gadus chalcogrammus*) and Pacific herring (*Clupea pallasii*) are primary prey targets and during fall, whales throughout the region may relocate and exploit seasonal aggregations of herring (Straley et al 2018). Line 628 on the revised manuscript.

Line 45 Again consult with an oceanographer to fully understand how the three oceanographic events interplay among each other.

As mentioned, following advice from a colleague, this portion of the discussion has been radically streamlined. Key comments are now confined to fully reported and established details of these events and more speculative references have been removed. Moreover, the inclusion of the 2 year advanced analyses with respect to ocean conditions in fact leads to a far more parsimonious interpretation of the results obtained in this study and renders the more complex discussion included in the original version of the manuscript more marginal. While it is still interesting to consider the stability and newly reported increases in numbers of humpback whales currently using California feeding regions, it is well beyond the scope of this study, and in recognition of this, the comments pertaining to this issue have been removed.

Page 18

Line 28 This is speculative and not based on actual data. Females MAY overwinter, a few have, in Alaska, but it is more likely they spend little time on the breeding grounds (or only go part way) and more time feeding because the migration is staggered hence giving the appearance of whales staying all winter in Alaska.

This correction has been made and Straley et al 2018 has been cited as this paper provides full details on the (low) incidence of over-wintering and increased persistence prior to or following migration in humpback whales in Alaskan waters

Line 42 ish Again carrying capacity needs to be clear as to what you mean by this term.

These comments have been removed as the issues is now discussed in detail under the sub-heading – Did the population reach carrying capacity?

References

Straley, J.M., Deep-Sea Research Part II (2017), <http://dx.doi.org/10.1016/j.dsr2.2017.08.008>

Declining reproductive rates in Hawaii's humpback whales, *Megaptera novaeangliae*, in response to recent climate anomalies in the North Pacific

Cartwright^{a,b}, A. Venema^a, V. Hernandez^a, C. Wyels^{a,c}, J. Cesere^c and D. Cesere^c

^aThe Keiki Kohola Project, Kihei, HI 96753

^bCalifornia State University Channel Islands, Department of Environmental Science and Resource Management, One University Drive, Camarillo, CA 93012

^cCalifornia State University Channel Islands, Department of Mathematics, One University Drive, Camarillo, CA 93012

^dCesere Brothers Fine Art Photography, Paia, HI 96779

Keywords: Climate change, marine heatwave, marine mammals, reproduction, humpback whales

1. Summary

Alongside changing ocean temperatures and ocean chemistry, anthropogenic climate change is now impacting the fundamental processes that support marine systems. However, where natural climate aberrations mask or amplify the impacts of anthropogenic climate change, identifying key detrimental changes is challenging. In these situations, long-term, systematic field studies allow the consequences of anthropogenically driven climate change to be distinguished from the expected fluctuations in natural resources. In this study, we describe fluctuations in encounter rates for humpback whales, *Megaptera novaeangliae*, between 2008 and 2018. Encounter rates for humpback whales were assessed during transect surveys of the Au'au Channel, Maui, Hawaii. Rates initially increased, tracking projected growth rates for this population segment reaching a peak in 2013, then declining through 2018. Specifically, between 2013 and 2018, mother-calf encounter rates dropped by 76.5%, suggesting that the reproductive rate of the newly-designated Hawaii Distinct Population Segment of humpback whales is currently in rapid decline. As this decline coincides with changes in the Pacific Decadal Oscillation, the development of the NE Pacific marine heat wave and the evolution of the 2016 El Niño, this decline may comprise another example of the impact of this potent trifecta of climatic events within the North Pacific.

*Author for correspondence (rachel.cartwright@csuci.edu).

†Present address: California State University Channel Islands, Department of Environmental Science and Resource Management, One University Drive, Camarillo, CA 93012

rachel 1/5/2019 3:45 PM

Comment [1]: Minor improvement in grammar

Rachel 1/5/2019 5:31 AM

Deleted: ; they reached a peak in 2013, then began to decline

Rachel 1/4/2019 7:31 AM

Deleted: 8

rachel 1/5/2019 3:45 PM

Comment [2]: Change made to improve precision in the abstract

Rachel 1/4/2019 7:29 AM

Deleted: almost 80%,

Rachel 1/4/2019 3:21 PM

Deleted:

2. Introduction

As global atmospheric carbon dioxide levels pass the 400-ppm threshold [1] the biological impacts of climate change on marine systems are becoming increasingly widespread[2]. Within the marine realm, changes in ocean temperature and ocean chemistry are generally recognized as the primary direct consequences of climate change [3,4]however mounting evidence indicates that the fundamental processes which support marine systems are also being impacted [2,5–7]. For some marine organisms and ecosystems, these changes may signal the onset of a downward spiral from which recovery is unlikely [8–10]. For other marine fauna, the eventual outcome of these changes may be harder to predict [11,12] or could be beneficial [13–15]. Accurately predicting possible outcomes is challenging, especially where naturally occurring climate anomalies and fluctuations may act synergistically, potentially amplifying or masking perturbations associated with anthropogenically driven climate change (e.g. [16,17]). Additionally, while oceanographic conditions have been closely monitored in recent years, the required complimentary long-term data sets documenting marine biological resources are comparatively sparse[2,18]. One possible solution is to mine past studies and assemble data that can be used to investigate the links between marine resources and environmental forcing [18]This key information can then be applied to ensure that management strategies accurately target the most detrimental impacts of climate change.

Within the North Pacific system, the last decade may be characterized as a period of pronounced climate variability. At high latitudes, mean sea surface temperatures (SSTs) have been rising steadily (estimated rate $0.7^{\circ}\text{C decade}^{-1}$; [19] while local increases in SSTs have been even more pronounced: Between 2015 and 2017, seasonal summer temperature anomalies of +4 to +7°C were reported for the Barents and Chukchi Seas [19]. As sea temperatures rise, arctic sea ice levels are falling and a profoundly different climate regime is emerging across the Pacific Arctic [7]At lower latitudes, increases in mean SSTs in the North Pacific have been more moderate ($0.12^{\circ}\text{C decade}^{-1}$;[20]), however any potential stability has been eclipsed by a trifecta of other climate anomalies. In the summer of 2014, the Pacific Decadal Oscillation (PDO), a basin-wide system that acts at a multi-decadal level [21], flipped from a strong, consistent negative phase to a pronounced positive phase [22,23]. In its negative phase, the PDO is characterized by cooler water temperatures across the Central Pacific and strong upwellings in coastal waters along the Eastern North Pacific [24]. During a positive phase, SSTs rise and coastal upwellings weaken. Coinciding with this transition, an additional oceanic anomaly, comprised of a massive lens of warm water, began to develop in the NE Pacific. Originally the anomaly was centered in the offshore waters of the Gulf of Alaska, however during the summer of 2014, it began to move east, quickly spreading along the shelf of North America and coastal Alaska, and acquiring the widely used nickname “the blob” [25]. By the winter of 2014, associated SST anomalies of $>+3^{\circ}\text{C}$ were reported [17,26]. The summer of 2015 then saw rising SSTs in the West-central Pacific, signs associated with the beginning of a strong El Niño / Southern Oscillation (ENSO) event [27]. The typical hallmarks of these 2-3

Rachel 12/19/2018 4:39 PM

Formatted: Not Highlight

Rachel 12/19/2018 4:39 PM

Deleted: sentinel

year systems also include warming SSTs and reduced coastal upwellings; potentially, these effects further amplified the on-going anomalies already playing out in Central North Pacific.

To date, a wide range of mass mortality events and other biological disruptions across the North Pacific and the Gulf of Alaska have been causally connected to these unusual conditions [17,26]. The warming of coastal waters associated with all three of these anomalies led to increases in rainfall and freshwater coastal runoff [28] and reduced surface winds [25]. As a result, stratification increased, minimizing coastal upwellings. Nutrient transport into the mixed layer was then suppressed, leading to extremely low productivity, as evidenced by low chlorophyll levels beginning in the winter of '14 [28]. Compounding the inherent challenges associated with low productivity levels, horizontal advection, whereby cool water species migrate north and warm water species expand their range, potentially triggered a range of cascading impacts [25]. So far, casualties associated with these disruptions include a mass die-off of common murre, *Uria aalge*, along the northern coastline of the Gulf and widespread mortalities in tufted puffins (*Fratercula cirrhata*) in the Bering Sea [29]. A large-scale mortality event for Cassin's Auklets (*Ptychoramphus aleuticus*) along the Pacific Northwest coastline has also been attributed to this combination of unusual conditions [30].

For migratory mysticetes such as the humpback whale, *Megaptera novaeangliae*, fitness and success is entirely dependent on the availability of adequate prey resources on high latitude feeding grounds. As capital breeders, migratory mysticetes exploit high latitude prey resources during summertime feeding seasons, then depend entirely on stored energy reserves to support seasonal migration and wintertime breeding activities in low latitude regions [31]. For female mysticetes, the successful completion of each stage of reproduction is contingent upon the adequate availability of stored energy reserves. Prior to pregnancy, an increase in stored energy reserves has been detected in several migratory mysticetes (e.g. gray whales, *Eschrichtius robustus* [32], fin whales, *Balaenoptera physalus* [33] and North Atlantic right whales, *Eubalaena glacialis* [34]). Based on comparative studies of other large mammals, potentially this triggers ovulation via the release of leptin from adipose cells. Leptin then orchestrates the secretion of gonadotrophin releasing hormone and luteinizing hormone from the hypothalamus and the pituitary, which in turn stimulate ovulation [35]. Continuing through the post-conception period, accumulated energy reserves support fetal growth and development; maternal mysticetes typically utilize up to 25% of these stores during this period [36]. Subsequently, during the postnatal period, remaining reserves are mobilized to meet the demands of early lactation. Over this period, changes in body shape indicate that maternal females typically use a further 20-35% of their stored reserves during this period [34,37].

Potentially reflecting these elevated energetic requirements, multiple studies have demonstrated clear connections between mysticete reproductive rates, nutritional resources and oceanic conditions. For example, in gray whales, reproductive rates increased following seasons in which sea-ice conditions

Rachel 1/4/2019 3:38 PM

Deleted: adiposity

Rachel 1/2/2019 6:47 AM

Deleted: increase

Rachel 1/1/2019 8:35 AM

Deleted: adipose stores

Rachel 1/1/2019 8:36 AM

Deleted: their adipose

Rachel 1/1/2019 8:36 AM

Deleted: adipose stores

extended temporal access to preferred feeding grounds in the Bering Sea, [38,39]. Once climatic conditions changed and access to feeding regions was limited, reproductive rates declined [38]. Similarly, in North Atlantic right whales, notable increases in reproductive rates during the 90's were closely matched to increases in the availability of their preferred prey, *Calanus finmarchicus*. These prey increases were driven by favorable oceanic conditions in the Gulf of Maine, which were in turn related to climatic anomalies in the near Arctic [40]. When a distinct climatic shift altered oceanic conditions in the Gulf of Maine, reducing the availability of *C. finmarchicus*, North Atlantic right whale reproductive rates also went into decline [41,42] . Additional studies describe both short and long term periodicity in these fluctuations, ranging from one to six years previous depending on the life cycle of the prey species [43]. Taken cumulatively, these studies elucidate the mechanisms through which mysticete reproductive rates respond to prey availability and oceanic conditions.

In this study, we provide details of a decade-long study documenting fluctuations in the reproductive rates of humpback whales in the waters around Maui, Hawaii. These whales comprise the Hawaii Distinct Population Segment (DPS) of humpback whales, a new definition that broadly describes the portion of North Pacific humpback whales that use Hawaiian waters as a breeding ground [44]Current estimates suggest the Hawaii DPS comprises around 60% of the larger North Pacific population [44,45]Our study was conducted over an 11-year period from 2008 to 2018, in the Au'au Channel between West Maui and Lanai. This region comprises primary nursery habitat and is used by approximately 85% of humpback whale mother and calf pairs in Hawaiian waters over the winter breeding season [46–49]. Data from the first time-period (2008–2010) stems from a previously published study [47] but is used here to extend the time-series, as advocated by Doney et al.[2].

At the outset, the initial goal of this study was to detect seasonal changes in habitat use within the study area. As unforeseen fluctuations in encounter rates became increasingly apparent, the study was extended to capture changes in comparative encounter rates, and between 2013 and 2018, we documented a clearly evident decline in the numbers of mother-calf pairs encountered within the study area. We review a range of possible scenarios that could account for these observations. These, including changing patterns of habitat use, or the possibility that this population segment may have reached, or surpassed its carrying capacity. Finally, we compare the changes in encounter rates to oceanographic indicators, using models that incorporate key oceanographic indices to capture the recent climate anomalies in the North Pacific, with a view to detecting any potential linkages that may exist between these recent events and the fluctuations in reproductive rates in the Hawaii DPS documented in this study..

3. Materials and Methods

- Rachel 1/1/2019 8:44 AM
Deleted:
- Rachel 1/2/2019 7:00 AM
Deleted: However a
- Rachel 1/1/2019 9:30 AM
Deleted: . Overall, we found very little evidence of any changes in patterns of habitat use. In contrast,
- Rachel 1/1/2019 9:31 AM
Deleted: over the course of the study.
- Rachel 1/2/2019 6:09 AM
Deleted: ng c
- Rachel 1/2/2019 6:08 AM
Deleted: es
- Rachel 1/2/2019 6:08 AM
Deleted: in
- Rachel 1/2/2019 6:07 AM
Deleted: and
- Rachel 1/2/2019 6:07 AM
Deleted:
- Rachel 1/1/2019 9:32 AM
Deleted:
- Rachel 1/1/2019 9:36 AM
Formatted: Not Highlight
- Rachel 1/2/2019 7:02 AM
Deleted: .

Study Site

The study was conducted in shoreline to mid-channel waters, along the eastern portion of the Au'au Channel, West Maui (~20°52'N, 156°40'W). The Au'au channel lies between the islands of Maui and Lanai and features gently sloping shoreline gradients, maximum water depths of ~150 m and complex mid-channel topography that includes sea mounts and ridgelines, interspersed between steep-sided sandy basins [50]. The study area (figure 1) was designed to include the range of habitats available in this area, extending from the Maui shoreline to either the mid- or deepest point of the channel at each minute of latitude, whichever lay furthest from the Maui shoreline (these locations fell between 8 and 10 km offshore). Northern and southern limits were set within the lee provided by the West Maui Mountains, thereby minimizing local variations in sightability and sea state across the study area. Lahaina Small Boat Harbor lies within the study area; this is a key hub for local whale watching and other ocean-tourism related activities. The study area in total covered 124.5 km².

Timing

The study spans the years 2008 to 2018, however timing of data collection varied over the course of the study (see Table 1 and Table S1). In 2008 to 2010, all surveys were conducted in the latter portion of the season (mid-March onwards). These results have been published previously [47] and were used to establish maternal patterns of habitat use in the region. Over 2013 and 2014, surveys were conducted during three two-week periods, in mid-January, mid-February and mid-March. Survey protocols used in 2008 to 2010 were repeated to allow comparisons of the results. The aim of these surveys was to capture any seasonal or temporal variability in patterns of habitat use. Surveys were re-instated in 2016, but limited to late season (mid-March) due to poor weather. In 2017 and 2018, complete surveys that matched all aspects of the 2013-2014 surveys in terms of timing and protocol were completed using the same survey protocols. All surveys were restricted to favorable sea conditions (\leq Beaufort 3) in order to ensure consistency in weather / sightability conditions.

Survey Design

A system of parallel waypoints at 1 minute of latitude intervals across the study area was established, as shown in figure 1. Inner waypoints were located between 250 to 500 meters from the nearest shoreline depending on local topography; outer waypoints were located at the deepest or mid-point of the channel, whichever lay further offshore). Surveys were conducted across the study area along equally spaced zigzag sampling transects set at 2° intervals between these waypoints. This ensures equal probability of coverage across the site, only complete transects are included and completed transects then comprise independent samples [51].

Daily starting points were chosen based on prevailing weather, in order to ensure consistent sighting condition while still maintaining proportional coverage of different habitat types across the study area.

Rachel 12/20/2018 6:34 AM

Deleted: offshore

Rachel 12/20/2018 6:46 AM

Deleted: .

Rachel 1/4/2019 3:54 PM

Deleted: Over 2013 and 2014, the study was extended across the entire season; surveys were conducted during three two-week periods each year and followed the same survey protocols used in 2008 to 2010.

Rachel 1/4/2019 3:55 PM

Deleted: Surveys were re-instated in 2016, however logistic and local weather challenges limited surveys to the late season time window (mid-March).

Rachel 1/4/2019 3:58 PM

Deleted: . In 2018, additional surveys were added in mid-channel waters to capture any local relocation.

Rachel 1/4/2019 3:59 PM

Deleted: uniform

Rachel 1/4/2019 3:59 PM

Deleted: of

Rachel 12/20/2018 7:00 AM

Deleted: <2

Rachel 12/19/2018 5:27 PM

Deleted: (figure 1

Rachel 12/20/2018 10:27 AM

Deleted: -

Following the protocols established by Strindberg and Buckland [51], as long as only complete transects are included, completed transect legs between waypoints comprise independent samples. While more stringent methods are required for establishing abundance estimates, this survey method provided comparative encounter rates within the study area, and as identical methods dating back to 2008 were applied, encounter rates could be compared between different time intervals.

Survey effort varied between years over the course of the study (see Table S3 for full details), reflecting the different goals of the study during each time period. Within the first survey period (2008-2010), the primary goal of the study was to establish habitat use patterns within the study area. In order to provide an adequate sample size to ensure appropriate power during data analysis, multiple surveys of each transect line were conducted (see Cartwright et al [47]). During 2013-2014 and 2017-2018, the primary goal of this study was to detect seasonal changes in habitat use. To meet the requirements of this study, surveys were set up to ensure consistent coverage within the study area during early, mid- and late season over a two year period. Over the span of the two seasons, the entire set of zig-zag transect lines (86.7 km) was covered within each of three two-week windows. Cumulatively, this provided three replicates of the entire set of transect lines, within a two year window. In 2016, the surveys were re-instated with the primary goal being to quantify temporal changes in sighting rates between years. Essential to this goal was the maintenance of consistent methods, so that the data could be compared with baseline data compiled previously. Based on distances covered within each season during the 2103-2104 surveys, a goal of approximately 50 kms coverage along the original lines was chosen, as this would provide comparable data to that obtained during the 2013 surveys (see table S3). As previously, only completed transect lines could be included. This ensures that sightings were proportional to the available habitat in the area, as advocated by Strindberg and Burnham [51], but leads to a small amount of variability in total distance travelled. The two week time windows used in 2013-2014 were adopted as target time periods. To balance the competing goals of maximizing the sample size while still replicating the basic survey design established previously, each zig-zag transect was surveyed at least once. If the goal of covering 50 km had not been met, a single additional line was covered, but no line was surveyed more than 2 times in any sample period.

Throughout the full extent of the study, two different survey vessels were used throughout the study (a 6 m and an 8 m powerboat). Vessels travelled at approximately 9km hr^{-1} (5 knots) along the survey lines.

Throughout the extent of the study, the same research team members supervised the collection of field data. Two experienced and fully trained designated naked-eye observers scanned on opposite sides of the vessel; any sightings within 90 degrees on either side of the forward bow and within an estimated 1 km to either side of the survey line were recorded. Locations of sighted whales were recorded after the whales(s) left the surface, as latitude and longitude waypoints, using handheld GPS units in the 2008-2010 study and on GPS enabled Ipads from 2013 onwards.

R. Soc. open sci.

Rachel 1/4/2019 4:23 PM

Deleted: T

Rachel 12/20/2018 5:46 PM

Deleted: .

Generally, as humpback whales dive they leave a footprint (a vortex of flattened water) that persists at the surface, so wherever possible, this was used as a marker. Group composition was established following protocols described in Cartwright et al. 2012 [47]. In brief, calves were identified based on comparative body size of less than 1/3 of the maternal female's (mother's) body length, the mother was recognized by her close association to the calf. All other whales within 2 body lengths of the mother or other whales in the group and moving in a coordinated pattern were assumed to be associated and designated as escorts to the mother and calf group. These are presumed to be male whales, associated in pursuit of potentially seeking mating opportunities with receptive females [52,53] All other individuals were identified as adults of unknown gender.

rachel 1/5/2019 3:45 PM

Comment [3]: Additional references provided

Groups containing calves spend more time at the surface, so detection probability, $g(0)$, was likely slightly higher for mother-calf pairs vs. adult animals (based on surface vs. submerged time estimates in Mobley et al [54], travelling at 9.4 km hr^{-1} , mean $g(0) = 0.313$ for adult groups vs. 0.360 for mother-calf groups). Detection probability would also vary slightly with group size; consequently, the results presented here reflect relative, not absolute densities of whales in the region. Still, as effective strip width for humpback whales on boat based surveys in Californian waters with a set speed of 5 knots has been previously estimated as 3.2 km [55], we assume that sightability within a 1 km strip width within Hawaiian waters would approach 100%. Fluke photo ID's (as per [56]) and surface images documenting body markings, lesions and other scars were compiled for all sighted animals and used post-hoc to eliminate any chance of pseudo replication over the course of the day, between survey vessels or within regions of over-lap at the beginning or the end of any successive transects. Any whales that were identified as resights within the same day, either in the field during data collection or during post-hoc analysis, were recorded only once, at the first encounter. . Resights between different days were rare (detected in < 2% of sightings) and these were included in the data set, as the focal estimated rate in this study was not an abundance estimate, rather the goal was to estimate the daily encounter rate per km travelled.

Rachel 12/19/2018 5:30 PM

Deleted: o

In 2018, in response to exceptionally low mother-calf group encounter rates within the study area and anecdotal reports from local whale-watching vessels of higher numbers of whales in mid-channel waters, a small set of ad-hoc additional surveys were conducted running along transects between the outer waypoints of the study area (see table S4). Encounter rates for these surveys were not incorporated into the analysis of habitat use nor included in yearly encounter rates for the larger study area and are provided here simply as an anecdotal set of observations that might potentially inform future in-depth studies in this area This outer transect bisected the deeper, mid-channel waters of the Au'au Channel. Survey protocols were consistent with those used in the main study area; a complete mid-channel transect between odd-numbered transects (1 to 11) was conducted during the early, mid and late season time windows in 2018.

Rachel 1/1/2019 12:28 PM

Deleted: added

Rachel 1/1/2019 12:24 PM

Deleted: . These surveys were conducted along transects

Rachel 1/1/2019 12:25 PM

Deleted: As these waypoints were located in either the mid-point or the deepest point of the channel within the study area,

Rachel 1/1/2019 12:25 PM

Deleted: t

Rachel 1/1/2019 12:26 PM

Deleted: The primary goal of these mid-channel surveys was to detect any local, fine scale changes in habitat use in mother-calf pairs that could potentially explain the notable changes in encounter rates in the main study area. Mid-channel survey data were used for within-year comparisons for 2018 only. They were not incorporated into the analysis of habitat use nor included in yearly encounter rates for the larger study area.

Figure 1: The study area

Data analysis

Spatial analysis

A Geographic Information System model (GIS) was constructed using ArcGIS 10.5 (Environmental Systems Research Institute ([ESRI](http://www.esri.com))). A base map was obtained from ESRI (https://services.arcgisonline.com/ArcGIS/rest/services/Ocean/World_Ocean_Base/MapServer), and coastline data came from the Hawaii Data Clearinghouse. Water depth was obtained from the Main Hawaiian Islands Multibeam Synthesis website and incorporated as a 50 m bathymetric grid (<http://www.soest.hawaii.edu/HMRG/Multibeam/index.php>). A 750 m buffer constructed around the survey line provided coverage of 86% of the study area without overlap between mid-sections of adjacent transects. As sightings within an estimated 1 km had originally been recorded, this also reduced any potential edge effect, ensuring that all sightings across the width of the transect strip were captured. Sightings that fell beyond the buffer were discarded, as were sightings from incomplete transects. Although this did reduce the sample size slightly, Strindberg and Buckland [51] advocate these steps as a method of maintaining equal probability coverage across the survey area. For each encounter, water depths were compiled using the extract values to points function within the spatial analyst toolbox and distance from shore was obtained using the near function under the proximity tool within the Analysis toolbox.

Encounter rates

Encounter rates were calculated for all surveys, as the number of individual whales encountered (i.e. the total number of whales in the group, including any calves) and the numbers of groups encountered per km travelled along the transect lines. Encounter rates were classified by year of sighting and sub-classed by season (early, mid-, late), and by social composition (mother-calf (MC) vs. adult (A) groups), based on the presence or absence of a calf in the group.

Oceanographic conditions

Satellite-derived sea surface temperatures (SSTs) for the study area were accessed via <https://coastwatch.pfeg.noaa.gov/erddap>. The Multi-scale Ultra-high Resolution (MUR) SST Analysis fv04.1, Global, 0.01°, 2002-present dataset (available at https://podaac.jpl.nasa.gov/Multi-scale_Ultra-high_Resolution_MUR-SST) was used as this provided high-resolution SST data for the study area over the entire time span of the study. First, monthly mean composite SSTs were acquired for the Au'au Channel (20.7N to 21.0N; -156.9E to -156.6E) and used to determine the variability in local wintertime (January to March) SSTs over the course of the study. Subsequently, daily SSTs were compiled for the two-week window

R. Soc. open sci.

Rachel 12/20/2018 6:31 PM

Deleted: Environmental parameters

within which the surveys had been conducted each year (according to dates shown in table 1) and used to calculate mean estimates of SSTs for each unique survey period.

Monthly index values for the Pacific Decadal Oscillation (PDO) were obtained at <http://research.jisao.washington.edu/pdo/PDO.latest.txt>; this index provides standardized monthly values for the PDO, derived as the leading principal component of monthly SST anomalies in the North Pacific, poleward of 20°N [21]. El-Niño Southern Oscillation (ENSO) data were obtained from NOAA (National Ocean and Atmospheric Administration) at http://origin.cpc.ncep.noaa.gov/products/analysis_monitoring/ensostuff/ONI_v5.php. This index, referred to as the ONI index (ONI: Oceanic Niño Index), is the operational definition used by NOAA to predict the likely development and persistence of El Niño weather events. Additionally, we compiled indices from the TNI index database (TNI: Trans-Niño Index; <https://www.esrl.noaa.gov/psd/data/correlation/tni.data>). This index measures the gradient in SST anomalies between the central and eastern equatorial Pacific and is used specifically to identify Central Pacific El Niño events [57,58]. First these indices were reviewed to establish the basic trends in oceanic conditions over the duration of the study. Next, reflecting the findings of Seyboth et al. [59] and Nash et al. [60], indices were advanced by one, then two years. Monthly values from May to October were compiled to provide a mean composite annual value that reflected North Pacific oceanic conditions in the preceding feeding season (i.e. for example; encounter rates for 2008 were associated with oceanic indices in 2007 feeding season). These advanced indices were then used in the models constructed (see below).

Statistical analysis

Statistical analyses were conducted using SPSS version 24, and R version 3.4.0 [61]. Where standard parametric and non-parametric inferential tests were used, significance was set at 0.05 and Bonferroni corrections included where multiple identical tests were conducted. Where models were constructed, model performance was evaluated based on Δ AICc values and deviance explained, as statistical tests and associated p-values may be unreliable when used with smaller datasets, such as this [62].

Changes in habitat use within the study area between years, social groups and seasons were investigated using survey data from 2013-2014 and 2017-2018, as surveys were conducted throughout the season in these years. Variables were incorporated into a multi-variate ANOVA test (MANOVA); depth of water and distance from shore for each group were incorporated as dependent variables and season (early, mid- and late season), study period (2013-2014 and 2017-2018) and group composition (mother-calf (MC) vs. adult groups (A)) were used as fixed factors. After confirming that all assumptions of the multivariate MANOVA were met, interpretation was based on Wilks Lambda statistics. To detect any long-term trends over the full extent of the study, a second analysis was conducted using group locations from 2008 to 2018. It should be

Rachel 1/4/2019 4:42 PM

Deleted: and m

Rachel 12/21/2018 9:02 AM

Deleted: initially

noted this analysis included only late season sightings, so fixed factors were limited to study period (2008-2010, 2013-2014 and 2017-2018) and group composition.

Variations in encounter rates were investigated using series of generalized additive models (GAMs; [63]), constructed using the “mgcv” package for R [64]. In the first sequence of models temporal trends were investigated. Separate models were constructed for numbers of individual whales encountered and for mother-calf groups; encounter rates were used as the response variable while season and year were used as explanatory variables. Season was included as a factor. Based on the examination of exploratory scatter plots, year was included as a smoothed variable to allow detection of non-linear trends. Thin plate penalized regression splines were used (this is the default setting in “mgcv”) and as the sample dataset is small, the number of knots was constrained to 4, as advocated for a sample of this size [65].

In a second set of models, the impact of key oceanic parameters was explored. Encounter rates for individual whales, for mother-calf groups and for adult only groups comprised the response variables, with separate models were constructed for each variable. Local SSTs, PDO, ONI and TNI indices were incorporated as explanatory variables. For the local SST, mean estimates for the specific two-week window in which the surveys were conducted (described in the methods section above) were used. For the three oceanic indices (PDO, ONI and TNI), the composite mean summertime values (May through October) were compiled. Initially, these summertime composite values were then advanced by one year, so that they reflected conditions in the previous feeding season. In a follow-up analysis, these data were advanced by two seasons, so encounter data for 2008 was compared to composite mean summertime values for 2006 and so forth through the extent of the time-period included in the dataset. Prior to testing, the data were screened for normality, the detection of outliers and for co-linearity among the explanatory variables; the data were normal, no outliers were present, indications of collinearity were limited to a moderate, negative correlation between the TNI indices and local SST’s in Hawaii ($r = -0.753$), however the significance of this outcome did not withstand correction for multiple testing ($p = 0.031$, $\alpha / k = 0.0125$). All other r values fell below 0.6. As in the first set of models, continuous variables were smoothed to allow for the detection of non-linear trends, the default setting of thin plate penalized regression splines were used and the number of knots was constrained to 4 for each model [65].

4. Results

Across the 11-year period from 2008 to 2018, a total distance of 1334.15 km was covered and the locations of 366 groups that included 875 whales were recorded. Between-year differences in the mean Julian date for surveys was ≤ 3 days. Distance surveyed also varied between years (see table 3 and table S3). In 2008-2010, surveys covered a distance of 731 km but limited to late season and a total of 148 groups were sighted.

- Rachel 1/1/2019 11:55 AM
Deleted: ; e
- Rachel 1/1/2019 12:00 PM
Deleted: and
- Rachel 12/21/2018 9:04 AM
Deleted: two
- Rachel 1/1/2019 11:56 AM
Deleted: , with l
- Rachel 1/1/2019 11:56 AM
Deleted: the
- Rachel 1/1/2019 11:57 AM
Deleted: ,
- Rachel 1/1/2019 11:57 AM
Deleted:)
- Rachel 1/1/2019 11:58 AM
Deleted: previous
- Rachel 1/1/2019 11:58 AM
Deleted: were used
- Rachel 12/21/2018 9:28 AM
Deleted: .
- Rachel 12/21/2018 9:29 AM
Deleted: t
- Rachel 12/21/2018 9:07 AM
Deleted: and there were no i
- Rachel 12/21/2018 9:10 AM
Deleted: (
- Rachel 12/21/2018 9:10 AM
Deleted: r values < 0.6).
- Rachel 12/20/2018 6:20 PM
Deleted: 1372.5
- Rachel 1/1/2019 12:13 PM
Deleted: 385
- Rachel 1/1/2019 12:13 PM
Deleted: 918
- Rachel 12/20/2018 6:18 PM
Deleted:
- Rachel 12/20/2018 6:19 PM
Deleted: , however
- Rachel 12/20/2018 6:09 PM
Deleted: Between-year differences in the mean Julian date for surveys was ≤ 3 days
- Rachel 1/4/2019 5:22 PM
Deleted: 1

Surveys conducted at two week intervals throughout 2013-2014 covered a total distance of 260.2 km and 149 groups were sighted. During 2017-2018, surveys were again conducted at two week intervals throughout the season. A total distance of 292.5 km was surveyed and 65 groups were encountered.

Table 1: Survey dates for transect surveys conducted in the Au'au Channel, Maui, 2008-2018

Fine scale patterns of habitat use

Comparing sightings recorded throughout the season in 2013-2014 and 2017-2018, group locations (represented by depth of water and distance from shore) varied according to season but not with the survey time- period or group composition (MANOVA for season; $F_4 = 3.271$, $p = 0.012$, for survey time- period; $F_2 = 1.022$, $p = 0.362$ and for group composition; $F_4 = 2.163$, $p = 0.118$). Within the effect of season, post-hoc testing indicated that depth of water was not significantly different between seasons ($F_2 = 1.572$, $p = 0.210$), but distance from shore was significantly different ($F_2 = 6.517$, $p = 0.002$). Groups were encountered significantly closer to shore during mid-season, vs. early season (using pooled data for all social groups; mean distance from shore = 5.8 km in early season and 4.3 km in mid-season, $\alpha/k = 0.0166$; post-hoc Tukey; $p = 0.012$), but differences between early vs. late season ($p = 0.033$) and mid- vs. late season were not significant ($p = 0.990$). When 2-way and 3-way interactions were examined, the only significant interaction included all three factors, but this did not withstand corrections for multiple testing ($\alpha/k = 0.0125$; $F_4 = 2.637$, $p = 0.034$). Repeating the analysis over the full extent of the study (2008 to 2018; late season data only), there were no significant differences in group locations between different survey time- periods ($F_4 = 1.604$, $p = 0.173$), according to group composition ($F = 0.891$, $p = 0.412$) or associated with the interaction between these factors ($F = 1.478$, $p = 0.208$) (see table 2).

Table 2: Locations of humpback whales classified by social composition in the Au'au Channel, Maui, between 2008 and 2018

Encounter rates

All groups, individual counts,

Assessing encounter rates for individual whales, trends over the course of the study were non-linear. Late season sightings encompassed the longest time range, spanning from 2008 to 2018. Over this 11-year period, encounter rates increased initially, from 0.42 whales km^{-1} in 2008 to a peak of 1.12 whales km^{-1} in 2013, and then declined to a low of 0.14 whales km^{-1} by 2018 (see figure 2 table S3). However it should be noted, these estimates reflect late season encounter rates and trends may be impacted by changing seasonal peaks. Comparing data collected in 2013, 2014, 2017 and 2018 may be more robust as surveys were conducted

Rachel 1/1/2019 12:22 PM

Deleted: p

Rachel 1/4/2019 5:21 PM

Deleted: Surveys conducted at two week intervals during 2013-2014 and 2017-2018 covered a total distance of 260.2 km; 214 groups that included 511 individual whales were encountered. Of these, 149 groups were encountered in 2013-2014 and 65 groups were encountered in 2017-2018, reflecting the changes in encounter rates between these time periods.

Rachel 1/4/2019 5:22 PM

Deleted: Based on the outcome of the MANOVA

Rachel 1/1/2019 12:18 PM

Deleted: study

Rachel 1/1/2019 12:19 PM

Deleted: study

Rachel 1/1/2019 12:21 PM

Deleted: study

Rachel 1/4/2019 7:52 AM

Deleted:

Rachel 1/4/2019 7:44 AM

Deleted: table 3,

throughout the season. Comparing mean encounter rates for the entire season across two-year brackets (2013-2014 vs 2017-2018) encounter rates fell by 63% (from 1.39 whales km⁻¹ in 2013-14 to 0.51 whales km⁻¹ in 2017-18). Comparing mean encounter rates for individual years (2013, 2014, 2017 and 2018), encounter rates fell from 1.66 whales km⁻¹ in 2013, to 0.90 whales km⁻¹ in 2014, 0.63 whales km⁻¹ in 2017 and 0.38 whales km⁻¹ in 2018. With respect to seasonality, encounter rates for all individual whales typically peaked during mid-season, with the single, but notable exception of 2018 when sighting rates peaked in the early portion of the season (see figure 2).

Rachel 12/21/2018 9:14 AM

Deleted: for

Figure 2: Encounter rates for humpback whales (counts of individuals) in the Au'Au Channel, Maui, between 2008 and 2018

Sightings of mother-calf vs. adult groups

When classified by social composition, trends varied between different group compositions. For mother-calf groups (MC) over the longest time span (2008 to 2018), late season encounter rates increased from 0.10 MC groups km⁻¹ in 2008 to 0.32 MC groups km⁻¹ in 2013, then declined to a low of 0.04 MC groups km⁻¹ in 2018. Similarly, late season encounter rates for adult groups (A) increased from 0.06 A groups km⁻¹ in 2008 to 0.11 A groups km⁻¹ between 2008 and 2013, then declined to a low of 0.02 A groups km⁻¹ by 2018. Comparing mean encounter rates for the entire season across two-year brackets (2013-2014 vs 2017-2018) encounter rates fell by 76.5% for MC groups (from 0.34 groups km⁻¹ in 2013-14 to 0.08 groups k⁻¹ in 2017-18) and by 39% for A groups (from 0.23 groups km⁻¹ in 2013-14 to 0.14 groups km⁻¹ in 2017-18). Comparing mean encounter rates for 2013, 2014, 2017 and 2018: For MC groups, mean annual encounter rates declined consistently between 2013 and 2018 (from 0.40 MC groups km⁻¹ in 2013, to 0.23 MC groups k⁻¹ in 2014 to 0.11 MC groups k⁻¹ in 2017 and 0.04 MC groups km⁻¹ in 2018). In contrast, adult encounter rates declined between 2013 and 2014, but then remained relatively consistent between 2014 and 2018 (0.28 A groups km⁻¹ in 2013, 0.14 A groups km⁻¹ in 2014, 0.16 A groups km⁻¹ in 2017 and 0.12 A groups km⁻¹ in 2018; see figure 3, table 3).

Rachel 12/21/2018 9:16 AM

Deleted: for

rachel 1/5/2019 3:45 PM

Comment [4]: A small rounding error was picked up in the final check through 0.078, should be rounded to 0.08 not 0.07

Rachel 1/4/2019 7:55 AM

Deleted: 79.4

Rachel 1/4/2019 7:53 AM

Deleted: 07

Table 3: Encounter rates for humpback whale groups classified by group composition, sighted during transect surveys conducted in the Au'Au Channel, Maui, 2008-2018

Figure 3: Encounter rates for a) mother-calf groups and b) adult groups during transect surveys of the Au'Au Channel, Maui, between 2008 and 2018

Modelling results

GAMs constructed to further explore temporal and seasonal changes in encounter rates indicated that for encounter rates for all individuals, the best performing models were obtained using year as a smoothed predictor. Year explained 49.9% of the deviance for encounter rates for individual whales. The smoothed plot

revealed a unimodal non-linear trend, peaking in the winter of 2013. For encounter rates for mother-calf groups, again year performed best, explaining 64.6% of the deviance, with a similar uni-modal trend over this period. Notably, season performed better in encounter rates for all individuals vs. mother-calf groups (28.1 vs. 7.5%), inferring that individual encounter rates potentially fluctuated more over the season in comparison to mother-calf encounter rates. This supports the general trend evident in the data, whereby encounter rates for adult groups were notably low during late seasons from 2014 onwards.

Table 4: Generalized additive models describing temporal trends in encounter rates for humpback whales in the Au'Au Channel, Maui, Hawaii

Figure 4: Non-linear trends in encounter rates for a) all individual whales and b) mother-calf groups during transect surveys of the Au'Au Channel, Maui, between 2008 and 2018

Oceanographic conditions

Variation in local wintertime sea surface temperatures

Between 2008 and 2018, local wintertime (January to March) SSTs for the Au'Au Channel ranged from a minimum of 22.7 °C during March 2009 to a maximum of 25.5°C in March 2017. There was no evidence of any linear trend (see figure 5; Pearson's $CC = 0.540$, $p = 0.087$), however, when data were divided into years around the peak sighting year (2013), and SST's from 2008 to 2013 were compared to SST's from 2014 to 2018, the difference between these two groups was significant (using a two sample t-test; $t = -2.952$, D.F. = 9, $p = 0.016$), with slightly higher mean temperatures in 2014 to 2018, compared to 2008 to 2013 (+0.6°C).

Figure 5: Mean composite sea surface temperatures for the winter season (January to March) for the Au'Au Channel, 2008-2018.

Variations in system level climate anomalies

Trends in the three indices selected to represent the occurrence and strength of ocean anomalies in the North Pacific over the course of the study (the PDO, ONI and the TNI) indicated pronounced climatic changes during this time. At the beginning of the study period, the PDO index was in a strong negative phase. It diminished slightly between September 2009 and May 2010 then returned to a pronounced negative condition until the beginning of 2014, when it flipped to a positive integer. Positive PDO values were then persistently high throughout 2014, 2015 and 2016, but have fallen in the last two years. The most recent El Niño was signalled in the ONI index, which displayed a pronounced peak in 2015. The TNI index, which predicts Central Pacific El Niño events, saw monthly values exceeding -2 during 2009 and 2010, and again in

Rachel 12/20/2018 12:05 PM

Deleted: Mid-channel surveys in 2018 - Table 5: Encounter rates for line surveys conducted in the mid-channel waters of the Au'Au Channel, Maui, 2018 -

Rachel 12/20/2018 12:06 PM

Deleted: Table 5: Encounter rates for line surveys conducted in the mid-channel waters of the Au'Au Channel, Maui, 2018 -

Rachel 12/20/2018 12:07 PM

Deleted: using the onset of climate anomalies as a splitting point

Rachel 12/20/2018 12:08 PM

Deleted: 4

Rachel 12/21/2018 9:19 AM

Deleted: when

Rachel 12/20/2018 12:08 PM

Deleted: pre-2014

Rachel 12/20/2018 12:08 PM

Deleted: (

Rachel 12/20/2018 12:08 PM

Deleted:)

Rachel 12/20/2018 12:08 PM

Deleted: post 2014

Rachel 12/20/2018 12:08 PM

Deleted: (

Rachel 12/20/2018 12:09 PM

Deleted:)

Rachel 12/20/2018 12:09 PM

Deleted: in the more recent time period

more recent years, beginning in the summer of 2013. This index typically changes prior to the onset of the broader El Niño event [57] (see figure 6).

Figure 6: Mean annual values for key climate indices that influence oceanographic conditions in the North Pacific, between 2008 and 2018.

Modelling results

Results from the GAMs constructed to detect interplay between oceanic indicators and encounter rates indicated slightly different inferences for all individuals for all individuals including mother-calf pairs vs. mother-calf groups. Inferences also varied notably between mother-calf and adult only groups.

Notably, in all models, the local mean SST performed poorly, explaining between 0 and 21% of the variability, and inferring that the encounter rates of any of the specific groups were not influenced by variation in the local SST. When oceanic indicators were advanced by 1 year, for encounter rates for individual whales, the model containing the PDO index performed the best, explaining 27% of the variability, however the ENSO indices (ONI and TNI) performed similarly, explaining 21.3 and 23.4% of the deviance respectively, while the local mean SST performed poorly. For encounter rates for MC groups, the PDO index performed the best, explaining 35.5% of the variability, the TNI index also performed well, explaining 34.5% of the deviance, but had a less persuasive AICc value, while the ONI index performed poorly. Notably, plots for the PDO smoothers for individual encounter rates and for mother-calf encounter rates approached an inverse, linear relationship (see figure 7).

When oceanic indicators were advanced by two seasons (so 2008 encounter rates were tied to 2006 composite mean summertime values and so forth), for mother-calf groups, the performance of the models improved considerably: Variability explained by the PDO increased to 66.8%. Likewise the models for the TNI and ONI indices also improved to 53.2% and 46.2% respectively, however plots were no longer linear, potentially suggesting a more complex association or possibly an interaction between the indices used. Unfortunately, the sample size in this study precludes testing these interactions. Encounter rates for all individual whales reflected the trends described for mother-calf pair. In contrast though, for adult-only groups, neither the 1 year and 2 year advanced indices performed well, with deviance explaining ranging from 0 (ONI) to 27% (TNI advanced by 2 years) (see table 5). Potentially, this suggests that trends in encounter rates for adult only groups are not closely aligned with individual oceanic indices.

Table 5: Generalized additive models describing the influence of oceanic climate conditions in the North Pacific on encounter rates for humpback whales in the Au’Au Channel, Maui, Hawaii

Figure 7: Smoothed

Rachel 1/1/2019 12:42 PM
Deleted: f

Rachel 1/1/2019 12:42 PM
Deleted: while

Rachel 12/21/2018 9:21 AM
Deleted: . T

Rachel 12/21/2018 9:21 AM
Deleted: local

Rachel 1/5/2019 8:15 PM
Deleted: 0 .

Rachel 1/5/2019 8:16 PM
Deleted: 6

Rachel 1/5/2019 8:16 PM
Deleted: .

trends in encounter rates as PDO index varies for a) all individual whales and b) mother-calf groups during transect surveys of the Au'au Channel, Maui, between 2008 and 2018

5. Discussion

This study describes fine scale patterns of habitat use and changes in encounter rates for humpback whales in the waters of the Au'au Channel, West Maui over the last 11 years. This region comprises the favored breeding area for the newly-designated Hawaiian Distinct Population Segment [44]. Over the course of the study, while fine scale changes in patterns of habitat use within the study area were minimal, we documented pronounced non-linear fluctuations in encounter rates. Essentially, between 2008 and 2013 encounter rates for individual whales increased, potentially surpassing projected growth estimates for this population segment (5 to 6.6% per year; [45]). Individual encounter rates peaked in 2013, and then declined between 2013 and 2018. However, trends in these declines varied between different social groups. Encounter rates for adult groups dropped between 2013 and 2014, but then remained relatively consistent between 2014 and 2018. For mother-calf groups after 2013, encounter rates declined year-by-year through the end of the study period. Comparing mean encounter rates for two year brackets (2013-2014 vs. 2017-2018), encounter rates for mother-calf groups fell by 79.4% and looking at changes between individual years, between 2013 and 2018, rates fell by 92.5% (encounter rates in 2013; 0.40 MC groups km⁻¹, vs. 2018; 0.03 MC groups km⁻¹). These observations stand in stark contrast to recent reports of humpback whale baby booms documented elsewhere [66].

Is the reproductive rate for the Hawaii DPS in decline?

The changes in encounter rates for mother-calf groups documented here potentially infer that the reproductive rate for the Hawaii DPS may be in decline. However before reaching this conclusion, several key issues and questions need to be addressed. A pressing issue is to confirm that the current study was staged in an area where encounter rates for mother-calf pairs would accurately represent regional trends. The area where the study was conducted, namely the eastern portion of the Au'au Channel between the islands of Maui and Lanai, is currently perceived to be core mother-calf habitat within Hawaiian waters [46–49,67]. The most recent island-wide aerial surveys, though somewhat outdated, suggest that approximately 85% of mother-calf groups in the islands are found in the Au'au Channel [46,54] and females exhibit a preference for this portion of the Au'au Channel in the years when they calve [54]. Beyond this focal area, alternate studies report trends in mother-calf encounter rates that match the findings of this study: For example, a recent study that included the shorelines of Central and South Maui spanning the years 2014 to 2017, reported that proportions of mother-calf groups sighted fell from 54% of all group sightings in 2014 to 37% in 2017 [68]. By comparison, in this study the proportion of mother-calf groups sighted fell from 62% of all groups in 2014 to 41% in 2017. The slight differences likely represent differences in coverage, as Currie et al 2018 [68] included

a higher proportion of deeper, offshore waters in their study area, in comparison to this study. Moreover, looking across the island chain to the coastal waters along the Kohala coastline of the island of Hawaii, a region that is also recognized as favored mother-calf habitat [69], a shore-based long-term study of sightings and occupancy rates in this area reports declines in the range of 75% for sightings of mother-calf pairs over the last five years (Hawaii Marine Mammal Consortium, S. Yin, personal communication).

Rachel 1/1/2019 12:50 PM
Deleted: Looking across the island chain to ... [2]

Potentially, a localized shift in habitat use could account for these changes. In this study, preferred depths for those mother-calf pairs sighted within the study area remained consistent between 2008 and 2018, the study was based in near shore waters. However, and in the 2018 season, when encounter rates in the main study area were exceptionally low, ad-hoc surveys conducted in mid channel waters did not provide any evidence of higher than expected numbers of mother-calf pairs in this adjacent area (see supplemental table S4). Likewise, in the other regional studies referenced ([68], unpublished HMMC study), included more extensive offshore / deeper waters within their focal study areas, neither study reports any indication of changing patterns of localized habitat use.

Rachel 1/1/2019 12:53 PM
Deleted: Alternatively...otentially, a localiz ... [3]

A more large scale relocation of mother-calf pairs within the island chain offers a further plausible explanation for changes in encounter rates in nearshore waters around Maui and the island of Hawaii. The Penguin Banks, a shallow, but remote offshore bank that lies approximately 40 km south of the Island of Molokai, is recognized as an alternate area that is favored by mother-calf pairs [46]. Unfortunately, reflecting the remote location and typically rough surface conditions, this region remains un-surveyed in recent years. Further afield still, within the North-western Hawaiian Island (NWHI) chain, recent acoustic studies have detected the presence of adult humpback whales [70]. Notably, as mother-calf pairs are acoustically cryptic [71] they would likely go undetected during acoustic monitoring. Previous researchers have suggested that protected waters within the NWHI chain were used as breeding grounds during much earlier periods (>300 years ago) when ocean temperatures may have been warmer [72]. Extracted satellite data compiled in this study indicates that water temperatures within the Au'au Channel (22.7-25.5°C) currently remain well within the range used by humpback whales in breeding regions (21-28°C; [73]). Notwithstanding, further effort to extend current studies into these more remote areas is certainly warranted.

Rachel 1/1/2019 2:02 PM
Deleted: also ...ffers a ... further plausible p ... [4]

At this point, while a pronounced habitat shift in mother-calf pairs within Hawaiian waters remains a possibility, recent sighting data from the high latitude feeding grounds used by the Hawaii DPS supports the assumption that a decline in the reproductive rate of this population segment is ongoing. Feeding grounds used by the Hawaii DPS include coastal regions of Northern British Columbia, South-east Alaska, Prince William Sound, coastal regions of the Northern Gulf of Alaska and portions of South-west Alaska [45]. Collectively > 80% of the Hawaii DPS head to these areas and typically whales display a high degree of site fidelity between years [45,74] Across these areas, sightings of mother-calf pairs have declined steeply over the last five years. In Glacier Bay, South East Alaska, sighting data extend back more than 30 years [74] and

Rachel 1/1/2019 1:31 PM
Deleted: [5]

falling reproductive rates have recently been documented. Declines began in 2014; in 2016, park observers reported the lowest crude birth-rate since studies began in 1986 and in 2017, the crude birth rate was the second lowest, with only 2 calves sighted through the entire summer [75]. Preliminary results from a survey covering a broader area of South-east Alaska during the summer of 2017 documented only 3 calf sightings (SPLISH surveys; cited by [75]) and further north in Prince William Sound, South-west Alaska, no calves have been sighted since 2015 (O. Vonziogesar; personal communication).

Looking ahead, on-going long term studies focused in high latitude regions where high site fidelity allows for the reproductive rates of individual female whales to be documented (e.g. Glacier Bay) will provide valuable insight as these trends play out in future years. Within Hawaiian waters, more extensive studies that reach beyond the protected waters of the Au'au Channel and other shoreline areas are necessary before an unequivocal conclusion can be reached. In the meantime though, the consistent reports of low rates of mother-calf sightings across both breeding and feeding regions certainly suggest that the reproductive rate of the Hawaii DPS is currently in decline.

Did this population segment reach carrying capacity?

The term carrying capacity is used in many different contexts across the biological literature [76]. In the context of this study, the term carrying capacity refers to the size of a population when it reaches equilibrium. At this point, the biotic potential of the population in terms of growth is checked by environmental resistance and the numbers within the population remain stable [77]. Current studies demonstrate that, in relation to this definition, carrying capacity is not fixed or static. Rather, carrying capacity varies both spatially and temporally, as habitat quality or the availability of the key resources fluctuate over time (e.g. [78–81]). Establishing carrying capacity is typically challenging, but as the impacts of surpassing carrying capacity are generally detrimental, both for the specific population and the habitat that supports the population (see Chapman [76]), determining the status of a population with respect to its carrying capacity can be a useful step towards interpreting changes in the vital rates of a specific population and developing appropriate management plans.

In large-bodied mammals with long life-spans, density dependent changes in vital rates typically occur as populations approach carrying capacity. Reproductive rates may slow down and then decline rapidly once carrying capacity has been reached [82]. To some degree, this describes the overall trends seen in this study: Beginning in 2008, sighting rates were in a period of rapid increase. They reached a peak in 2013 and this peak was followed by a period of steep decline. It should be noted that surveys were not conducted in the two years prior to the peak recorded here (2011 and 2012), therefore it is possible that the true peak could have been slightly earlier than this date. Notwithstanding, if this population segment reached, or even

Rachel 1/1/2019 1:33 PM

Deleted: Furthermore,

Rachel 1/1/2019 1:33 PM

Deleted: p

Rachel 12/21/2018 6:47 PM

Deleted: At the time of writing, sighting data for 2018 were not yet available.

Rachel 12/21/2018 6:47 PM

Deleted: very

Rachel 1/1/2019 2:09 PM

Deleted: will provide a more comprehensive picture and

Rachel 1/1/2019 2:09 PM

Deleted: probably

Rachel 1/1/2019 2:09 PM

Deleted: cy in

Rachel 1/1/2019 2:10 PM

Deleted: s

Rachel 1/2/2019 11:31 AM

Moved (insertion) [1]

Rachel 1/2/2019 11:58 AM

Deleted: Theoretically, i

Rachel 1/2/2019 11:58 AM

Deleted: traits such as reproductive

Rachel 1/2/2019 11:31 AM

Deleted: accelerate

Rachel 1/2/2019 11:32 AM

Deleted: , and

Rachel 1/3/2019 9:04 AM

Deleted: this point

Rachel 1/3/2019 9:05 AM

Deleted: has been passed

Rachel 1/2/2019 11:33 AM

Deleted: -

Rachel 1/2/2019 12:00 PM

Deleted: This

surpassed its carrying capacity during this period, then this provides a robust explanation that would account for non-linear trends in sighting rates in mother-calf groups on the breeding grounds, and changes in reproductive rate of this DPS.

However a variety of observations suggest that the explanation may lie elsewhere. Firstly, rates of change in reproductive rates around carrying capacity are forecast to be approximately symmetrical (see Fig 2(a), Fowler [82]). In this case, changes were asymmetrical; the rate of decline fluctuated, and ultimately far exceeded the rate of increase prior to the peak encounter rates recorded in 2013. Furthermore, Eberhardt and Siniff [83] describe a range of additional indicators that should be apparent if a population is approaching carrying capacity; these include increased aggression between mating-age adults, increased rates of offspring mortality and signs of poor body condition, disease and malnutrition in the general population [83]. Leading up to 2013, there were no reports of any increase in sightings of malnourished whales. Similarly, there was no spike in reported calf mortalities in the region, nor any reports of increased male aggression. If this population segment overshot carrying capacity prior to 2013, this could explain the steep decline in the reproductive rate, but in this scenario too, signs of the population approaching carrying capacity should still have been apparent prior to 2013.

Contrastingly, in recent months the first reports of poor condition in this sub-population have emerged. On their feeding grounds, an increasing number of humpback whales sighted are exhibiting signs of malnutrition and high rates of parasitism [75,84]. Reports from Alaska waters also indicate increasing numbers of whales persisting for longer periods on the feeding grounds prior to migration, and therefore potentially engaged in searching for food over longer periods [84] and in Hawaii waters, findings from the current study indicate a reduction in female reproductive rates. All three of these observations comprise declining condition indices, as described by Eberhardt and Siniff [83]. Based on the decision matrix laid out by [85] this combination of declining condition indicators alongside declining abundance indicators (numbers of offspring produced or encounter rates per kilometre), indicate a decline in carrying capacity (i.e. a reduction in size of the population that can be supported within the current habitat used by the population segment). Causes for such declines may vary widely. In some instances, changes can be specifically related to the availability of a single resource (e.g. [80]). In this case, given the increased reports of poor condition and malnutrition, nutritional stress emerges as a possible explanation for the current changes in evidence within this DPS. Supporting this conclusion, wide range of unusual mortality and mass casualty events in unrelated fauna has been reported from across the Gulf of Alaska and connected marine systems and linked to reduced productivity and prey resources in this region [26,29,30]. Finally, if the reproductive decline detected here was solely related to surpassing carrying capacity, the impact should be limited to the focal species, humpback whales and directly related prey populations. In fact, a wide range of unusual mortality and mass

Rachel 12/21/2018 6:51 PM
Deleted: as well as

Rachel 1/2/2019 11:31 AM
Moved up [1]: Theoretically, in large-bodied mammals with long life-spans, density dependent changes in traits such as reproductive rates typically accelerate as populations approach carrying capacity, and then decline rapidly once this point has been passed [73].

Rachel 1/2/2019 11:33 AM
Deleted: Certainly this describes the overall trends seen here

Rachel 1/2/2019 11:35 AM
Deleted: h

Rachel 12/20/2018 12:23 PM
Deleted: carrying capacity (K).

Rachel 1/2/2019 12:01 PM
Deleted: k

Rachel 1/2/2019 11:36 AM
Deleted: In recent months the first reports of malnourished whales have emerged [73] however

Rachel 1/2/2019 12:02 PM
Deleted: ment leading up to 2013

Rachel 1/5/2019 8:24 PM
Deleted: [86]

casualty events in unrelated fauna has been reported from across the Gulf of Alaska and connected regions [26,29,30]cc.

Fluctuations in prey resources

Beginning in the winter of 2013, oceanic conditions within the North Pacific entered a period of unusual flux that has potentially impacted multiple facets within the marine system, including the prey resources that support North Pacific humpback whales. Chronologically, the PDO began its shift in late 2013 and flipped to a positive phase in early 2014. During 2014, the TNI index, indicating the evolution of a Central Pacific El Niño event [55], reached a magnitude > 2, signalling the build-up of warm water in the Central North Pacific and concurrently, the warm water anomaly in the North Pacific (known as “the blob”) spread into the coastal domains of Alaska and British Columbia [25]. In concert, all three of these anomalies would suppress coastal upwellings, draw down coastal productivity [28] and lead to pronounced horizontal advection of key prey species [25,26].

From a dietary perspective, humpback whales display a degree of diet plasticity that may potentially provide some resilience to large-scale oceanic shifts and associated changes in prey availability. In the California Current System, humpback whales are known to switch between a predominantly fish based diet and a predominantly krill based diet, as changes in the North Pacific Gyre Oscillation (NPGO) impact SSTs and alter upwellings that drive relative prey availability along the California-Oregon seaboard [86,87]. In Alaska waters, prey preferences vary seasonally and regionally [84,88–91]. During the summer months, some regions of South-east Alaska, mature euphausiids (*Thysanoessa sp.*, *Euphausia pacifica*) comprise a preferred prey [88,92], while in other areas, such as Icy Strait and Glacier Bay, small schooling fish such as capelin (*Mallotus villosus*), juvenile wall-eye Pollock (*Gadus chalcogrammus*) and Pacific herring (*Clupea pallasii*) are primary prey targets. During fall, targeted prey vary regionally between herring and euphausiids, and by winter, those humpback whales present feed exclusively on herring (Straley). In South-west Alaska (Kodiak Island), local humpbacks feed exclusively on fish, but will bypass juvenile pollock, *Gadus chalcogrammus*, in favor of schooling capelin, *Mallotus villosus* [91].

Although in theory, this varied diet some provide some degree of resilience in Alaska humpbacks during periods when prey resources are disrupted, the climate anomalies that began in late 2013 have been causally linked to reduced availability of across the range of these preferred prey. In coastal regions in Alaska, the increase in freshwater runoff has been linked to declines in herring [93]. In South-east Alaska, the slowing of coastal upwellings has led to changes in both quality and quantity of euphausiid prey [94], while in the open

Rachel 1/2/2019 12:17 PM

Deleted: . Recent reviews of whaling data clearly demonstrate that, in fact, most modern populations of large whales currently exist well below previous carrying capacities [76] and at this point, there appears to be at least no clear evidence, beyond the change in reproductive rate reported here, that would lend support to the conclusion that this population segment reached or surpassed its carrying capacity in the years leading up to or around 2013.

Rachel 1/2/2019 11:40 AM

Deleted: .

Rachel 1/2/2019 12:41 PM

Deleted: P

Rachel 1/3/2019 9:35 AM

Deleted: availability

Rachel 1/2/2019 12:41 PM

Deleted: and reproductive rates.

Rachel 1/2/2019 12:41 PM

Deleted: nutritional stress within this population segment provides an alternate explanation that could account for declining reproductive rates and reduced numbers of adult whales currently seen in Maui waters could be the recent emergence ... [7]

Rachel 1/3/2019 10:14 AM

Deleted: in some areas

Rachel 12/20/2018 12:26 PM

Deleted: will

Rachel 1/3/2019 10:15 AM

Deleted: humpbacks are known to feed on ... [8]

Rachel 1/1/2019 1:41 PM

Deleted: however clearly defined regional ... [9]

Rachel 12/21/2018 6:49 AM

Deleted: have been reported and these cc ... [10]

Rachel 12/21/2018 6:52 AM

Deleted: Specifically

Rachel 12/21/2018 6:52 AM

Deleted: ,

Rachel 12/21/2018 6:52 AM

Deleted: i

Rachel 12/21/2018 6:51 AM

Deleted: euphausiids

Rachel 12/21/2018 6:53 AM

Deleted: the

Rachel 12/21/2018 6:51 AM

Deleted: (*Thysanoessa sp.*, *Euphausia pacifica*)

Rachel 12/21/2018 6:52 AM

Deleted:)

Rachel 12/21/2018 7:02 AM

Deleted: while

Rachel 12/21/2018 7:02 AM

Deleted: i

Rachel 1/4/2019 7:03 PM

Deleted: T

Rachel 1/3/2019 10:19 AM

Deleted: the summer of 2014

Rachel 1/2/2019 1:13 PM

Deleted: both

waters of the Gulf, horizontal advection has disrupted regional assemblages of planktonic fauna. Specifically, smaller, southern species of copepods have replaced the higher quality, larger species typically seen in the Gulf [22,95]. As copepods form the cornerstone of pelagic food webs in the Gulf, numerous higher trophic levels have been impacted, including planktivorous fish populations[22,95]

In female mysticetes, as adequate energy stores are a requirement for both the initiation and successful maintenance of reproduction [33,35,37] reductions in availability of preferred prey and associated nutritional stress are mechanistically linked to a depression in reproductive rates. In the results obtained in this study, the performance of the GAM models suggested that some degree of linkage exists between reproductive rates observed in Hawaiian waters and conditions in feeding regions, as indicated by the oceanic indices included in the analysis. Notably, the performance of the models improved when the oceanic indices were advanced by two seasons compared to advancing the indices by a single season, suggesting that the suppression of ovulation in response to reduced prey availability in the year prior to pregnancy may be the key mechanism at play here. A suppression in ovulation rates would likely reduce the gamut of reproductive activities for female humpback whales, including the frequency of migration to the breeding grounds. Within breeding areas, females without young calves quickly leave the breeding grounds [96] and in the later portions of the season adult males typically seek mating opportunities with maternal females that may go into post-partum ovulation [97]. In periods of nutritional stress, it is easy to foresee that this would become a rare event. Reflecting these reduced mating opportunities, males would likely depart earlier for the feeding grounds. If so, this could explain the especially low numbers of adults seen in Maui waters during late seasons in recent years.

At this point, a final explanation for the declining sighting rates for mother-calf humpback whale pairs in Hawaiian waters requires a broad swath of further research, as many questions remain unresolved. As advocated by Doney et al. [2], a more definitive answer addressing the causes of the changes seen will require the availability of biological datasets that document fluctuations in population dynamics and key life history traits over the long term time scale. Potentially, such details will facilitate a much clearer understanding of these complex interactions, and allow for improved accuracy in predicting future fluctuations within this and other distinct population segments of humpback whales.

6. Conclusion

Beginning in the winter of 2014, the combination of recent climatic anomalies in the Eastern North Pacific, namely the regime shift of the PDO to a positive phase, the development and subsequent migration of the warm water anomaly known as the blob, and the evolution of a Central Pacific El Niño event combined to create oceanic conditions along the Alaska coastline that reduced productivity and impacted key humpback

Rachel 1/4/2019 7:04 PM

Deleted: , such as herring and capelin

Rachel 1/3/2019 10:19 AM

Deleted: Recent oceanographic studies have established that north of 38°N, the PDO is the central driving force in productivity and zooplankton levels, dictating upwelling variance and impacting the strength of alongshore currents that lead to horizontal advection [77]. Notably, Newman et al. [23] also determined that trends seen in the general ENSO (ONI) index do not appear to be closely linked to changes in productivity and zooplankton levels in coastal regions of the North Pacific. Rather, these authors propose that the interplay between the PDO and the NPGO provides a coherent framework, referred to as the horizontal-advection bottom-up forcing hypothesis, that explains the fluctuations in planktonic assemblages and associated fish stocks around the North Pacific [77]. Relating these findings to the current study, the NPGO index essentially captures the decadal expressions of the Central Pacific El Niño. Consequently this is closely related to the TNI index used in this study [55]. In our models, the variations in the general ENSO (ONI) index did not predict the changes in reproductive rates observed. In terms of predicting fluctuations in the reproductive rate, the best models in this study included the PDO index, which explained 35% of the deviance in reproductive rates, and the TNI index, which explained 34% of the deviance in the reproductive rate. Potentially, in their principal feeding grounds along the coastline of the North Pacific and the Gulf of Alaska, these changing patterns in the PDO and the TNI indices would lead to reduced productivity and horizontal advection of key prey that could account for nutritional s... [11]

Rachel 12/21/2018 7:02 PM

Formatted: Font:Italic

Rachel 1/3/2019 12:16 PM

Formatted: Font:Italic, Not Highlight

Rachel 12/21/2018 7:02 PM

Formatted: Font:Italic

Rachel 12/20/2018 6:48 PM

Deleted: adiposity

Rachel 12/21/2018 7:13 PM

Deleted:

Rachel 12/21/2018 7:20 PM

Deleted: : Females are known to forego migration in resting years between reproductive cycle... [12]

Rachel 12/21/2018 7:16 PM

Deleted: w

Rachel 1/1/2019 1:52 PM

Deleted: As ovulation requires a degree of adiposity

Rachel 1/1/2019 1:52 PM

Deleted: i

Rachel 1/4/2019 7:06 PM

Deleted: ; this could certainly

Rachel 1/4/2019 7:06 PM

Deleted:

Rachel 1/3/2019 10:38 AM

Deleted:

Rachel 1/3/2019 10:39 AM

Deleted: summer of

whale prey resources [17,22,28]. Highlighting the mechanistic links that exist between climate anomalies, physical oceanographic processes and biological resources, these climate anomalies currently provide a plausible explanation for the recent decline in reproductive rates within the Hawaii DPS documented in this study.

Understanding the interactions and connectivity between different climate anomalies is certainly complex. Consequently, we would encourage experts in this field to review the data presented here, as this could provide a more nuanced analysis of the connections between fluctuating humpback whale reproductive rates within local population segments, regional prey availability and recent marine regime shifts. Future studies that document health indicators, such as body condition, especially in female humpback whales, will provide valuable insight into the resilience of specific population segments, such as the Hawaii DPS, as they negotiate the challenges presented as marine systems shift towards a new climate regime. A better understanding of anthropogenic energetic stressors, such as whale-watching in both feeding and breeding regions, could elucidate means by which ameliorative management practices could help minimize maternal stress during this transition.

Humpback whales are widely recognized as a sentinel species [39,60]. Changes in a key life history trait such as the reproductive rate provide an easily detected signal of other more subtle changes, such as alterations in prey quality, which in turn may ultimately carry much broader impact. Climatic models have confirmed that anthropogenic climate change exacerbated the formation of the recent warm water anomaly in the North Pacific and the associated warming of the Gulf of Alaska [29]. Events like this provide insight into the functioning and challenges of a warmer ocean and lessons learnt during events such as this may prove to be exceptionally valuable in the years that lie ahead.

In the meantime, the Hawaii DPS of humpback whales carries considerable ecosystem capital, not only in terms of ecosystem services [98,99], but also more directly, supporting extensive whale watching and associated tourism activities in both Alaska and Hawaii. For the purposes of down listing, this species was divided into separate stocks. NMFS established that the health of one stock should be considered independently from other groups [44]. Following this line of reasoning, as the current changes described here meet the trigger requirements laid out in the post delisting monitoring plan [100], we suggest that increased protection and a review of the status of this stock is clearly justified at this time.

Acknowledgments

This project could not have been completely without the generous support and endless encouragement provided by Mark Pervical, Worcester UK. Additionally, logistic support received from Lee James and the

captains and crew of Ultimate Whalewatch, Maui, HI made a very valuable contribution to the completion of this work. Additional support during fieldwork was provided by Terence Mangold, Mark Danielson, Sergio Burgos, Rob Schneider, Sarah Scrivano, Michaela Miller, Tad Chamberlin, Nicola Dempster and Charlotte Dempster. We are grateful to the many other volunteers who have assisted the Keiki Kohola project over the years, and sincerely appreciate the keen and enthusiastic participation of the students of California State University Channel Islands. Additionally, we sincerely appreciate the input and feedback provided by Joe Mobley and Jan Straley during their reviews of a previous version of this manuscript.

Ethical Statement

The study was conducted under NMFS scientific research permit # 10018 and #17845, and under associated Hawaii State permits. Full details of the precise research protocols used in this study were carefully reviewed by the Office of Protected Species, prior to issuance of the above research permit to ensure that these methods comply with all ethical guidelines. Inherent in this review, is the requirement that every effort be made to minimize any impact on animals during research activities. As this detailed and extensive review had been conducted by experts in this field, further ethical review by the co-operating institution, California State University Channel Islands was not required. All research protocols additionally comply with the Endangered Species Act (1973) and the Marine Mammal Protection Act (1972) (https://apps.nmfs.noaa.gov/docs_cfm/laws_and_regulations.cfm).

Funding Statement

This project was funded through donations to the Keiki Kohola Project, principally received from Mark Pervical, Worcester UK. Donations from the Instructionally Related Activities fund at California State University Channel Islands supported student participation in the project. Cesere Fine Art photography provided additional financial support; funds were also received from Apple Computers Inc and Whale Trust, Maui.

Data Accessibility

The datasets supporting this study are provided in the supplementary materials, accompanying this publication.

Competing Interests

We have no competing interests

Authors' Contributions

R.C. designed the project, participated in data collection, conducted the statistical analysis and wrote the manuscript. A.V. designed the project, led fieldwork and participated in data analysis. V.H. conducted

R. Soc. open sci.

extensive fieldwork and participated in data analysis. C.W. led student teams in the field, conducted fieldwork and assisted in manuscript preparation. J.C. and D.C. were extensively involved in fieldwork, assisting in data collection and photo-cataloguing. All authors assisted in editing the manuscript and approved the final version prior to publication.

1. World Meteorological Organization. 2018 *WMO statement on the status of the global climate in 2017*. (doi:978-92-63-11212-5)
2. Doney SC *et al.* 2012 Climate Change Impacts on Marine Ecosystems. *Ann. Rev. Mar. Sci.* **4**, 11–37. (doi:10.1146/annurev-marine-041911-111611)
3. Bindoff NL, Willebrand J, Artale V CA. 2007 Observations: oceanic climate change and sea level. In *Climate Change 2007: The Physical Science Basis: Contribution of Working Group I to the Fourth Assessment Report of the Intergovernmental Panel on Climate Change* (eds DQ S Solomon, MM M Manning, Z Chen), pp. 385–432. Cambridge: Cambridge Univ. Press.
4. Doney SC, Fabry VJ, Feely RA, Kleyppas JA. 2009 Ocean Acidification: the Other Co2. *Annu. Rev.* , 212–251. (doi:10.1146/annurev.marine.010908.163834.)
5. Kroeker KJ, Kordas RL, Crim RN, Singh GG. 2010 Meta-analysis reveals negative yet variable effects of ocean acidification on marine organisms. *Ecol. Lett.* **13**, 1419–1434. (doi:10.1111/j.1461-0248.2010.01518.x)
6. DiLorenzo E, Miller AJ, Schneider N, McWilliams JC. 2005 The Warming of the California Current System: Dynamics and Ecosystem Implications. *J. Phys. Oceanogr.* **35**, 336–362. (doi:10.1175/JPO-2690.1)
7. Wood KR, Bond NA, Danielson SL, Overland JE, Salo SA, Whitefield J. 2015 A decade of environmental change in the Pacific Arctic region. *Prog. Oceanogr.* **136**, 12–31. (doi:10.1016/J.POCEAN.2015.05.005)
8. Kovacs KM, Lydersen C. 2008 Climate change impacts on seals and whales in the North Atlantic Arctic and adjacent shelf seas. *Sci. Prog.* **91**, 117–150. (doi:10.3184/003685008X324010)
9. MacCracken JG. 2012 Pacific Walrus and climate change: observations and predictions. *Ecol. Evol.* **2**, 2072–2090. (doi:10.1002/ece3.317)
10. Ainsworth TD, Heron SF, Ortiz JC, Mumby PJ, Grech A, Ogawa D, Eakin CM, Leggat W. 2016 Climate change disables coral bleaching protection on the Great Barrier Reef. *Science* **352**, 338–42. (doi:10.1126/science.aac7125)
11. Palacios DM, Baumgartner MF, Laidre KL, Gregr EJ. 2013 Beyond correlation: integrating environmentally and behaviourally mediated processes in models of marine mammal distributions. *Endanger. Species Res.* **22**, 191–203. (doi:10.3354/esr00558)
12. Laidre KL *et al.* 2015 Arctic marine mammal population status, sea ice habitat loss, and conservation recommendations for the 21st century. *Conserv. Biol.* **29**, 724–737. (doi:10.1111/cobi.12474)

13. Kaschner K, Tittensor DP, Ready J, Gerrodette T, Worm B. 2011 Current and Future Patterns of Global Marine Mammal Biodiversity. *PLoS One* **6**, e19653. (doi:10.1371/journal.pone.0019653)
14. Clucas G V. *et al.* 2015 A reversal of fortunes: climate change ‘winners’ and ‘losers’ in Antarctic Peninsula penguins. *Sci. Rep.* **4**, 5024. (doi:10.1038/srep05024)
15. Moore SE. 2016 Is it ‘boom times’ for baleen whales in the Pacific Arctic region? *Biol. Lett.* **12**, 20160251. (doi:10.1098/rsbl.2016.0251)
16. Rodgers KS, Jokiel PL, Brown EK, Hau S, Sparks R. 2015 Over a Decade of Change in Spatial and Temporal Dynamics of Hawaiian Coral Reef Communities. *Pacific Sci.* **69**, 1–13. (doi:10.2984/69.1.1)
17. DiLorenzo E, Mantua N. 2016 Multi-year persistence of the 2014/15 North Pacific marine heatwave. *Nat. Clim. Chang.* **6**, 1042–1047. (doi:10.1038/nclimate3082)
18. Ducklow HW, Doney SC, Steinberg DK. 2009 Contributions of Long-Term Research and Time-Series Observations to Marine Ecology and Biogeochemistry. *Ann. Rev. Mar. Sci.* **1**, 279–302. (doi:10.1146/annurev.marine.010908.163801)
19. Richter-Menge J, Overland JE, Mathis JT, Osborne E. 2017 Arctic Report Card 2017. Available at <https://arctic.noaa.gov/Report-Card/Report-Card-2017>
20. Hausfather Z, Cowtan K, Clarke DC, Jacobs P, Richardson M, Rohde R. 2017 Assessing recent warming using instrumentally homogeneous sea surface temperature records. *Sci. Adv.* **3**, e1601207. (doi:10.1126/sciadv.1601207)
21. Mantua NJ, Hare SR, Zhang Y, Wallace JM, Francis RC. 1997 Pacific interdecadal climate oscillation with impacts on salmon production. *Am. Meteorological Soc.* **78**, 1069–1079. (doi:10.1175/1520-0477(1997)078<1069:apicow>2.0.co;2)
22. Peterson W, Bond N, Press MR-P, 2016 undefined. In press. The Blob is gone but has morphed into a strongly positive PDO/SST pattern. *PICES Press* 24, 46–47,50. (doi:19.1111/gcb.13054.Peterson)
23. Newman M *et al.* 2016 The Pacific Decadal Oscillation, Revisited. *J. Clim.* **29**, 4399–4427. (doi:10.1175/JCLI-D-15-0508.1)
24. Mundy PR. 2005 *The Gulf of Alaska: Biology and Oceanography. Alaska Sea Grant College Program.* (doi:10.4027/gabo.2005)
25. Bond NA, Cronin MF, Freeland H, Mantua N. 2015 Causes and impacts of the 2014 warm anomaly in the NE Pacific. *Geophys. Res. Lett.* **42**, 3414–3420. (doi:10.1002/2015GL063306)
26. Cavole LM *et al.* 2016 Biological Impacts of the 2013–2015 Warm-Water Anomaly in the Northeast Pacific: Winners, Losers, and the Future. *Oceanography* **29**, 273–285. (doi:10.2307/24862690)
27. L’Heureux ML *et al.* 2017 Observing and predicting the 2015/16 El Niño. *Bull. Am. Meteorol. Soc.* **98**, 1363–1382. (doi:10.1175/BAMS-D-16-0009.1)
28. Whitney FA. 2015 Anomalous winter winds decrease 2014 transition zone productivity in the NE Pacific. *Geophys. Res. Lett.* **42**, 428–431. (doi:10.1002/2014GL062634.Received)
29. Walsh JE *et al.* 2018 The High Latitude Marine Heat Wave of 2016 and Its Impacts on Alaska. *Bull. Am.*

- Meteorol. Soc.* **99**, S39–S43. (doi:10.1175/BAMS-D-17-0105.1)
30. Jones T *et al.* 2018 Massive Mortality of a Planktivorous Seabird in Response to a Marine Heatwave. *Geophys. Res. Lett.* **45**, 3193–3202. (doi:10.1002/2017GL076164)
 31. Baker CS *et al.* 1986 Migratory movement and population structure of humpback whales (*Megaptera novaeangliae*) in the central and eastern North Pacific. *Mar. Ecol. Prog. Ser.* **7**, 105–119.
 32. Rice DW, Wolman AA. 1971 *The life history and ecology of the gray whale (Eschrichtius robustus)*. *American Society of Mammalogy, Special Publication 3: 1-21*. American Society of Mammalogists.
 33. Lockyer C. 1986 Body fat condition in Northeast Atlantic fin whales, *Balaenoptera physalus*, and its relationship with reproduction and food resource. *Can. J. Fish. Aquat. Sci.* **43**, 142–147. (doi:10.1139/f86-015)
 34. Miller CA, Best PB, Perryman WL, Baumgartner MF, Moore MJ. 2012 Body shape changes associated with reproductive status, nutritive condition and growth in right whales *Eubalaena glacialis* and *E. australis*. *Mar. Ecol. Prog. Ser.* **459**, 135–156. (doi:10.3354/meps09675)
 35. Zieba DA, Amstalden M, Williams GL. 2005 Regulatory roles of leptin in reproduction and metabolism: A comparative review. *Domest. Anim. Endocrinol.* **29**, 166–185. (doi:10.1016/j.domaniend.2005.02.019)
 36. Lockyer C. 1981 Growth and energy budgets of large baleen whales from the southern hemisphere. *FAO Fish. Serv.* **5**, 379–487.
 37. Christiansen F, Dujon AM, Sprogis KR, Arnould JPY, Bejder L. 2016 Noninvasive unmanned aerial vehicle provides estimates of the energetic cost of reproduction in humpback whales. *Ecosphere* **7**, 1–18.
 38. Perryman WL, Donahue MA, Perkins PC, Reilly SB. 2002 Are observed fluctuations related to changes in seasonal ice cover? *Mar. Mammal Sci.* **18**, 121–144.
 39. Moore S, Huntington H. 2008 Arctic Marine Mammals and Climate Change : Impacts and Resilience. *Ecol. Appl.* **18**, 157–165. (doi:10.1890/06-0571.1)
 40. Meyer-Gutbrod EL, Greene CH, Sullivan PJ, Pershing AJ. 2015 Climate-associated changes in prey availability drive reproductive dynamics of the North Atlantic right whale population. *Mar. Ecol. Prog. Ser.* **535**, 243–258. (doi:10.3354/meps11372)
 41. Meyer-Gutbrod EL, Greene CH. 2014 Climate-Associated Regime Shifts Drive Decadal-Scale Variability in Recovery of North Atlantic Right Whale Population. *Oceanography.* **27**, 148–153. (doi:10.2307/24862197)
 42. Meyer-Gutbrod EL, Greene CH. 2018 Uncertain recovery of the North Atlantic right whale in a changing ocean. *Glob. Chang. Biol.* **24**, 455–464. (doi:10.1111/gcb.13929)
 43. Leaper R, Cooke J, Trathan P, Reid K, Rowntree V, Payne R. 2006 Global climate drives southern right whale (*Eubalaena australis*) population dynamics. *Biol. Lett.* **2**, 289–292. (doi:10.1098/rsbl.2005.0431)
 44. National Oceanographic and Atmospheric Administration (NOAA). 2016 Endangered and Threatened

Species; Identification of 14 distinct population segments of the humpback whale (*Megaptera novaeangliae*) and revision of species-wide listing. *Fed. Regist.* 81, 62260–62320.

45. Calambokidis J *et al.* 2008 SPLASH: Structure of populations, levels of abundance and status of humpback whales in the North Pacific. *Final Rep. Contract AB133F-03-RP0078. Cascadia Res. Collect. Olympia, WA* , 57.
46. Mobley JR, Bauer G, Herman L. 1999 Changes over a ten-year interval in the distribution and relative abundance of humpback whales (*Megaptera novaeangliae*) wintering in Hawaiian waters. *Aquat. Mamm.* **25**, 63–72.
47. Cartwright R, Gillespie B, LaBonte K, Mangold T, Venema A, Eden K, Sullivan M. 2012 Between a rock and a hard place: Habitat selection in female-calf humpback whale (*Megaptera novaeangliae*) pairs on the hawaiian breeding grounds. *PLoS One* **7**, e38004. (doi:10.1371/journal.pone.0038004)
48. Craig AS, Herman LM, Waterman JO, Pack AA. 2014 Habitat segregation by female humpback whales in Hawaiian waters: avoidance of males? *Behaviour* **151**, 613–631. (doi:10.1163/1568539X-00003151)
49. Pack AA, Herman LM, Craig AS, Spitz SS, Waterman JO, Herman EYK, Deakos MH, Hakala S, Lowe C. 2017 Habitat preferences by individual humpback whale mothers in the Hawaiian breeding grounds vary with the age and size of their calves. *Anim. Behav.* **133**, 131–144. (doi:10.1016/J.ANBEHAV.2017.09.012)
50. Grigg R, Grossman E, Earle S, Gittings S, Lott D, McDonough J. 2002 Drowned reefs and antecedent karst topography, Au’au Channel, S.E. Hawaiian Islands. *Coral Reefs* **21**, 73–82. (doi:10.1007/s00338-001-0203-8)
51. Strindberg S, Buckland ST. 2004 Zigzag survey designs in line transect sampling. *J. Agric. Biol. Environ. Stat.* **9**, 443–461. (doi:10.1198/108571104X15601)
52. Glockner-Ferrari DA, Ferrari MJ. 1985 Individual Identification, behavior, reproduction, and distribution of humpback whales, *Megaptera novaeangliae*, in Hawaii. US Marine Mammal Commission; 1985.
53. Mobley Jr. JR, Herman LM. 1985 Transience of social affiliations among humpback whales (*Megaptera novaeangliae*) on the Hawaiian wintering grounds. *Can. J. Zool.* **63**, 762–772. (doi:10.1139/z85-111)
54. Mobley J, Spitz S, Grotefendt R. 2001 Abundance of humpback whales in Hawaiian waters: Results of 1993-2000 aerial surveys. Report to the Hawaiian Islands Humpback Whale National Marine Sanctuary, 9, (2001).
55. Barlow J, Forney KA, Barlow J. 2007 Abundance and population density of cetaceans in the California Current ecosystem. *Fish. Bull.* **105**, 509–526.
56. Katona S, Baxter B, Brazier O, Kraus S, Perkins J, Whitehead H. 1979 Identification of Humpback Whales by Fluke Photographs. In *Behavior of Marine Animals*, pp. 33–44. Boston, MA: Springer US. (doi:10.1007/978-1-4684-2985-5_2)
57. Trenberth KE, Stepaniak DP, Trenberth KE, Stepaniak DP. 2001 Indices of El Niño Evolution. *J. Clim.* **14**,

- 1697–1701. (doi:10.1175/1520-0442(2001)014<1697:LIOENO>2.0.CO;2)
58. Zou Y. 2015 The Impacts of the Eastern Pacific and Central Pacific El Niño on North American Winter Climate. Doctoral dissertation, UC Irvine.
 59. Seyboth E, Groch KR, Dalla Rosa L, Reid K, Flores PAC, Secchi ER. 2016 Southern Right Whale (*Eubalaena australis*) Reproductive Success is Influenced by Krill (*Euphausia superba*) Density and Climate. *Sci. Rep.* **6**, 1–8. (doi:10.1038/srep28205)
 60. Bengtson Nash SM *et al.* 2018 Signals from the south; humpback whales carry messages of Antarctic sea-ice ecosystem variability. *Glob. Chang. Biol.* **24**, 1500–1510. (doi:10.1111/gcb.14035)
 61. The R foundation for statistical computing. 2006 R version 3.4.0.
 62. Young RL, Weinberg J, Vieira V, Ozonoff A, Webster TF. 2011 Generalized Additive Models and Inflated Type I Error Rates of Smoother Significance Tests. *Comput. Stat. Data Anal.* **55**, 366–374. (doi:10.1016/j.csda.2010.05.004)
 63. Hastie TJ, Tibshirani R. 1990 *Generalized additive models, volume 43 of Monographs on Statistics and Applied Probability*. Chapman & Hall CRC.
 64. Wood S. 2006 mgcv 1.3. R package.
 65. Ruppert D. 2002 Selecting the Number of Knots For Penalized Splines. *J. Comput. Graph. Stat.* **4**, 735–757.
 66. Pallin LJ, Baker CS, Steel D, Kellar NM, Robbins J, Johnston DW, Nowacek DP, Read AJ, Friedlaender AS. 2018 High pregnancy rates in humpback whales (*Megaptera novaeangliae*) around the Western Antarctic Peninsula, evidence of a rapidly growing population. *R. Soc. Open Sci.* **5**, 180017. (doi:10.1098/rsos.180017)
 67. Craig AS, Herman LM. 2000 Habitat preferences of female humpback whales *Megaptera novaeangliae* in the Hawaiian Islands are associated with reproductive status. *Mar. Ecol. Prog. Ser.* **193**, 209–216.
 68. Currie J, Stack S. 2018 Utilizing Occupancy Models and Platforms-of- Opportunity to Assess Area Use of Mother-Calf Humpback Whales Utilizing Occupancy Models and Platforms-of-Opportunity to Assess Area Use of Mother-Calf Humpback Whales. *Open J. Mar. Sci.* , 276–292. (doi:10.4236/ojms.2018.82014)
 69. Smultea MA. 1994 Segregation by humpback whale (*Megaptera novaeangliae*) cows with a calf in coastal habitat near the island of Hawaii. *Can. J. Zool.* **72**, 805–811. (doi:10.1139/z94-109)
 70. Lammers MO, Fisher-Pool PI, Au WWL, Meyer CG, Wong KB, Brainard RE. In press. Humpback whale *Megaptera novaeangliae* song reveals wintering activity in the Northwestern Hawaiian Islands. *Mar. Ecol. Prog. Ser.* **423**, 261–268. (doi:10.2307/24874596)
 71. Videsen SKA, Bejder L, Johnson M, Madsen PT. 2017 High suckling rates and acoustic crypsis of humpback whale neonates maximise potential for mother-calf energy transfer. *Funct. Ecol.* **31**, 1561–1573. (doi:10.1111/1365-2435.12871)
 72. Herman LM. 1979 Humpback Whales in Hawaiian Waters: A Study in Historical Ecology. *Pacific Sci.* **33**,

1–16.

73. Rasmussen K, Palacios DM, Calambokidis J, Sabori MT, Rosa LD, Secchi ER, Steiger GH, Allen JM, Stone GS. 2007 Southern Hemisphere humpback whales wintering off Central America : insights from water temperature into the longest mammalian migration. *Biol. Lett.* , 302–305. (doi:10.1098/rsbl.2007.0067)
74. Gabriele CM, Neilson JL, Straley JM, Baker CS, Cedarleaf JA, Saracco JF. 2017 Natural history, population dynamics, and habitat use of humpback whales over 30 years on an Alaska feeding ground. *Ecosphere* **8**, 1–18. (doi:10.1002/ecs2.1641)
75. Nielson J, Gabriele CM, Taylor-Thomas LF. 2018 Humpback Whale Monitoring in Glacier Bay and Adjacent Waters 2017 Annual Progress Report. Natural Resource Report NPS/GLBA/NRR—2018/1660. National Park Service, Fort Collins, Colorado.
76. Chapman EJ, Byron CJ. 2018 The flexible application of carrying capacity in ecology. *Glob. Ecol. Conserv.* **13**, e00365. (doi:10.1016/J.GECCO.2017.E00365)
77. Odum, Eugene P. 1959 *Fundamentals of Ecology*. WB Saunders company.
78. Duarte A, Hatfield JS, Swannack TM, Forstner MRJ, Green MC, Weckerly FW. 2016 Simulating range-wide population and breeding habitat dynamics for an endangered woodland warbler in the face of uncertainty. *Ecol. Modell.* **320**, 52–61. (doi:10.1016/J.ECOLMODEL.2015.09.018)
79. Laidre KL, Jameson RJ, Demaster DP. 2001 An estimation of carrying capacity for sea otters along the california coast. *Mar. Mammal Sci.* **17**, 294–309. (doi:10.1111/j.1748-7692.2001.tb01272.x)
80. Braithwaite JE, Meeuwig JJ, Jenner KCS. 2012 Estimating Cetacean Carrying Capacity Based on Spacing Behaviour. *PLoS One* **7**, e51347. (doi:10.1371/journal.pone.0051347)
81. MacCracken JG, Lemons PR, Garlich-Miller JL, Snyder JA. 2014 An index of optimum sustainable population for the Pacific Walrus. *Ecol. Indic.* **43**, 36–43. (doi:10.1016/J.ECOLIND.2014.02.013)
82. Fowler CW. 1981 Density dependence as related to life history. *Ecology* **62**, 602–610.
83. Eberhardt LL, Siniff DB. 1977 Population Dynamics and Marine Mammal Management Policies. *J. Fish. Res. Board Canada* **34**, 183–190. (doi:10.1139/f77-028)
84. Straley JM, Moran JR, Boswell KM, Vollenweider JJ, Quinn II TJ, Witteveen BH, Rice SD. 2018 Seasonal presence and potential influence of humpback whales on wintering Pacific herring populations in the Gulf of Alaska. *Deep Sea Res. Part II Top. Stud. Oceanogr.* **147**, 173–186. (doi:10.1016/J.DSR2.2017.08.008)
85. Gerrodette T, DeMaster DP. 1990 Quantitative determination of optimum sustainable population level. *Mar. Mammal Sci.* **6**, 1–16. (doi:10.1111/j.1748-7692.1990.tb00221.x)
86. DiLorenzo E *et al.* 2013 Synthesis of Pacific Ocean Climate and Ecosystem Dynamics. *Oceanography* **26**, 68–81. (doi:10.5670/oceanog.2013.76)
87. Fleming AH, Clark CT, Calambokidis J, Barlow J. 2016 Humpback whale diets respond to variance in ocean climate and ecosystem conditions in the California Current. *Glob. Chang. Biol.* **22**, 1214–1224.

- (doi:10.1111/gcb.13171)
88. Szabo A. 2015 Immature euphausiids do not appear to be prey for humpback whales (*Megaptera novaeangliae*) during spring and summer in Southeast Alaska. *Mar. Mammal Sci.* **31**, 677–687. (doi:10.1111/mms.12183)
 89. Boswell KM, Rieucan G, Vollenweider JJ, Moran JR, Heintz RA, Blackburn JK, Csepp DJ. 2016 Are spatial and temporal patterns in Lynn Canal overwintering Pacific herring related to top predator activity? *Can. J. Fish. Aquat. Sci.* **73**, 1307–1318. (doi:10.1139/cjfas-2015-0192)
 90. Krieger KJ, Wing BL. 1986 NOAA Technical Memorandum NMFS F / NWC-98 Hydroacoustic Monitoring of Prey to Determine Humpback Whale Movements. NOAA Technical Memorandum NMFS F/NWC-98.
 91. Witteveen BH, Foy RJ, Wynne KM, Tremblay Y. 2008 Investigation of foraging habits and prey selection by humpback whales (*Megaptera novaeangliae*) using acoustic tags and concurrent fish surveys. *Mar. Mammal Sci.* **24**, 516–534. (doi:10.1111/j.1748-7692.2008.00193.x)
 92. Szabo A, Batchelder H. 2014 Late spring and summer patterns of euphausiid reproduction in Southeast Alaska fjord waters. *Mar. Ecol. Prog. Ser.* **516**, 153–161. (doi:10.3354/meps11003)
 93. Ward EJ, Adkison M, Couture J, Dressel SC, Litzow MA, Moffitt S, Hoem Neher T, Trochta J, Brenner R. 2017 Evaluating signals of oil spill impacts, climate, and species interactions in Pacific herring and Pacific salmon populations in Prince William Sound and Copper River, Alaska. *PLoS One* **12**, e0172898. (doi:10.1371/journal.pone.0172898)
 94. Fergusson E, Orsi J, Gray A. 2017 Long-term zooplankton and temperature trends in Icy Strait, Southeast Alaska. In *Ecosystem Considerations 2017, Status of the Gulf of Alaska Marine Ecosystem, Stock Assessment and Fishery Evaluation Report*. (eds S Zador, E Yasumiishi), pp. 92–94. Anchorage, Alaska: North Pacific Fishery Management Council.
 95. Kintisch E. 2015 Marine science. ‘The Blob’ invades Pacific, flummoxing climate experts. *Science* **348**, 17–18. (doi:10.1126/science.348.6230.17)
 96. Craig AS, Herman LM. 1997 Sex differences in site fidelity and migration of humpback whales (*Megaptera novaeangliae*) to the Hawaiian Islands. *Can. J. Zool.* **75**, 1923–1933. (doi:10.1139/z97-822)
 97. Craig AS, Herman LM, Pack AA. 2002 Male mate choice and male-male competition coexist in the humpback whale (*Megaptera novaeangliae*). *Can. J. Zool.* **80**, 745–755. (doi:10.1139/z02-050)
 98. Roman J, Doughty CE, Roman J, Faurby S, Wolf A, Haque A, Bakker ES. 2015 Global nutrient transport in a world of giants. (doi:10.1073/pnas.1502549112)
 99. Roman J, Mccarthy JJ. 2010 The Whale Pump : Marine Mammals Enhance Primary Productivity in a Coastal Basin. *PLoS One* **5**, e13255. (doi:10.1371/journal.pone.0013255)
 100. National Marine Fisheries Service. 2016 Post-Delisting Monitoring Plan for Nine Distinct Population Segments of the Humpback Whale (*Megaptera novaeangliae*). National Marine Fisheries Service, Office of Protected Resources, Silver Spring, MD. 25 pp. + Appendices.

Tables

Table 1: Survey dates for transect surveys conducted in the Au'au Channel, Maui, 2008-2018

Season	Year	Timing of surveys	Mean Julian Date
		(Start date - end date)	(Based on Jan 1)
Early	2013 - 2014	1/18 - 2/5	24
	2017 - 2018	1/15 - 2/6	25
Mid	2013 - 2014	2/19 - 3/2	55
	2017 - 2018	2/15 - 2/25	52
Late	2008 - 2010	3/16 - 3/29	81
	2013 - 2014	3/18 - 3/27	81
	2016	3/19 - 3/22	80
	2017 - 2018	3/19 - 3/22	80

Rachel 12/20/2018 5:25 PM

Deleted: 6

Table 2: Locations of humpback whales classified by social composition in the Au'au Channel, Maui, between 2008 and 2018

Season	Year	Water Depth		Distance from shore	
		Mean (s.d.) m		Mean (s.d.) km	
		Mother-calf groups	Adult groups	Mother-calf groups	Adult groups
Early	2013 - 2014	59.0 (13.4)	61.7 (9.7)	4.68 (2.31) ^a	6.03 (2.04) ^a
	2017 - 2018	72.6 (1.95)	59.4 (12.6)	6.97 (0.95) ^a	5.74 (1.83) ^a
Mid	2013 - 2014	61.5 (12.4)	59.8 (13.9)	4.52 (1.86) ^b	4.89 (1.88) ^b
	2017 - 2018	50.5 (15.7)	62.0 (10.1)	2.66 (2.26) ^b	5.05 (2.12) ^b
Late	2008 - 2010	58.9 (14.3)	62.8 (10.91)	4.65 (2.27)	5.03 (1.97)
	2013 - 2014	53.0 (14.2)	57.1 (18.5)	3.59 (2.18)	5.57 (2.67)
	2017 - 2018	66.3 (9.6)	61.8 (15.8)	6.16 (2.08)	5.73 (1.89)

Table 3: Encounter rates for humpback whale groups classified by group composition, sighted during transect surveys conducted in the Au'au Channel, Maui, 2008-2018

Year	All groups	Calf groups	% Calf groups	Distance surveyed (km)	Encounter rate (mother-calf groups km ⁻¹)	Encounter rate (adult groups km ⁻¹)
Brackets						
2013-2014	149	89	59.7	260.2	0.34	0.23
2017-2018	65	23	35.5	292.5	0.07	0.14
Entire seasons						
2013	115	68	59.1	168.0	0.40	0.28
2014	34	21	61.8	92.2	0.23	0.14
2017	41	17	41.5	146.7	0.11	0.16
2018	24	6	25.0	145.9	0.03	0.12
Late season only						
2008	44	27	61.4	270.5	0.10	0.06
2009	77	42	54.5	355.3	0.12	0.10
2010	27	17	63.0	105.3	0.16	0.09
2013	23	17	73.9	52.5	0.32	0.11
2014	12	9	75.0	34.2	0.26	0.09
2016	4	3	75.0	50.3	0.06	0.02
2017	7	6	85.7	44.9	0.13	0.02
2018	3	2	66.7	49.7	0.04	0.02

Table 4: Generalized additive models describing temporal trends in encounter rates for humpback whales in the Au’Au Channel, Maui, Hawaii

Explanatory variables	Deviance		
	Explained (%)	AICc	ΔAICc
Encounter rates – all social groups			
s(year, 2.34)	49.9	22.80	0.00
Factor (season)	28.1	27.92	5.12
Encounter rates –mother- calf groups			
s(year, 2.37)	64.6	-29.33	0.00
Factor (season)	7.5	-11.71	14.61

Rachel 12/21/2018 6:02 AM
Deleted: Table 5: Encounter rates for line surveys conducted in the mid-channel waters of the Au’Au Channel, Maui, 2018
 Distance travelled (km) ... (13)

Table 6a: Generalized additive models describing the influence of oceanic climate conditions in the North Pacific on encounter rates for humpback whales in the Au’Au Channel, Maui, Hawaii

Explanatory variables	Indices advanced by 1 year			Indices advanced by 2 years		
	Deviance Explained	AICc	ΔAICc	Deviance Explained	AICc	ΔAICc

	(%)			(%)		
Encounter rates – individual whales						
s(SST, ~)	<0.01	29.20	3.31	<0.01	28.91	7.17
s(PDO, 0.867)	27.0	25.89	0.00	55.5	21.74	0.00
s(ONI, 1.713)	21.3	28.79	2.90	43.3	25.61	3.87
s(TNI, 2.563)	23.4	30.05	4.19	43.6	25.65	3.91
Encounter rates – mother-calf groups						
s(SST, 2.58)	21.3	-13.07	6.35	<0.01	-14.78	11.72
s(PDO, 1.39)	35.5	-19.42	0.00	66.8	-26.50	0.00
s(ONI, 0)	<0.01	-14.46	4.96	53.2	-20.89	5.61
s(TNI, 2.77)	34.3	-15.62	3.77	46.2	-18.79	7.71
Encounter rates – adult only groups						
s(SST, ~)	<0.01	-21.68	0.27	<0.01	-21.47	0.21
s(PDO, 0.38)	6.21	-21.70	0.25	0.85	-21.64	0.04
s(ONI, 1.59)	18.2	-21.95	0.00	<0.01	-21.68	0.00
s(TNI, ~)	<0.01	-21.68	0.27	27.0	-21.42	0.26

SST: Sea Surface Temperature

PDO: Pacific Decadal Oscillation

ONI: Oceanic Niño Index

TNI: Trans-Niño Index

Figures

Figure 1: The study area

Figure 2: Encounter rates for humpback whales (counts of individuals) in the Au'au Channel, Maui, between 2008 and 2018

rachel 1/5/2019 3:45 PM
Comment [5]: The horizontal axis has been broken to emphasize breaks in the time series

Figure 3: Encounter rates for a) mother- calf groups and b) adult groups during transect surveys of the Au'au Channel, Maui, between 2008 and 2018

rachel 1/5/2019 3:45 PM
Comment [6]: Horizontal axis edited to emphasize breaks in the time series

Figure 4: Non-linear trends in encounter rates for a) all individual whales and b) mother-calf groups during transect surveys of the Au'Au Channel, Maui, between 2008 and 2018

a) All individuals

b) Mother-calf groups

Figure 5: Mean composite sea surface temperatures for the winter season (January to March) for the Au'au Channel, 2008-2018.

While there was no evidence of any linear trend (Pearson's $CC = 0.540$, $p = 0.087$), when pre-2014 SST's (from 2008 to 2013) are compared to post 2014 SST's (from 2014 to 2018), the difference between these two groups was significant ($+ 0.6^{\circ}\text{C}$; using a two sample t-test; $t = -2.952$, D.F. = 9, $p = 0.016$).

Figure 6: Mean annual values for key climate indices that influence oceanographic conditions in the North Pacific, between 2008 and 2018.

Figure 7: Trends in encounter rates in relation to changes in the Pacific Decadal Oscillation index for a) all individual whales and b) mother-calf groups during transect surveys of the Au’Au Channel, Maui, between 2008 and 2018.

a) All individual encounter rates

b) Mother-calf encounter rates

Appendix D

Response to reviewers

In response to the comments received, the following changes have been made:

A slight change in the wording of the title has been made. Permission for this change was requested while the paper was in final review and this change was agreed to by the editor. The new title is a little less definitive. It better reflects the content on the study, and allows for the possibility that the direction of changes seen might change in future years.

The new title reads as follows:

*“Fluctuating reproductive rates in Hawaii’s humpback whales, *Megaptera novaeangliae*, reflect recent climate anomalies in the North Pacific”*

In response to comments from the associate editor:

Comment: I would change this sentence in the abstract to keep all the verbs in agreement, from “Rates initially increased, tracking projected growth rates for this population segment, reaching a peak in 2013, and then declining through 2018.” To “Rates initially increased, tracking projected growth rates for this population segment, reached a peak in 2013, and then declined through 2018.

The grammatical improvements suggested for the abstract have been made so that the verbs are in agreement. The new sentence now reads “. Initially rates increased, tracking projected growth rates for this population segment. Rates reached a peak in 2013, then declined through 2018.

Additionally, the text “ for humpback whales” in the previous line has been removed, as it was un-necessary and repetitious. The tense for the final sentence has also been changed to align with the rest of the abstract.

P5 line 125 – *I think you need to add it to the text, maybe after this sentence, “The study spans the January through March season from 2008 to 2018, however timing...”*

The wording has been changed as suggested, - more specific details on timing have been added so the sentence now reads “The study spans the wintertime (January through March) season from 2008 to 2018” . The term wintertime is already in use

elsewhere in the text, so this provides an easy point of reference. The second segment of the sentence has been edited slightly to maintain the flow of the text and ensure that the first sentence isn't too long.

Line 129 – *Survey protocols used in 2008 to 2010 were repeated in 2013 and 2014 to allow comparisons of the results.*

Text altered to clarify the meaning as suggested. New text reads “Survey protocols used in 2008 to 2010 were repeated in 2013 and 2014 to allow comparisons of the results”.

Line 132 – **typo altered**

Line 133 – *Shouldn't this be <4 or less than or equal to 3? I understood from your response that surveys were conducted in Beaufort = 3.*

The following statement was included to clarify the constraints placed in survey conditions. Surveys were conducted in conditions of Beaufort 3 or less – now stated as Beaufort \leq 3

Line 305 – *In a second set of models, the impact of key oceanic parameters (impact on what? On variation in whale encounter rates?)*

The purpose of the second set of models is now explained. The new text now reads “In a second set of models, the impact of key oceanic parameters on variations in encounter rates was explored”. Underlined text was added

Response to reviewer 1 – the typos mentioned in lines 129 and 133 have been corrected.

Additionally, in reviewing the paper prior to final submission, I noted several places where small grammatical or minor changes could improve the flow of the text – these minor changes are listed below:

Line 13 -14 – Abstract – As the study now refers entirely to previous seasons (i.e. 2008 to 2018), I changed the tense and the word order slightly, to reflect this and keep the tense consistent in the abstract.

Line 69 – change in text so that the word period is not used twice in the same sentence.

Line 126 – “ over the course of the study” replaced by “between years”

Line 159 –The use of 2017-2018 surveys in the habitat analysis is clarified and explained on line 282, so here in the method, I have kept to more chronological overview. I think that this better explains the initial purpose of each the three sets of surveys.

Line 283 – the following text was added to explain the use of the 2017-2018 survey data in the habitat analysis. “Survey data from 2017-2018 was also incorporated as this had been collected following the same protocols”.

Line 562 – the term on-going has been removed

Line 616 – new paragraph break inserted

Line 618 – slight change in word order to improve the flow of the text

Line 625 – Breakdown of a run on sentence.

Line 631 – The original sentence was not well –written, there was a word missing that made the sentence hard to follow. The new sentence reads as follows:
Supporting this conclusion, the wide range of unusual mortality and mass casualty events in unrelated fauna reported from across the Gulf of Alaska and connected marine systems have also been attributed to the reduced productivity and prey resources currently evident in this region

Line 636 – Heading changed to a question format for consistency throughout the discussion

Line 654 – small changes in sentence structure to reduce the length of the final sentence

Line 732 – the order of the final statements has been changed to match the order provided in the screenshot sent with the decision letter.

Line 744 – full details of permit authorities have been included.